# Graph Neural Networks with Local Graph Parameters

**Pablo Barceló[1,2], Floris Geerts[3], Juan Reutter[1,2], Maksimilian Ryschkov[3]**
[1] Department of Computer Science, PUC, Chile
[2] Millennium Institute for Foundational Research on Data, Chile
[3] Department of Computer Science, University of Antwerp, Belgium
[pbarcelo,jreutter]@ing.puc.cl, [floris.geerts,maksimilian.ryschkov]@uantwerpen.be

## Abstract

Various recent proposals increase the distinguishing power of Graph Neural Networks (GNNs) by propagating features between $k$-tuples of vertices. The distinguishing power of these "higher-order" GNNs is known to be bounded by the $k$-dimensional Weisfeiler-Leman (WL) test, yet their $\mathcal{O}(n^k)$ memory requirements limit their applicability. Other proposals infuse GNNs with local higher-order graph structural information from the start, hereby inheriting the desirable $\mathcal{O}(n)$ memory requirement from GNNs at the cost of a one-time, possibly non-linear, preprocessing step. We propose local graph parameter enabled GNNs as a framework for studying the latter kind of approaches. We precisely characterize their distinguishing power, in terms of a variant of the WL test, and in terms of the graph structural properties that they can take into account. Local graph parameters can be added to any GNN architecture, and are cheap to compute. In terms of expressive power, our proposal lies in the middle of GNNs and their higher-order counterparts. Further, we propose several techniques to aid in choosing the right local graph parameters. Our results connect GNNs with deep results in finite model theory and finite variable logics. Our experimental evaluation shows that adding local graph parameters often has a positive effect on a variety of GNNs, datasets and graph learning tasks.

## 1 Introduction

**Context.** Graph neural networks (GNNs) [Merkwirth and Lengauer, 2005, Scarselli et al., 2009], and its important class of Message Passing Neural Networks (MPNNs) [Gilmer et al., 2017], are one of the most popular methods for graph learning tasks. Such MPNNs use an iterative message passing scheme, based on the adjacency structure of the underlying graph, to compute vertex (and graph) embeddings in some real Euclidean space.

The expressive (or distinguishing) power of MPNNs is, however, rather limited [Morris et al., 2019, Xu et al., 2019]. Indeed, MPNNs will always identically embed two vertices (graphs) when these vertices (graphs) cannot be distinguished by the one-dimensional Weisfeiler-Leman (WL) algorithm. Two graphs $G_1$ and $H_1$ and vertices $v$ and $w$ that cannot be distinguished by WL (and thus any MPNN) are shown in Fig. 1. The expressive power of WL is well-understood [Cai et al., 1992, Dell et al., 2018, Arvind et al., 2020] and basically can only use *tree-based* structural information in the graphs to distinguish vertices. Hence, no MPNN can detect that vertex $v$ in Fig. 1 is part of a 3-clique, whereas $w$ is not. Similarly, MPNNs cannot detect that $w$ is part of a 4-cycle, whereas $v$ is not. Further limitations of WL in terms of graph properties can be found, e.g., in Arvind et al. [2020], Chen et al. [2020] and Tahmasebi and Jegelka [2020].

To remedy the weak expressive power of MPNNs, so-called *higher-order* MPNNs were proposed [Maron et al., 2019a, Morris et al., 2019, 2020], whose expressive power is well-understood and

measured in terms of the $k$-dimensional WL procedures ($k$-WL) [Maron et al., 2019a, Chen et al., 2019a, Geerts, 2020, Sato, 2020, Azizian and Lelarge, 2021]. In a nutshell, $k$-WL operates on $k$-tuples of vertices and allows to distinguish vertices (graphs) based on structural information related to *graphs of treewidth $k$* [Dvorak, 2010, Dell et al., 2018]. By definition, WL = 1-WL. As an example, 2-WL can detect that vertex $v$ in Fig. 1 belongs to a 3-clique or a 4-cycle since both have treewidth two. While more expressive than WL, the GNNs based on $k$-WL require $\mathcal{O}(n^k)$ operations in *each iteration*, where $n$ is the number of vertices, hereby hampering their applicability.

A more practical approach is to extend the expressive power of MPNNs *whilst preserving their $\mathcal{O}(n)$ cost in each iteration*. Various such extensions [Kipf and Welling, 2017, Chen et al., 2019a, Li et al., 2019, Ishiguro et al., 2020, Bouritsas et al., 2020, Geerts et al., 2021] achieve this by infusing MPNNs with *local graph structural information from the start*. That is, the iterative message passing scheme of MPNNs is run on vertex labels that contain quantitative information about local graph structures.

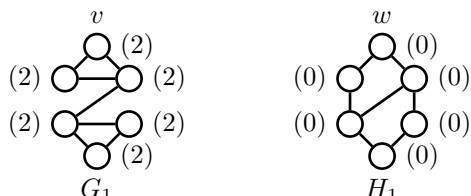

Figure 1: Two graphs that are indistinguishable by the WL-test. The numbers between round brackets indicate how many homomorphic images of the 3-clique each vertex is involved in.

It is easy to see that such architectures can go beyond the WL test: for example, adding triangle counts to MPNNs suffices to distinguish the vertices $v$ and $w$ and graphs $G_1$ and $H_1$ in Fig. 1. Moreover, the cost is *a single preprocessing step* to count local graph parameters, thus maintaining the $\mathcal{O}(n)$ cost in the iterations of the MPNN. While there are some partial results showing that local graph parameters increase expressive power [Bouritsas et al., 2020, Li et al., 2019], their precise expressive power and relationship to higher-order MPNNs was unknown, and there is little guidance in terms of which local parameters do help MPNNs and which ones do not. The main contribution of this paper is a precise characterization of the expressive power of MPNNs with local graph parameters and its relationship to the hierarchy of higher-order MPNNs.

**Our contributions.**  In order to nicely formalize local graph parameters, we propose to extend vertex labels with *homomorphism counts* of small graph patterns.[1] More precisely, given graphs $P$ and $G$, and vertices $r$ in $P$ and $v$ in $G$, we propose to augment the initial features of $v$ with the number of homomorphisms from $P$ to $G$ that map $r$ to $v$, denoted by $\mathrm{hom}(P^r, G^v)$, as a way to capture local structural information. More generally, homomorphism counts for a collection of graphs are considered. Indeed, we propose $\mathcal{F}$-MPNNs where $\mathcal{F} = \{P_1^r, \ldots, P_\ell^r\}$ is a set of (graph) patterns, which extend MPNNs by (i) first allowing a preprocessing step that labels each vertex $v$ of a graph $G$ with the vector $\big(\mathrm{hom}(P_1^r, G^v), \ldots, \mathrm{hom}(P_\ell^r, G^v)\big)$, and (ii) then run an MPNN on this labelling. Our main contributions are the following:

**1.** We precisely characterize the expressive power of $\mathcal{F}$-MPNNs by means of an extension of WL, denoted by $\mathcal{F}$-WL. This characterization gracefully extends the characterization for standard MPNNs, mentioned earlier, by setting $\mathcal{F} = \emptyset$, and provides insights in the expressive power of existing MPNN extensions, most notably the Graph Substructure Networks of Bouritsas et al. [2020].

**2.** We compare $\mathcal{F}$-MPNNs to higher-order MPNNs, which are characterized in terms of the $k$-WL-test. On the one hand, while $\mathcal{F}$-MPNNs strictly increase the expressive power of the WL-test, for any finite set $\mathcal{F}$ of patterns, 2-WL can distinguish graphs which $\mathcal{F}$-MPNNs cannot. On the other hand, for each $k \geq 1$ there are patterns $P$ such that $\{P\}$-MPNNs can distinguish graphs which $k$-WL cannot.

**3.** We deal with the challenging problem of pattern selection and comparing $\mathcal{F}$-MPNNs based on the patterns included in $\mathcal{F}$. We prove two partial results: one establishing when a pattern $P$ in $\mathcal{F}$ is redundant, and another result indicating when $P$ does add expressive power, based on the treewidth of $P$ compared to the treewidth of other patterns in $\mathcal{F}$.

**4.** Our theoretical results are complemented by an experimental study in which we show that for various GNN architectures, datasets and graph learning tasks, all part of the recent benchmark by Dwivedi et al. [2020], the augmentation of initial features with homomorphism counts of graph patterns has often a positive effect, and the cost for computing these counts incurs little to no overhead.

---

[1]We recall that homomorphisms are edge-preserving mappings between the vertex sets.

As such, we believe that $\mathcal{F}$-MPNNs not only provide an elegant theoretical framework for understanding local graph parameter enabled MPNNs, they are also a valuable alternative to higher-order MPNNs as a way to increase the expressive power of MPNNs. In addition, and as will be explained in Section 2, $\mathcal{F}$-MPNNs provide a unifying framework for understanding the expressive power of several other existing extensions of MPNNs. Proofs of our results and further details on the relationship to existing approaches and experiments can be found in the supplementary material.

**Related Work.** Works related to the distinguishing power of the WL-test, MPNNs and their higher-order variants are cited throughout the paper. Beyond distinguishability, GNNs are analyzed in terms of universality and generalization properties [Maron et al., 2019b, Keriven and Peyré, 2019, Chen et al., 2019b, Garg et al., 2020, Azizian and Lelarge, 2021], local distributed algorithms [Sato et al., 2019, Loukas, 2020], randomness in features [Sato et al., 2021, Abboud et al., 2021] and using local context matrix features [Vignac et al., 2020]. Other extensions of GNNs are surveyed, e.g., in Wu et al. [2021], Zhou et al. [2018] and Chami et al. [2021]. Related are also the Graph Homomorphism Convolutions by NT and Maehara [2020] which apply SVMs directly on a homomorphism count representation of vertices. Finally, our approach is reminiscent of graph representations by means of graphlet kernels [Shervashidze et al., 2009], but then on the level of vertices.

**Limitations of our approach.** One of the limitations of $\mathcal{F}$-MPNNs is that their expressive power depends on the set $\mathcal{F}$ of patterns. Our work offers tools and guidelines to help in this search, but the best set of patterns must still be found by trial-and-error. However, as we show in Section 6, MPNNs almost always benefit from any set of additional features, assuming that the resulting model goes beyond the WL test. Cliques and cycles are two types of simple patterns that are guaranteed to extend the WL test, and indeed we show that striking gains can be obtained by simply adding these features to existing benchmarks. For simplicity of exposition we focus on vertex-labelled undirected graphs but all our results can be extended to edge-labelled directed graphs.

## 2 Local Graph Parameter Enabled MPNNs

We here introduce our MPNNs with local graph parameters. We first recall some graph concepts.

**Graphs.** We consider undirected vertex-labelled graphs $G = (V, E, \chi)$, with $V$ the set of vertices, $E$ the set of edges and $\chi$ a mapping assigning a label to each vertex in $V$. The set of neighbors of a vertex is denoted by $N_G(v) = \{u \in V \mid \{u, v\} \in E\}$. A *rooted graph* is a graph in which one of its vertices is declared as its root. We denote a rooted graph by $G^v$, where $v \in V$ is the root and depict them as graphs in which the root is a blackened vertex, such as e.g., ⬡. Given graphs $G = (V_G, E_G, \chi_G)$ and $H = (V_H, E_H, \chi_H)$, an *homomorphism h* is a mapping $h : V_G \to V_H$ such that (i) $\{h(u), h(v)\} \in E_H$ for every $\{u, v\} \in E_G$, and (ii) $\chi_G(u) = \chi_H(h(u))$ for every $u \in V_G$. For rooted graphs $G^v$ and $H^w$, an homomorphism must additionally map $v$ to $w$. We denote by $\mathrm{hom}(G, H)$ the number of homomorphisms from $G$ to $H$; similarly for rooted graphs.

**MPNNs with local graph parameters.** Let $\mathcal{F} = \{P_1^r, \ldots, P_\ell^r\}$ be a set of rooted graphs, which we refer to as *patterns*. As a uniform formalization of local graph structural information, we propose the use of *homomorphism counts* of patterns in $\mathcal{F}$ to enhance the initial feature of vertices. To illustrate the idea, consider the graphs in Fig. 1. As mentioned, these graphs cannot be distinguished by the WL-test, and therefore cannot be distinguished by the broad class of MPNNs. If we allow a *preprocessing stage*, however, in which the initial labelling of every vertex $v$ is extended with the number of (homomorphic images of) 3-cliques in which $v$ participates (indicated by numbers between brackets in Fig. 1), then clearly vertices $v$ and $w$ (and the graphs $G_1$ and $H_1$) can be distinguished based on this extra structural information. In fact, the initial labelling already suffices for this purpose. In our setting, this will correspond to selecting the rooted 3-clique ⬡ and attach to each vertex $v$, $\mathrm{hom}(⬡, G^v)$. Information about 4-cycles requires adding $\mathrm{hom}(⬡, G^v)$ as well.

We therefore propose $\mathcal{F}$-*enabled* MPNNs, or just $\mathcal{F}$-MPNNs, defined in the same way as MPNNs [Gilmer et al., 2017] with the crucial difference that the initial feature vector of a vertex $v$ (a one-hot encoding of its label $\chi_G(v)$) is augmented with all the homomorphism counts from patterns in $\mathcal{F}$.

Formally, in each round $d$ an $\mathcal{F}$-MPNN $M$ labels each vertex $v$ in graph $G$ with a feature vector $\mathbf{x}_{M,\mathcal{F},G,v}^{(d)}$ which is inductively defined as follows:

$$\mathbf{x}_{M,\mathcal{F},G,v}^{(0)} := \big(\chi_G(v), \mathsf{hom}(P_1^r, G^v), \ldots, \mathsf{hom}(P_\ell^r, G^v)\big)$$

$$\mathbf{x}_{M,\mathcal{F},G,v}^{(d)} := \textsc{Upd}^{(d)}\Big(\mathbf{x}_{M,\mathcal{F},G,v}^{(d-1)}, \textsc{Comb}^{(d)}\big(\{\!\{\mathbf{x}_{M,\mathcal{F},G,v}^{(d-1)} \mid u \in N_G(v)\}\!\}\big)\Big), \text{ for } d > 0,$$

where $\textsc{Comb}^{(d)}$ and $\textsc{Upd}^{(d)}$ are an *aggregating* and *update* function, respectively, as in standard MPNNs, and where $\{\!\{\}\!\}$ denotes a multi-set. We note that standard MPNNs are $\mathcal{F}$-MPNNs with $\mathcal{F} = \emptyset$. As for MPNNs, we can equip $\mathcal{F}$-MPNNs with a READOUT function that aggregates all final feature vectors into a single feature vector in order to classify or distinguish graphs.

We remark that *any* MPNN architecture can be turned into an $\mathcal{F}$-MPNN by a simple homomorphism counting preprocessing step. As such, we propose a *generic plug-in for a large class of* GNN *architectures*. Better still, $\mathsf{hom}(P^r, G^v)$ can be computed in time $\mathcal{O}(|V_G|^{\mathsf{tw}(P^r)+1})$ [Díaz et al., 2002], where $\mathsf{tw}(P^r)$ denotes the treewidth of pattern $P^r$. The treewidth of patterns used in $\mathcal{F}$-MPNNs is typically small. And indeed, homomorphism counts of small graph patterns can be efficiently computed in practice, even on large datasets [Zhang et al., 2020]. We also remark that the use of rooted patterns is important because different vertices in a pattern may embed differently around a target vertex in a graph. For example, $\mathsf{hom}(\,\rule{0pt}{0pt}, G^v)$ can be different from $\mathsf{hom}(\,\rule{0pt}{0pt}, G^v)$. The choice of a root in a graph $P$ can be avoided by including different rooted versions of $P$ in $\mathcal{F}$. In fact, it suffices to include one rooted version $P^r$ for each vertex $r$ in a distinct orbit in $P$. We note that for symmetric graphs, such as cliques and cycles, all vertices lie in the same orbit and a single, arbitrary, choice of root vertex suffices. Alternatively, one could define $\mathsf{hom}(P, G^v) := \sum_{r \in V_P} \mathsf{hom}(P^r, G^v)$ which ignores how the pattern is locally mapped into a graph. We speculate, however, that this results in a model that is less powerful than $\mathcal{F}$-MPNNs.

Despite their simplicity, we will show that $\mathcal{F}$-MPNNs can substantially increase the power of MPNNs by varying $\mathcal{F}$, only paying a one-time preprocessing cost.

**$\mathcal{F}$-MPNNs as unifying framework.** An important aspect of $\mathcal{F}$-MPNNs is that they *allow a principled analysis of the power of existing extensions of* MPNN*s*. For example, taking $\mathcal{F} = \{\,\rule{0pt}{0pt}\}$ suffices to capture degree-aware MPNNs [Geerts et al., 2021], such as the Graph Convolution Networks (GCNs) [Kipf and Welling, 2017], which use the *degree of vertices*; taking $\mathcal{F} = \{L_1, L_2, \ldots, L_\ell\}$ for rooted paths $L_i$ of length $i$ suffices to model the *walk counts* used in Chen et al. [2019a]; and taking $\mathcal{F}$ as the set of labeled trees of depth one precisely corresponds to the use of the WL-*labelling obtained after one round* by Ishiguro et al. [2020]. Furthermore, $\{C_\ell\}$-MPNNs, where $C_\ell$ denotes the cycle of length $\ell$, correspond to the extension proposed in Section 4 in Li et al. [2019].

In addition, $\mathcal{F}$-MPNNs are close in spirit to the *Graph Substructure Networks* (GSNs) by Bouritsas et al. [2020], which use subgraph isomorphism counts of graph patterns. We recall that an isomorphism from $G$ to $H$ is a *bijective* homomorphism $h$ from $G$ to $H$ which additionally satisfies (i) $\{h^{-1}(u), h^{-1}(v)\} \in E_G$ for every $\{u, v\} \in E_H$, and (ii) $\chi_G(h^{-1}(u)) = \chi_H(u)$ for every $u \in V_H$. When $G$ and $H$ are rooted graphs, isomorphisms should preserve the roots as well. Now, in a GSN, the feature vector of each vertex $v$ is augmented with the subgraph isomorphism counts, $\mathsf{sub}(P^r, G^v)$, for rooted patterns $P^r$ in a set of $\mathcal{P}$ of patterns, and this is followed by the execution of an MPNN, just as for our $\mathcal{F}$-MPNNs. Importantly, $\mathcal{F}$-MPNNs can be used to bound the expressive power of GSNs, as we will show later. We note that computing $\mathsf{sub}(P^r, G^v)$ is, in general, more costly than computing $\mathsf{hom}(P^r, G^v)$ [Curticapean et al., 2017]. This partially motivates our choice of using homomorphism counts instead of subgraph isomorphism counts. Moreover, homomorphism counts of tree patterns underly existing characterizations of the expressive power of MPNNs [Dell et al., 2018, Grohe, 2020a]. As we will see shortly, these characterizations gracefully extend to $\mathcal{F}$-MPNNs.

## 3 The Expressive Power of $\mathcal{F}$-MPNNs

We next provide exact characterizations of the expressive power of $\mathcal{F}$-MPNNs. Our results extend those for standard MPNNs [Xu et al., 2019, Morris et al., 2019, Dell et al., 2018].

**Characterization in terms of $\mathcal{F}$-WL.** We bound the expressive power of $\mathcal{F}$-MPNNs in terms of what we call the $\mathcal{F}$-WL-*test*. This test extends the WL-test [Weisfeiler and Lehman, 1968, Grohe, 2017] in the same way as $\mathcal{F}$-MPNNs extend standard MPNNs: by including homomorphism counts

of patterns in $\mathcal{F}$ in the initial labelling. The $\mathcal{F}$-WL-test, for $\mathcal{F} = \{P_1^r, \ldots, P_\ell^r\}$, is a vertex labelling algorithm that iteratively computes a label $\chi_{\mathcal{F},G,v}^{(d)}$ for each vertex $v$ of a graph $G$, as follows:

$$\chi_{\mathcal{F},G,v}^{(0)} := \big(\chi_G(v), \mathsf{hom}(P_1^r, G^v), \ldots, \mathsf{hom}(P_\ell^r, G^v)\big)$$
$$\chi_{\mathcal{F},G,v}^{(d)} := \mathrm{HASH}\big(\chi_{\mathcal{F},G,v}^{(d-1)}, \{\!\{\chi_{\mathcal{F},G,u}^{(d-1)} \mid u \in N_G(v)\}\!\}\big), \text{ for } d > 0.$$

The $\mathcal{F}$-WL-test stops in round $d$ when no new pair of vertices are distinguished, that is, $\chi_{\mathcal{F},G,v_1}^{(d-1)} = \chi_{\mathcal{F},G,v_2}^{(d-1)}$ implies $\chi_{\mathcal{F},G,v_1}^{(d)} = \chi_{\mathcal{F},G,v_2}^{(d)}$, for any vertices $v_1$ and $v_2$ in $G$. The standard WL-test corresponds to $\{\emptyset\}$-WL. We can use the $\mathcal{F}$-WL-test to compare vertices of the same graphs, or different graphs. We say that the $\mathcal{F}$-WL-test *cannot distinguish vertices* if their final labels are the same, and that the $\mathcal{F}$-WL-test *cannot distinguish graphs* $G$ and $H$ if the multiset containing each label computed for $G$ is the same as that of $H$. Similarly as for MPNNs and the WL-test [Xu et al., 2019, Morris et al., 2019], the $\mathcal{F}$-WL-test provides an upper bound for the expressiveness of $\mathcal{F}$-MPNNs.

**Proposition 1.** *If two vertices of a graph cannot be distinguished by the $\mathcal{F}$-WL-test, then they cannot be distinguished by any $\mathcal{F}$-MPNN either. Moreover, if two graphs cannot be distinguished by the $\mathcal{F}$-WL-test, then they cannot be distinguished by any $\mathcal{F}$-MPNN either.*

Furthermore, simply adding local parameters from a set $\mathcal{F}$ of patterns to the GIN architecture of Xu et al. [2019] results in $\mathcal{F}$-MPNNs that match the expressive power of the $\mathcal{F}$-WL-test.

**Characterization in terms of $\mathcal{F}$-pattern trees.** At the core of several results about the WL-test lies a characterization linking the test with homomorphism counts of (rooted) trees [Dell et al., 2018, Grohe, 2020a]. In view of the connection to MPNNs, it tells that MPNNs only use *quantitative tree-based* structural information from the underlying graphs. We next extend this characterization to $\mathcal{F}$-WL by using homomorphism counts of so-called $\mathcal{F}$-pattern trees. Proposition 1 then reveals that $\mathcal{F}$-MPNNs can use quantitative information of *richer graph structures* than the trees used by MPNNs.

To define $\mathcal{F}$-pattern trees we need the *graph join operator* $\star$. Given two rooted graphs $G^v$ and $H^w$, the join graph $(G \star H)^v$ is obtained by taking the disjoint union of $G^v$ and $H^w$, followed by identifying $w$ with $v$. The root of the join graph is $v$. For example, the join of ⬡ and ⬡ is ⬡.

Furthermore, if $G$ is a graph and $P^r$ is a rooted graph, then joining a vertex $v$ in $G$ with $P^r$ results in the disjoint union of $G$ and $P^r$, where $r$ is identified with $v$. Let $\mathcal{F} = \{P_1^r, \ldots, P_\ell^r\}$. An $\mathcal{F}$-*pattern tree* $T^r$ is obtained from a standard rooted tree $S^r = (V, E, \chi)$, called the *backbone* of $T^r$, followed by joining every vertex $s \in V$ with any number of copies of patterns from $\mathcal{F}$. Examples of $\mathcal{F}$-pattern trees, for $\mathcal{F} = \{$⬡$\}$ are shown in Fig. 2, where grey colored vertices

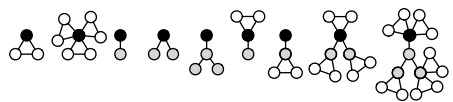

Figure 2: Examples of $\mathcal{F}$-pattern trees.

are part of the backbones of the $\mathcal{F}$-pattern trees. We define the *depth* of an $\mathcal{F}$-pattern tree as the depth of its backbone. Standard trees are $\mathcal{F}$-pattern trees in which no patterns are joined with backbone vertices. We next use $\mathcal{F}$-pattern trees to characterize the expressive power of $\mathcal{F}$-WL and thus, by Proposition 1, of $\mathcal{F}$-MPNNs.

**Theorem 1.** *For any finite collection $\mathcal{F}$ of patterns, vertices $v$ and $w$ in a graph $G$ are indistinguishable by the $\mathcal{F}$-WL-test if and only if $\mathsf{hom}(T^r, G^v) = \mathsf{hom}(T^r, G^w)$ for every rooted $\mathcal{F}$-pattern tree $T^r$. Similarly, $G$ and $H$ are indistinguishable by the $\mathcal{F}$-WL-test if and only if $\mathsf{hom}(T, G) = \mathsf{hom}(T, H)$ for every (unrooted) $\mathcal{F}$-pattern tree $T$.*

The proof of this Theorem requires a generalization of the techniques from Dell et al. [2018] and Grohe [2020a,b], used to characterize the expressiveness of WL in terms of homomorphism counts of trees. In fact, we can use our proof to recover said results simply by setting $\mathcal{F} = \emptyset$. We note that $\mathcal{F}$-MPNNs are more expressive than MPNNs (recall the graphs $G_1$ and $H_1$ and $\mathcal{F} = \{$⬡$\}$).

We can even make the above theorem more precise. When $\mathcal{F}$-WL is run for $d$ rounds, then *only $\mathcal{F}$-patterns trees of depth $d$ are required.* This tells that increasing the number of rounds of $\mathcal{F}$-WL results in that more complicated structural information is taken into account. As an illustration, consider the graphs $G_2$ and $H_2$ and vertices $v$ and $w$ shown in Fig. 3 and let us consider $\mathcal{F}$-WL with $\mathcal{F} = \{$⬡$\}$. We argue that $v$ and $w$ cannot be distinguished by $\mathcal{F}$-WL based on the initial labelling only, but they can be distinguished after one round.

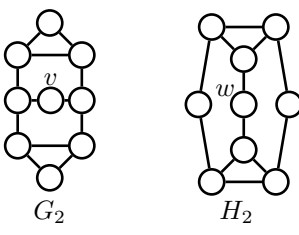

Figure 3: $\{\,\mathord{\vcenter{\hbox{$\cdot$}}}\mathord{\ooalign{}}\,\}$–MPNNs require one round to distinguish $v$ from $w$.

Indeed, by definition, $\mathcal{F}$-WL cannot distinguish $v$ from $w$ based on the initial labelling since these vertices have the same triangle count (zero). Hence, no $\mathcal{F}$-MPNN can distinguish these vertices either. If run for one round, Theorem 1 implies that $\mathcal{F}$-WL cannot distinguish $v$ from $w$ if and only if $\mathsf{hom}(T^r, G_2^v) = \mathsf{hom}(T^r, H_2^w)$ for any $\mathcal{F}$-pattern tree of depth at most 1. It is readily verified that $\mathsf{hom}(\,\mathord{\vcenter{\hbox{$\bullet$}}}\mathord{\ooalign{}}\,, G_2^v) = 0 \neq 4 = \mathsf{hom}(\,\mathord{\vcenter{\hbox{$\bullet$}}}\mathord{\ooalign{}}\,, H_2^w)$, and thus $\mathcal{F}$-WL distinguishes $v$ from $w$ after one round. Also, $G_2$ and $H_2$ can be distinguished by $\mathcal{F}$-WL after one round. Moreover, $G_2$ and $H_2$ are indistinguishable by WL showing again that $\mathcal{F}$-MPNNs are more expressive than MPNNs. The example shows that more rounds allow $\mathcal{F}$-MPNNs to detect more complex patterns based on $\mathcal{F}$-pattern trees. This is in contrast to, e.g., the Graph Homomorphism Convolutions by NT and Maehara [2020] in which only homomorphism counts of patterns in $\mathcal{F}$, and not the counts of the derived $\mathcal{F}$-patterns trees, are taken into account.

**Implications for other** MPNN **extensions.** Importantly, Theorem 1 discloses the boundaries of $\mathcal{F}$-MPNNs. To illustrate this for some specific instances of $\mathcal{F}$-MPNNs mentioned earlier, the expressive power of degree-based MPNNs [Kipf and Welling, 2017, Geerts et al., 2021] is captured by $\{L_1\}$-pattern trees, and walk count MPNNs [Chen et al., 2019a] are captured by $\{L_1, \ldots, L_\ell\}$-pattern trees. These pattern trees are just trees, since joining paths to trees only results in bigger trees. Thus, Theorem 1 tells that all these extensions are still bounded by WL (albeit needing fewer rounds). In contrast, beyond WL, $\{C_\ell\}$-pattern trees capture cycle count MPNNs [Li et al., 2019]. Our results also shed light on the expressive power of GSNs [Bouritsas et al., 2020] which, as already mentioned, use subgraph isomorphism counts of patterns $P \in \mathcal{P}$. More precisely, a $\mathcal{P}$-GSN augments vertex features by including $\mathsf{sub}(P^r, G^v)$ for rooted versions $P^r$ of $P \in \mathcal{P}$, where $r$ ranges over representative vertices of the different orbits in $P$. Let us denote by $\mathcal{P}^+$ this set of rooted patterns.

**Fact 1** (Curticapean et al. [2017]). *For any set $\mathcal{P}$ of rooted patterns, there exists a pattern set $\mathsf{s}(\mathcal{P})$ such that for each $P^r \in \mathcal{P}$, $\mathsf{sub}(P^r, G^v)$ can be computed based on $\{\mathsf{hom}(Q^r, G^v) \mid Q^r \in \mathsf{s}(\mathcal{P})\}$.*

In other words, subgraph isomorphism counts can be computed in terms of homomorphism counts, albeit by using different pattern sets. For example, if $\mathcal{P} = \{\,\mathord{\vcenter{\hbox{$\bullet$}}}\,\mathord{\vcenter{\hbox{$\circ$}}}\mathord{\ooalign{}}\,\}$, then $\mathsf{s}(\mathcal{P}) = \{\,\mathord{\vcenter{\hbox{$\bullet$}}}\,\mathord{\vcenter{\hbox{$\circ$}}}\mathord{\ooalign{}}, \mathord{\vcenter{\hbox{$\bullet$}}}\mathord{\ooalign{}}, \mathord{\vcenter{\hbox{$\bullet$}}}\mathord{\ooalign{}}, \mathord{\ooalign{}}\,\}$ which contains all different homomorphic images of $\,\mathord{\vcenter{\hbox{$\bullet$}}}\,\mathord{\vcenter{\hbox{$\circ$}}}\mathord{\ooalign{}}$. As a consequence, we obtain:

**Proposition 2.** $\mathcal{P}$-GSN*s cannot distinguish more vertices than* $\mathsf{s}(\mathcal{P}^+)$-MPNN*s can.*

So, we can bound the expressive power of GSNs in terms of $\mathcal{F}$-MPNNs. In the supplementary material we also show that $\mathcal{F}$-MPNNs are bounded by GSNs that use some special set of patterns derived from $\mathcal{F}$. We conclude by observing that the proof technique underlying Theorem 1 can be directly applied to GSNs. The key insight is to redefine the notion of homomorphism from pattern trees to graphs. Indeed, consider a $\mathcal{P}$-GSN and a corresponding $\mathcal{P}^+$-pattern tree $T^r$. We define $\mathsf{shom}(T^r, G^v)$ as the number of mappings $h : V_T \to V_G$ such that (i) $h(r) = v$; (ii) $h$ is a homomorphism when restricted to the backbone of $T^r$; and (iii) for each $P^r$ joined with a backbone vertex, $h(P^r)$ is *isomorphic* to $P^r$. We then have the following counterpart of Theorem 1 for GSNs:

**Theorem 2.** *For any finite collection $\mathcal{P}$ of patterns, vertices $v$ and $w$ in a graph $G$ are indistinguishable by $\mathcal{P}$-GSNs if and only if $\mathsf{shom}(T^r, G^v) = \mathsf{shom}(T^r, G^w)$ for every rooted $\mathcal{P}^+$-pattern tree $T^r$. Similarly, $G$ and $H$ are indistinguishable by $\mathcal{P}$-GSNs if and only if $\mathsf{shom}(T, G) = \mathsf{shom}(T, H)$ for every (unrooted) $\mathcal{P}^+$-pattern tree $T$.*

## 4 A Comparison with the $k$-WL-test

Compared to the computationally intensive higher-order MPNNs based on the $k$-WL-tests [Maron et al., 2019a, Morris et al., 2019, 2020], $\mathcal{F}$-MPNNs are an alternative and efficient way to extend the expressive power of MPNNs (and thus the WL-test). In this section we situate $\mathcal{F}$-WL in the $k$-WL hierarchy. The definition of the $k$-WL-test can be found, e.g., in Morris et al. [2020], and is provided in the supplementary material as well. We also use the standard notion of *treewidth* of a graph (see e.g., Bodlaender [1993]). Intuitively, treewidth measures the tree-likeness of a graph. For example, trees have treewidth one, cycles have treewidth two, and the $k$-clique $K_k$ has treewidth $k - 1$. Furthermore, the treewidth of a pattern $P^r$ is the treewidth of its unrooted version $P$.

We have seen that $\mathcal{F}$-WL can distinguish graphs that WL cannot: just consider $\{K_3\}$-WL for the 3-clique $K_3$. To generalize this observation we need some notation. Let $\mathcal{F}$ and $\mathcal{G}$ be two sets of patterns and consider an $\mathcal{F}$-MPNN $M$ and a $\mathcal{G}$-MPNN $N$. We say that $M$ is *upper bounded in expressive power* by $N$ if for any graph $G$, if $N$ cannot distinguish vertices $v$ and $w$,[2] then neither can $M$. A similar notion is in place for pairs of graphs: if $N$ cannot distinguish graphs $G$ and $H$, then neither can $M$. More generally, let $\mathcal{M}$ be a class of $\mathcal{F}$-MPNNs and $\mathcal{N}$ be a class of $\mathcal{G}$-MPNNs. We say that the class $\mathcal{M}$ is *upper bounded in expressive power* by $\mathcal{N}$ if every $M \in \mathcal{M}$ is upper bounded in expressive power by an $N \in \mathcal{N}$ (which may depend on $M$). When $\mathcal{M}$ is upper bounded by $\mathcal{N}$ and vice versa, then $\mathcal{M}$ and $\mathcal{N}$ are said to have the *same expressive power*. A class $\mathcal{N}$ is *more expressive* than a class $\mathcal{M}$ when $\mathcal{M}$ is upper bounded in expressive power by $\mathcal{N}$, but there exist graphs that can be distinguished by MPNNs in $\mathcal{N}$ but not by any MPNN in $\mathcal{M}$.

Our first result is a consequence of the characterization of $k$-WL in terms of homomorphism counts of graphs of treewidth $k$ [Dvorak, 2010, Dell et al., 2018].

**Proposition 3.** *For each finite set $\mathcal{F}$ of patterns, the expressive power of $\mathcal{F}$-WL is bounded by $k$-WL, where $k$ is the largest treewidth of a pattern in $\mathcal{F}$.*

For example, since the treewidth of $K_3$ is 2, we have that $\{K_3\}$-WL is bounded by 2-WL. Similarly, $\{K_{k+1}\}$-WL is bounded in expressive power by $k$-WL.

Our second result tells how to increase the expressive power of $\mathcal{F}$-WL beyond $k$-WL. A pattern $P^r$ is a *core* if any homomorphism from $P$ to itself is injective. For example, any clique $K_k$ and cycle of odd length is a core.

**Theorem 3.** *Let $\mathcal{F}$ be a finite set of patterns. If $\mathcal{F}$ contains a pattern $P^r$ which is a core and has treewidth $k$, then there exist graphs that can be distinguished by $\mathcal{F}$-WL but not by $(k-1)$-WL.*

In other words, for such $\mathcal{F}$, $\mathcal{F}$-WL is not bounded by $(k-1)$-WL. For example, since $K_3$ is a core, $\{K_3\}$-WL is not bounded in expressive power by WL = 1-WL. More generally, $\{K_k\}$-WL is not bounded by $(k-1)$-WL. The proof of Theorem 3 is based on extending deep techniques developed in finite model theory, and that have been used to understand the expressive power of *finite variable logics* [Atserias et al., 2007, Bova and Chen, 2019]. This result is stronger than the one underlying the strictness of the $k$-WL hierarchy [Otto, 2017], which states that $k$-WL is strictly more expressive than $(k-1)$-WL. Indeed, Otto [2017] only shows the *existence* of a pattern $P^r$ of treewidth $k$ such that $(k-1)$-WL is not bounded by $\{P^r\}$-WL. In Theorem 3 we provide an *explicit recipe* for finding such a pattern $P^r$, that is, $P^r$ can be taken a core of treewidth $k$.

In summary, we have shown that there is a set $\mathcal{F}$ of patterns such that (i) $\mathcal{F}$-WL can distinguish graphs which cannot be distinguished by $(k-1)$-WL, yet (ii) $\mathcal{F}$-WL cannot distinguish more graphs than $k$-WL. This begs the question whether there is a finite set $\mathcal{F}$ such that $\mathcal{F}$-WL is equivalent in expressive power to $k$-WL. We answer this negatively.

**Proposition 4.** *For any $k > 1$, there does not exist a finite set $\mathcal{F}$ of patterns such that $\mathcal{F}$-WL is equivalent in expressive power to $k$-WL.*

In the proof of Proposition 4 we show a stronger claim. Indeed, we construct two graphs that can be distinguished by 2-WL but cannot be distinguished by any $\mathcal{F}$-WL. Since any two graphs that can be distinguished by 2-WL can also be distinguished by $k$-WL, for any $k > 2$, the proposition follows. In view of the connection between $\mathcal{F}$-MPNNs and GSNs mentioned earlier, we thus show that no GSN can match the power of $k$-WL, which was a question left open in Bouritsas et al. [2020]. We remark that if we allow $\mathcal{F}$ to consist of all (*infinitely many*) patterns of treewidth $k$, then $\mathcal{F}$-WL is equivalent in expressive power to $k$-WL [Dvorak, 2010, Dell et al., 2018].

## 5 When Do Patterns Extend Expressiveness?

Graph patterns are not learned, but must be passed as an input to MPNNs together with the graph structure. Thus, knowing which patterns work well, and which do not, is of *key importance for the power of the resulting $\mathcal{F}$-MPNNs*. This is a difficult question to answer since determining which patterns work well is clearly application-dependent. From a theoretical point of view, however, we can still look into interesting questions related to the problem of which patterns to choose. One such

---

[2]As for the $\mathcal{F}$-WL-test, $\mathcal{F}$-MPNNs cannot distinguish vertices if they are assigned the same feature vector.

a question, and the one studied in this section, is when a pattern adds expressive power over the ones that we have already selected. More formally, we study the following problem: *Given a finite set $\mathcal{F}$ of patterns, when does adding a new pattern $P^r$ to $\mathcal{F}$ extends the expressive power of the $\mathcal{F}$-WL-test?*

To positively answer this question, we need to find two graphs $G$ and $H$, show that they are indistinguishable by the $\mathcal{F}$-WL-test, but show that they can be distinguished by the $\mathcal{F} \cup \{P^r\}$-WL-test. As an example we show that longer cycles always add expressive power.

**Proposition 5.** *For any $\ell > 3$, $\{C_3^r, \ldots, C_\ell^r\}$-WL is more expressive than $\{C_3^r, \ldots, C_{\ell-1}^r\}$-WL.*

Here, $C_\ell$ denotes a cycle of length $\ell$. We also observe that, by Proposition 3, $\{C_3^r, \ldots, C_\ell^r\}$-WL is bounded by 2-WL for any $\ell \geq 3$ because cycles have treewidth two.

In general, it is quite challenging to find two graphs and to prove that they are indistinguishable by $\mathcal{F}$-WL but can be distinguished by the $\mathcal{F} \cup \{P^r\}$-WL-test. Instead, in this section we provide two techniques that can be used to partially answer the question posed above by only looking at properties of the sets of patterns. Our first result is for establishing when a pattern does not add expressive power to a given set $\mathcal{F}$ of patterns, and the second one when it does.

**Detecting when patterns are superfluous.**    Our first result states that instead of choosing complex patterns that are the joins of smaller patterns, one should opt for the smaller patterns.

**Proposition 6.** *Let $P^r = P_1^r \star P_2^r$ be a pattern that is the join of two smaller patterns. Then for any set $\mathcal{F}$ of patterns, we have that $\mathcal{F} \cup \{P^r\}$-WL is upper bounded by $\mathcal{F} \cup \{P_1^r, P_2^r\}$-WL.*

Stated differently, this means that adding to $\mathcal{F}$ any pattern which is the join of two patterns already in $\mathcal{F}$ does not add expressive power. For example, instead of the pattern ⬡ one should simply use the 3-clique ⬟. This is in line with other advantages of smaller patterns: their homomorphism counts are easier to compute, and, since they are less specific, they should tend to produce less over-fitting.

**Detecting when patterns add expressiveness.**    Joining patterns into new patterns does not give extra expressive power, but what about patterns which are not joins? We next provide a useful recipe for detecting when a pattern does add expressive power. We recall that the *core of a graph $P$* is its unique (up to isomorphism) induced subgraph which is both a homomorphic image of $P$ and a core.

**Theorem 4.** *Let $\mathcal{F}$ be a finite set of patterns and let $Q^r$ be a pattern whose core has treewidth $k$. Then, $\mathcal{F} \cup \{Q^r\}$-WL is more expressive than $\mathcal{F}$-WL if every pattern $P^r \in \mathcal{F}$ satisfies one of the following conditions: (i) $P^r$ has treewidth $< k$; or (ii) $P^r$ does not map homomorphically to $Q^r$.*

As an example, $\{K_3, \ldots, K_\ell\}$-WL is more expressive than $\{K_3, \ldots, K_{\ell-1}\}$-WL for any $\ell > 3$ because of the first condition. Similarly, $\{K_3, \ldots, K_\ell, C_m\}$-WL is more expressive than $\{K_3, \ldots, K_\ell\}$-WL for odd cycles $C_m$. Indeed, such cycles are cores and no clique $K_\ell$ with $\ell > 2$ maps homomorphically to $C_m$.

## 6  Experiments

We next showcase that GNN architectures benefit when homomorphism counts of patterns are added as additional vertex features. For patterns where homomorphism and subgraph isomorphism counts differ (e.g., cycles) we compare with GSNs [Bouritsas et al., 2020]. We use the benchmark for GNNs by Dwivedi et al. [2020], as it offers a broad choice of models, datasets and graph classification tasks.

**Selected GNNs.**    We select the best architectures from Dwivedi et al. [2020]: Graph Attention Networks (GAT) [Velickovic et al., 2018], Graph Convolutional Networks (GCN) [Kipf and Welling, 2017], GraphSage [Hamilton et al., 2017], Gaussian Mixture Models (MoNet) [Monti et al., 2017] and GatedGCN [Bresson and Laurent, 2017]. We leave out various linear architectures such as GIN [Xu et al., 2019] as they were shown to perform poorly on the benchmark.

**Learning tasks and datasets.**    As in Dwivedi et al. [2020] we consider (i) graph regression and the ZINC dataset [Irwin et al., 2012, Dwivedi et al., 2020]; (ii) vertex classification and the PATTERN and CLUSTER datasets [Dwivedi et al., 2020]; and (iii) link prediction and the COLLAB dataset [Hu et al., 2020]. We omit graph classification: for this task, the graph datasets from Dwivedi et al. [2020] originate from image data and hence vertex neighborhoods carry little information.

Table 1: Results for the ZINC dataset.

(a) Results for the ZINC dataset show that homomorphism (hom) counts of cycles improve every model. We compare the mean absolute error (MAE) of each model without any homomorphism count (baseline), against the model augmented with the hom count, and with subgraph isomorphism (iso) counts of $C_3$–$C_{10}$.

(b) The effect of different cycles for the GAT model over the ZINC dataset, using mean absolute error.

| MODEL | MAE (BASE) | MAE (HOM) | MAE (ISO) |
|---|---|---|---|
| GAT | 0.47±0.02 | **0.22±0.01** | 0.24±0.01 |
| GCN | 0.35±0.01 | **0.20±0.01** | 0.22±0.01 |
| GraphSage | 0.44±0.01 | **0.24±0.01** | 0.24±0.01 |
| MoNet | 0.25±0.01 | 0.19±0.01 | **0.16±0.01** |
| GatedGCN | 0.34±0.05 | **0.1353±0.01** | 0.1357±0.01 |

| SET ($\mathcal{F}$) | MAE |
|---|---|
| NONE | 0.47±0.02 |
| $\{C_3\}$ | 0.45±0.01 |
| $\{C_4\}$ | 0.34±0.02 |
| $\{C_6\}$ | 0.31±0.01 |
| $\{C_5, C_6\}$ | 0.28±0.01 |
| $\{C_3, \ldots, C_6\}$ | 0.23±0.01 |
| $\{C_3, \ldots, C_{10}\}$ | **0.22±0.01** |

**Patterns.** We extend the initial features of vertices with homomorphism counts of cycles $C_\ell$ of length $\ell \leq 10$, when molecular data (ZINC) is concerned, and with homomorphism counts of $k$-cliques $K_k$ for $k \leq 5$, when social or collaboration data (PATTERN, CLUSTER, COLLAB) is concerned. We use the $z$-score of the logarithms of homomorphism counts to make them standard-normally distributed and comparable to other features. Section 5 tells us that all these patterns will increase expressive power (Theorem 4 and Proposition 5) and are "minimal" in the sense that they are not the join of smaller patterns (Proposition 6). Similar pattern choices were used in Bouritsas et al. [2020]. We use DISC [Zhang et al., 2020][3], a tool specifically built to get homomorphism counts for large graph datasets. Each model is trained and tested independently using combinations of patterns.

**Higher-order GNNs.** We do not compare to higher-order GNNs since this was already done by Dwivedi et al. [2020]. They included ring-GNNs (which outperform 2WL-GNNs) and 3WL-GNNs in their experiments, and these were outperformed by our selected "linear" architectures. Although the increased expressive power of higher-order GNNs may be beneficial for learning, scalability and learning issues (e.g., loss divergence) hamper their applicability [Dwivedi et al., 2020]. Our approach thus certainly outperforms higher-order GNNs with respect to the benchmark.

**Methodology.** Graphs were divided between training/test as instructed by Dwivedi et al. [2020], and all numbers reported correspond to the test set. The reported performance is the average over four runs with different random seeds for the respective combinations of patterns in $\mathcal{F}$, model and dataset. Training times were comparable to the baseline of training models without any augmented features.[4] All models for ZINC, PATTERN and COLLAB were trained on a GeForce GTX 1080 Ti GPU, for CLUSTER a Tesla V100-SXM3-32GB GPU was used.

Next we summarize our results for each learning task separately. Here we report results using 16 message-passing layers for ZINC, PATTERN, and CLUSTER, and 3 message-passing layers for COLLAB, as in Dwivedi et al. [2020]. In the supplementary material we report comparable results using only 4 layers for ZINC and PATTERN.

**Graph regression.** The first task of the benchmark is the prediction of the solubility of molecules in the ZINC dataset [Irwin et al., 2012, Dwivedi et al., 2020], a dataset of about $12\,000$ graphs of small size, each of them consisting of one particular molecule. The results in Table 1a show that each of our models indeed improves by adding homomorphism counts of cycles and the best result is obtained by considering all cycles. GSNs were applied to the ZINC dataset as well [Bouritsas et al., 2020]. In Table 1a we also report results by using subgraph isomorphism counts (as in GSNs): using homomorphism counts provides comparable results with using subgraph isomorphism counts although homomorphism counts are typically more efficient to compute. By looking at the full results, we see that some cycles are much more important than others. Table 1b shows which cycles have the greatest impact for the worst-performing baseline, GAT. Remarkably, adding homomorphism counts makes the GAT model competitive with the best performers of the benchmark.

---

[3]We thank the authors for providing us with an executable.

[4]Code to reproduce our experiments is available at `https://github.com/MrRyschkov/LGP-GNN`

Table 2: Results for the PATTERN dataset show that homomorphism counts improve all models except GatedGCN. We compare weighted accuracy of each model without any homomorphism count (baseline) against the model augmented with the counts of the set $\mathcal{F}$ that showed best performance (best $\mathcal{F}$).

| MODEL + BEST $\mathcal{F}$ | ACCURACY BASELINE | ACCURACY BEST |
|---|---|---|
| GAT $\{K_3, K_4, K_5\}$ | $78.83 \pm 0.60$ | $85.50 \pm 0.23$ |
| GCN $\{K_3, K_4, K_5\}$ | $71.42 \pm 1,38$ | $82.49 \pm 0.48$ |
| GraphSage $\{K_3, K_4, K_5\}$ | $70.78 \pm 0,19$ | $85,85 \pm 0.15$ |
| MoNet $\{K_3, K_4, K_5\}$ | $85.90 \pm 0,03$ | $\mathbf{86.63 \pm 0.03}$ |
| GatedGCN $\{\emptyset\}$ | $86.15 \pm 0.08$ | $86.15 \pm 0.08$ |

Table 3: All models improve the Hits@50 metric over the COLLAB dataset. We compare each model without any homomorphism count (baseline) against the model augmented with the counts of the set of patterns that showed best performance (best $\mathcal{F}$).

| MODEL + BEST $\mathcal{F}$ | HITS@50 BASELINE | HITS@50 BEST |
|---|---|---|
| GAT $\{K_3\}$ | $50.32 \pm 0.55$ | $52.87 \pm 0.87$ |
| GCN $\{K_3, K_4, K_5\}$ | $51.35 \pm 1.30$ | $\mathbf{54.60 \pm 1.01}$ |
| GraphSage $\{K_5\}$ | $50.33 \pm 0.68$ | $51.39 \pm 1.23$ |
| MoNet $\{K_4\}$ | $49.81 \pm 1.56$ | $51.76 \pm 1.38$ |
| GatedGCN $\{K_3\}$ | $51.00 \pm 2.54$ | $51.57 \pm 0.68$ |

**Vertex classification.** The next task in the benchmark corresponds to vertex classification. Here we analyze two datasets, PATTERN and CLUSTER [Dwivedi et al., 2020], both containing over 12 000 artificially generated graphs resembling social networks or communities. The task is to predict whether a vertex belongs to a particular cluster or pattern, and all results are measured using the accuracy of the classifier. Also here, our results show that homomorphism counts, this time of cliques, tend to improve the accuracy of our models. Indeed, for the PATTERN dataset we see an improvement in all models but GatedGCN (Table 2), and three models are improved in the CLUSTER dataset (reported in the supplementary material). Once again, the best performer in this task is a model that uses our extended features. Note that we do not need to compare against subgraph isomorphism counts (GSNs), because for cliques, homomorphism counts coincide with subgraph isomorphism counts (up to a constant factor).

**Link prediction** In our final task we consider a single graph, COLLAB [Hu et al., 2020], with over 235 000 vertices, containing information about the collaborators in an academic network, and the task at hand is to predict future collaboration. The metric used in the benchmark is the Hits@50 evaluator [Hu et al., 2020]. Here, positive collaborations are ranked among randomly sampled negative collaborations, and the metric is the ratio of positive edges that are ranked at place 50 or above. Once again, homomorphism counts of cliques improve the performance of all models, see Table 3. An interesting observation is that the best set of features (cliques) does depend on the model, although the best model uses all cliques again. We do not compare against subgraph isomorphism counts (GSNs) for the same reason as mentioned above.

**Remarks.** The best performers in each task use homomorphism counts, in accordance with our theoretical results, showing that such counts do extend the power of MPNNs. Homomorphism counts are also cheap to compute. For COLLAB, the largest graph in our experiments, the homomorphism counts of all patterns we used, for all vertices, could be computed by DISC [Zhang et al., 2020] in less than 3 minutes. One important remark is that *selecting* the best set of features is still a challenging endeavor. Our theoretical results help us streamline this search, but for now it is still an exploratory task. In our experiments we first looked at adding each pattern individually, and then tried with combinations of those that showed the best improvements. This feature selection strategy incurs considerable cost and needs further investigation.

## 7 Conclusion

We propose $\mathcal{F}$-MPNNs as an efficient way to increase the expressive power of MPNNs and showed that enriching features with homomorphism counts of small patterns is a promising add-on to any GNN architecture. Graph parameter selection and a complete characterization of when adding a new pattern to $\mathcal{F}$ adds expressive power to the $\mathcal{F}$-WL-test deserves further study.

## Acknowledgments and Disclosure of Funding

The computational resources and services used in this work were provided by the HPC core facility CalcUA of the University of Antwerp, VSC (Flemish Supercomputer Center) and Imec IDLAB, funded by the Research Foundation - Flanders (FWO) and the Flemish Government. This work is also partially funded by ANID –Millennium Science Initiative Program – Code ICN17_002, Chile.

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
