# A   Proofs of Section 3

We use the following notions. Let $G$ and $H$ be graphs, $v \in V_G$, $w \in V_H$, and $d \geq 0$. The vertices $v$ and $w$ are said to be indistinguishably by $\mathcal{F}$-WL in round $d$, denoted by $(G, v) \equiv_{\mathcal{F}\text{-WL}}^{(d)} (H, w)$, iff $\chi_{\mathcal{F},G,v}^{(d)} = \chi_{\mathcal{F},H,w}^{(d)}$. Similarly, $G$ and $H$ are said to be indistinguishable by $\mathcal{F}$-WL in round $d$, denoted by $G \equiv_{\mathcal{F}\text{-WL}}^{(d)} H$, iff $\{\!\{ \chi_{\mathcal{F},G,v}^{(d)} \mid v \in V_G \}\!\} = \{\!\{ \chi_{\mathcal{F},H,w}^{(d)} \mid w \in V_H \}\!\}$. Along the same lines, $v$ and $w$ are indistinguishable by an $\mathcal{F}$-MPNN $M$, denoted by $(G, v) \equiv_{M,\mathcal{F}}^{(d)} (H, w)$, iff $\mathbf{x}_{M,\mathcal{F},G,v}^{(d)} = \mathbf{x}_{M,\mathcal{F},H,w}^{(d)}$. Similarly, $G$ and $H$ are said to be indistinguishable by $M$ in round $d$, denoted by $G \equiv_{M,\mathcal{F}}^{(d)} H$, iff $\{\!\{ \mathbf{x}_{M,\mathcal{F},G,v}^{(d)} \mid v \in V_G \}\!\} = \{\!\{ \mathbf{x}_{M,\mathcal{F},H,w}^{(d)} \mid w \in V_H \}\!\}$.

## A.1   Proof of Proposition 1

We show that the class of $\mathcal{F}$-MPNNs is upper bounded in expressive power by $\mathcal{F}$-WL. The proof is analogous to the proof in Morris et al. [2019] showing that MPNNs are bounded by WL.

We show a stronger result by upper bounding $\mathcal{F}$-MPNNs by $\mathcal{F}$-WL-test, layer by layer. More precisely, we show that for every $\mathcal{F}$-MPNN $M$, graphs $G$ and $H$, vertices $v \in V_G$, $w \in V_H$, and $d \geq 0$,

(1) $(G, v) \equiv_{\mathcal{F}\text{-WL}}^{(d)} (H, w) \implies (G, v) \equiv_{M,\mathcal{F}}^{(d)} (H, w)$; and

(2) $G \equiv_{\mathcal{F}\text{-WL}}^{(d)} H \implies G \equiv_{M,\mathcal{F}}^{(d)} H$.

Clearly, these imply that $\mathcal{F}$-MPNNs are bounded in expressive power by $\mathcal{F}$-WL, both when vertex and graph distinguishability are concerned.

**Proof of implication (1).** We show this implication by induction on the number of rounds.

Base case. We first assume $(G, v) \equiv_{\mathcal{F}\text{-WL}}^{(0)} (H, w)$. In other words, $\chi_{\mathcal{F},G,v}^{(0)} = \chi_{\mathcal{F},H,w}^{(0)}$ and thus, $\chi_G(v) = \chi_H(w)$ and for every $P^r \in \mathcal{F}$ we have $\mathrm{hom}(P^r, G^v) = \mathrm{hom}(P^r, H^w)$. By definition, $\mathbf{x}_{M,\mathcal{F},G,v}^{(0)}$ is a hot-one encoding of $\chi_G(v)$ combined with $\mathrm{hom}(P^r, G^v)$ for $P^r \in \mathcal{F}$, for every MPNN $M$, graph $G$ and vertex $v \in V_G$. Since these agree with the labelling and homomorphism counts for vertex $w \in V_H$ in graph $H$, we also have that $\mathbf{x}_{M,\mathcal{F},G,v}^{(0)} = \mathbf{x}_{M,\mathcal{F},H,w}^{(0)}$, as desired.

Inductive step. We next assume $(G, v) \equiv_{\mathcal{F}\text{-WL}}^{(d)} (H, w)$. By the definition of $\mathcal{F}$-WL this is equivalent to $(G, v) \equiv_{\mathcal{F}\text{-WL}}^{(d-1)} (H, w)$ and $\{\!\{ \chi_{\mathcal{F},G,v'}^{(d-1)} \mid v' \in N_G(v) \}\!\} = \{\!\{ \chi_{\mathcal{F},H,w'}^{(d-1)} \mid w' \in N_H(w) \}\!\}$. By the induction hypothesis, this implies $(G, v) \equiv_{M,\mathcal{F}}^{(d-1)} (H, w)$ and there exists a bijection $\beta : N_G(v) \to N_H(w)$ such that $(G, v') \equiv_{M,\mathcal{F}}^{(d-1)} (H, \beta(v'))$ for every $v' \in N_G(v)$, and every $\mathcal{F}$-MPNN $M$. In other words, $\mathbf{x}_{M,\mathcal{F},G,v}^{(d-1)} = \mathbf{x}_{M,\mathcal{F},H,w}^{(d-1)}$ and $\mathbf{x}_{M,\mathcal{F},G,v'}^{(d-1)} = \mathbf{x}_{M,\mathcal{F},H,\beta(v')}^{(d-1)}$ for every $v' \in N_G(v)$. By the definition of $\mathcal{F}$-MPNNs this implies that $\mathrm{COMB}^{(d)}\big( \{\!\{ \mathbf{x}_{M,\mathcal{F},G,v'}^{(d-1)} \mid v' \in N_G(v) \}\!\} \big)$ is equal to $\mathrm{COMB}^{(d)}\big( \{\!\{ \mathbf{x}_{M,\mathcal{F},H,w'}^{(d-1)} \mid w' \in N_H(w) \}\!\} \big)$ and hence also, after applying $\mathrm{UPD}^{(d)}$, $\mathbf{x}_{M,\mathcal{F},G,v}^{(d)} = \mathbf{x}_{M,\mathcal{F},H,w}^{(d)}$. That is, $(G, u) \equiv_{M,\mathcal{F}}^{(d)} (H, w)$, as desired.

**Proof of implication (2).** The implication $G \equiv_{\mathcal{F}\text{-WL}}^{(d)} H \implies G \equiv_{M,\mathcal{F}}^{(d)} H$ now easily follows. Indeed, $G \equiv_{\mathcal{F}\text{-WL}}^{(d)} H$ is equivalent to $\{\!\{ \chi_{\mathcal{F},G,v}^{(d)} \mid v \in V_G \}\!\} = \{\!\{ \chi_{\mathcal{F},H,w}^{(d)} \mid w \in V_H \}\!\}$. In other words, there exists a bijection $\alpha : V_G \to V_H$ such that $\chi_{\mathcal{F},G,v}^{(d)} = \chi_{\mathcal{F},H,\alpha(v)}^{(d)}$ for every $v \in V_G$. We have just shown that this implies $\mathbf{x}_{M,\mathcal{F},G,v}^{(d)} = \mathbf{x}_{M,\mathcal{F},H,\alpha(v)}^{(d)}$ for every $v \in V_G$ and for every $\mathcal{F}$-MPNN $M$. Hence, $\{\!\{ \mathbf{x}_{M,\mathcal{F},G,v}^{(d)} \mid v \in V_G \}\!\} = \{\!\{ \mathbf{x}_{M,\mathcal{F},H,w}^{(d)} \mid w \in V_H \}\!\}$, or $G \equiv_{M,\mathcal{F}}^{(d)} H$, as desired. $\qquad\square$

## A.2 Proof of Theorem 1

We show that for any finite collection $\mathcal{F}$ of patterns, graphs $G$ and $H$, vertices $v \in V_G$ and $w \in V_H$, and $d \geq 0$:

$$(G, v) \equiv_{\mathcal{F}\text{-WL}}^{(d)} (H, w) \iff \mathsf{hom}(T^r, G^v) = \mathsf{hom}(T^r, H^w), \tag{1}$$

for every $\mathcal{F}$-pattern tree $T^r$ of depth at most $d$. Similarly,

$$G \equiv_{\mathcal{F}\text{-WL}}^{(d)} H \iff \mathsf{hom}(T, G) = \mathsf{hom}(T, H), \tag{2}$$

for every (unrooted) $\mathcal{F}$-pattern tree of depth at most $d$.

For a given set $\mathcal{F} = \{P_1^r, \ldots, P_\ell^r\}$ of patterns and $\mathbf{s} = (s_1, \ldots, s_\ell) \in \mathbb{N}^\ell$, we denote by $\mathcal{F}^{\mathbf{s}}$ the graph pattern of the form $(P_1^{s_1} \star \cdots \star P_\ell^{s_\ell})^r$, that is, we join $s_1$ copies of $P_1$, $s_2$ copies of $P_2$ and so on.

**Proof of equivalence (1).** The proof is by induction on the number of rounds $d$.

$\boxed{\implies}$ We first consider the implication $(G, v) \equiv_{\mathcal{F}\text{-WL}}^{(d)} (H, w) \implies \mathsf{hom}(T^r, G^v) = \mathsf{hom}(T^r, H^w)$ for every $\mathcal{F}$-pattern tree $T^r$ of depth at most $d$.

Base case. Let us first consider the base case, that is, $d = 0$. In other words, we consider $\mathcal{F}$-pattern trees $T^r$ consisting of a single root $r$ adorned with a pattern $\mathcal{F}^{\mathbf{s}}$ for some $\mathbf{s} = (s_1, \ldots, s_\ell) \in \mathbb{N}^\ell$. We note that due to the properties of the graph join operator:

$$\mathsf{hom}(T^r, G^v) = \prod_{i=1}^{\ell} \big(\mathsf{hom}(P_i^r, G^v)\big)^{s_i}. \tag{3}$$

Since, $(G, v) \equiv_{\mathcal{F}\text{-WL}}^{(0)} (H, w)$, we know that $\chi_G(v) = \chi_H(w) = a$ for some $a \in \Sigma$ and $\mathsf{hom}(P_i^r, G^v) = \mathsf{hom}(P_i^r, H^w)$ for all $P_i^r \in \mathcal{F}$. This implies that the product in (3) is equal to

$$\prod_{i=1}^{\ell} \big(\mathsf{hom}(P_i^r, H^w)\big)^{s_i} = \mathsf{hom}(T^r, H^w),$$

as desired.

Inductive step. Suppose next that we know that the implication holds for $d - 1$. We assume now $(G, v) \equiv_{\mathcal{F}\text{-WL}}^{(d)} (H, w)$ and consider an $\mathcal{F}$-pattern tree $T^r$ of depth at most $d$. Assume that in the backbone of $T^r$, the root $r$ has $m$ children $c_1, \ldots, c_m$, and denote by $T_1^{c_1}, \ldots, T_m^{c_\ell}$ the $\mathcal{F}$-pattern trees in $T^r$ rooted at $c_i$. Furthermore, we denote by $T_i^{(r, c_i)}$ the $\mathcal{F}$-pattern tree obtained from $T_i^{c_i}$ by attaching $r$ to $c_i$; $T_i^{(r, c_i)}$ has root $r$. Let $\mathcal{F}^{\mathbf{s}}$ be the pattern in $T^r$ associated with $r$. The following equalities are readily verified:

$$\mathsf{hom}(T^r, G^v) = \mathsf{hom}(\mathcal{F}^{\mathbf{s}}, G^v) \prod_{i=1}^{m} \mathsf{hom}(T_i^{(r, c_i)}, G^v) = \mathsf{hom}(\mathcal{F}^{\mathbf{s}}, G^v) \prod_{i=1}^{m} \Big( \sum_{v' \in N_G(v)} \mathsf{hom}(T_i^{c_i}, G^{v'}) \Big). \tag{4}$$

Recall now that we assume $(G, v) \equiv_{\mathcal{F}\text{-WL}}^{(d)} (H, w)$ and thus, in particular, $(G, v) \equiv_{\mathcal{F}\text{-WL}}^{(0)} (H, w)$. Hence, by induction, $\mathsf{hom}(S^r, G^v) = \mathsf{hom}(S^r, H^w)$ for every $\mathcal{F}$-pattern tree $S^r$ of depth 0. In particular, this holds for $S^r = \mathcal{F}^{\mathbf{s}}$ and hence

$$\mathsf{hom}(\mathcal{F}^{\mathbf{s}}, G^v) = \mathsf{hom}(\mathcal{F}^{\mathbf{s}}, H^w).$$

Furthermore, $(G, v) \equiv_{\mathcal{F}\text{-WL}}^{(d)} (H, w)$ implies that there exists a bijection $\beta : N_G(v) \to N_H(w)$ such that $(G, v') \equiv_{\mathcal{F}\text{-WL}}^{(d-1)} (H, \beta(v'))$ for every $v' \in N_G(v)$. By induction, for every $v' \in N_G(v)$ there thus exists a unique $w' \in N_H(w)$ such that $\mathsf{hom}(S^r, G^{v'}) = \mathsf{hom}(S^r, H^{w'})$ for every $\mathcal{F}$-pattern tree $S^r$ of depth at most $d - 1$. In particular, for every $v' \in N_G(v)$ there exists a $w' \in N_H(w)$ such that

$$\mathsf{hom}(T_i^{c_i}, G^{v'}) = \mathsf{hom}(T_i^{c_i}, H^{w'}),$$

for each of the sub-trees $T_i^{c_i}$ in $T^r$. Hence, (4) is equal to

$$\mathsf{hom}(\mathcal{F}^{\mathbf{s}}, H^w) \prod_{i=1}^{m} \Big( \sum_{w' \in N_H(w)} \mathsf{hom}(T_i^{c_i}, H^{w'}) \Big),$$

which in turn is equal to $\mathsf{hom}(T^r, H^w)$, as desired.

$\boxed{\Longleftarrow}$ We next consider the other direction, that is, we show that when $\mathsf{hom}(T^r, G^v) = \mathsf{hom}(S^r, H^w)$ holds for every $\mathcal{F}$-pattern tree $T^r$ of depth at most $d$, then $(G, v) \equiv_{\mathcal{F}\text{-WL}}^{(d)} (H, w)$ holds. This is again verified by induction on $d$. This direction is more complicated and is similar to techniques used in Grohe [2020b]. In our induction hypothesis we further include that a *finite* number of $\mathcal{F}$-pattern trees suffices to infer $(G, v) \equiv_{\mathcal{F}\text{-WL}}^{(d)} (H, w)$ for graphs $G$ and $H$ and vertices $v \in V_G$ and $w \in V_H$.

Base case. Let us consider the base case $d = 0$ first. We need to show that $\chi_G(v) = \chi_H(w)$ and $\mathsf{hom}(P_i^r, G^v) = \mathsf{hom}(P_i^r, H^w)$ for every $P_i^r \in \mathcal{F}$, since this implies $(G, v) \equiv_{\mathcal{F}\text{-WL}}^{(0)} (H, w)$.

We first observe that $\mathsf{hom}(T^r, G^v) = \mathsf{hom}(T^r, H^w)$ for every $\mathcal{F}$-pattern tree $T^r$ of depth 0, implies that $v$ and $w$ must be assigned the same label, say $a$, by $\chi_G$ and $\chi_H$, respectively.

Indeed, if we take $T^r$ to consist of a single root $r$ labeled with $a$ (and thus $r$ is associated with the pattern $\mathcal{F}^{\mathbf{0}}$), then $\mathsf{hom}(T^r, G^v) = \mathsf{hom}(T^r, H^w)$ will be one if $\chi_G(v) = \chi_H(w) = a$ and zero otherwise. This implies that $\chi_G(v) = \chi_H(w) = a$.

Next, we show that $\mathsf{hom}(P_i^r, G^v) = \mathsf{hom}(P_i^r, H^w)$ for every $P_i^r \in \mathcal{F}$. It suffices to consider the $\mathcal{F}$-pattern tree $T_i^r$ consisting of a root $r$ joined with a single copy of $P_i^r$.

We observe that we only need a finite number of $\mathcal{F}$-pattern trees to infer $(G, v) \equiv_{\mathcal{F}\text{-WL}}^{(0)} (H, w)$. Indeed, suppose that $\chi_G$ and $\chi_H$ assign labels $a_1, \ldots, a_L$, then we need $L$ single vertex trees with no patterns attached and root labeled with one of these labels. In addition, we need one $\mathcal{F}$-pattern tree for each pattern $P_i^r \in \mathcal{F}$ and each label $a_1, \ldots, a_L$. That is, we need $L(\ell + 1)$ $\mathcal{F}$-pattern trees of depth 0.

Inductive step. We now assume that the implication holds for $d - 1$ and consider $d$. That is, we assume that if $\mathsf{hom}(T^r, G^v) = \mathsf{hom}(T^r, H^w)$ holds for every $\mathcal{F}$-pattern tree $T^r$ of depth at most $d - 1$, then $(G, v) \equiv_{\mathcal{F}\text{-WL}}^{(d-1)} (H, w)$ holds. Furthermore, we assume that only a finite number $K$ of $\mathcal{F}$-pattern trees $S_1^r, \ldots, S_K^r$ of depth at most $d - 1$ suffice to infer $(G, v) \equiv_{\mathcal{F}\text{-WL}}^{(d-1)} (H, w)$.

So, for $d$, let us assume that $\mathsf{hom}(T^r, G^v) = \mathsf{hom}(T^r, H^w)$ holds for every $\mathcal{F}$-pattern tree of depth at most $d$. We need to show $(G, v) \equiv_{\mathcal{F}\text{-WL}}^{(d)} (H, w)$ and that we can again assume that a finite number of $\mathcal{F}$-pattern trees of depth at most $d$ suffice to infer $(G, v) \equiv_{\mathcal{F}\text{-WL}}^{(d)} (H, w)$.

By definition of $(G, v) \equiv_{\mathcal{F}\text{-WL}}^{(d)} (H, w)$, we can, equivalently, show that $(G, v) \equiv_{\mathcal{F}\text{-WL}}^{(d-1)} (H, w)$ and that there exists a bijection $\beta : N_G(v) \to N_H(w)$ such that $(G, v') \equiv_{\mathcal{F}\text{-WL}}^{(d-1)} (H, \beta(v'))$ for every $v' \in N_G(v)$. That $(G, v) \equiv_{\mathcal{F}\text{-WL}}^{(d-1)} (H, w)$ holds, is by induction, since $\mathsf{hom}(T^r, G^v) = \mathsf{hom}(T^r, H^w)$ for every $\mathcal{F}$-pattern tree of depth at most $d$ and thus also for every $\mathcal{F}$-pattern tree of depth at most $d - 1$. We may thus focus on showing the existence of the bijection $\beta$.

Let $X, Y \in \{G, H\}$, $x \in V_X$ and $y \in V_Y$. We know, by induction and the proof of the previous implication, that $(X, x) \equiv_{\mathcal{F}\text{-WL}}^{(d-1)} (Y, y)$ if and only if $\mathsf{hom}(S_i^r, X^x) = \mathsf{hom}(S_i^r, Y^y)$ for each $i \in K$. Denote by $R_1, \ldots, R_e$ the equivalence class on $V_X \cup V_Y$ induced by $\equiv_{\mathcal{F}\text{-WL}}^{(d-1)}$. Furthermore, define $N_{j,X}(x) := N_X(x) \cap R_j$ and let $n_j = |N_{j,G}(v)|$ and $m_j = |N_{j,H}(w)|$ for $v \in V_G$ and $w \in V_H$, for each $j \in [e]$. If we can show that $n_j = m_j$ for each $j \in [e]$, then this implies the existence of the desired bijection.

Let $T_i^{r=a}$ be the $\mathcal{F}$-pattern tree of depth at most $d$ obtained by attaching $S_i^r$ to a new root vertex $r$ labeled with $a$. We may assume that $v$ and $w$ both have label $a$, since their homomorphism counts for the single root trees with labels from $\Sigma$. The root vertex $r$ is not joined with any $\mathcal{F}^{\mathbf{s}}$ (or alternatively it is joined with $\mathcal{F}^{\mathbf{0}}$). It will be convenient to denote the root of $S_i^r$ by $r_i$ instead of $r$. Then for each

$i \in [K]$:

$$\mathsf{hom}(T_i^{r=a}, G^v) = \sum_{v' \in N_G(v)} \mathsf{hom}(S_i^{r_i}, G^{v'}) = \sum_{j \in [e]} n_j \mathsf{hom}(S_i^{r_i}, G^{v'_j})$$

$$= \sum_{j \in [e]} m_j \mathsf{hom}(S_i^{r_i}, H^{w'_j}) = \sum_{w' \in N_H(w)} \mathsf{hom}(S_i^{r_i}, H^{w'}) = \mathsf{hom}(T_i^{r=a}, H^w),$$

where $v'_j$ and $w'_j$ denote arbitrary vertices in $N_{j,G}(v)$ and $N_{j,H}(w)$, respectively. Let us denote $\mathsf{hom}(S_i^{r_i}, G^{v'_j})$ by $a_{ij}$ and observe that this is equal to $\mathsf{hom}(S_i^{r_i}, H^{w'_j})$. Hence, we know that for each $i \in [K]$:

$$\sum_{j \in [e]} a_{ij} n_j = \sum_{j \in [e]} a_{ij} m_j.$$

Let us call a set $I \subseteq [K]$ compatible if all roots in $S_i^{r_i}$, for $i \in I$, have the same label. Consider a vector $\mathbf{s} = (s_1, \dots, s_K) \in \mathbb{N}^K$ and define its support as $\mathsf{supp}(\mathbf{s}) := \{i \in [K] \mid s_i \neq 0\}$. We say that $\mathbf{s}$ is compatible if its support is. For such a compatible $\mathbf{s}$ we now define $T^{r=a,\mathbf{s}}$ to be the $\mathcal{F}$-pattern tree with root $r$ labeled with $a$, with one child $c$ which is joined with (and inheriting the label from) the following $\mathcal{F}$-pattern tree of depth $d-1$:

$$\bigstar_{i \in \mathsf{supp}(\mathbf{s})} S_i^{s_i}.$$

In other words, we simply join together powers of the $S_i^{r_i}$'s that have roots with the same label. Then for every compatible $\mathbf{s} \in \mathbb{N}^K$:

$$\mathsf{hom}(T^{r=a,\mathbf{s}}, G^v) = \sum_{v' \in N_G(v)} \prod_{i \in [K]} \left( \mathsf{hom}(S_i^{r_i}, G^{v'}) \right)^{s_i} = \sum_{j \in [e]} n_j \prod_{i \in [K]} \left( \mathsf{hom}(S_i^{r_i}, G^{v'_j}) \right)^{s_i}$$

$$= \sum_{j \in [e]} m_j \prod_{i \in [K]} \left( \mathsf{hom}(S_i^{r_i}, H^{w'_j}) \right)^{s_i} = \sum_{w' \in N_H(w)} \prod_{i \in [K]} \left( \mathsf{hom}(S_i^{r_i}, H^{w'}) \right)^{s_i}$$

$$= \mathsf{hom}(T_i^{r=a,\mathbf{s}}, H^w),$$

where, as before, $v'_j$ and $w'_j$ denote arbitrary vertices in $N_{j,G}(v)$ and $N_{j,H}(w)$, respectively. Hence, for any compatible $\mathbf{s} \in \mathbb{N}^K$:

$$\sum_{j \in [e]} n_j \prod_{i \in [K]} a_{ij}^{s_i} = \sum_{j \in [e]} m_j \prod_{i \in [K]} a_{ij}^{s_i}.$$

We now continue in the same way as in the proof of Lemma 4.2 in Grohe [2020b]. We repeat the argument here for completeness. Define $\mathbf{a}_j^{\mathbf{s}} := \prod_{i=1}^K a_{ij}^{s_i}$ for each $j \in [e]$. We assume, for the sake of contradiction, that there exists a $j \in [e]$ such that $n_j \neq m_j$. We choose such a $j_0 \in [e]$ for which $S = \mathsf{supp}(\mathbf{a}_{j_0})$ is inclusion-wise maximal.

We first rule out that $S = \emptyset$. Indeed, suppose that $S = \emptyset$. This implies that $\mathbf{a}_{j_0} = \mathbf{0}$. Now observe that $\mathbf{a}_j$ and $\mathbf{a}_{j'}$ are mutually distinct for all $j, j' \in [e]$, $j \neq j'$. Indeed, if they were equal then this would imply that $R_j = R_{j'}$. Hence, $\mathsf{supp}(\mathbf{a}_j) \neq \emptyset$ for any $j \neq j_0$. We note that $n_j = m_j$ for all $j \neq j_0$ by the maximality of $S$. Hence, $n_{j_0} = n - \sum_{j \neq j_0} n_j = n - \sum_{j \neq j_0} m_j = m_{j_0}$, contradicting our assumption. Hence, $S \neq \emptyset$.

Consider $J := \{j \in [e] \mid \mathsf{supp}(\mathbf{a}_j) = S\}$. For each $j \in J$, consider the truncated vector $\hat{\mathbf{a}}_j := (a_{ij} \mid i \in S)$. We note that $\hat{\mathbf{a}}_j$, for $j \in J$, all have positive entries and are mutually distinct. Lemma 4.1 in Grohe [2020b] implies that we can find a vector (with non-zero entries) $\hat{\mathbf{s}} = (\hat{s}_i \mid i \in S)$ such that the numbers $\hat{\mathbf{a}}_j^{\hat{\mathbf{s}}}$ for $j \in J$ are mutually distinct as well. We next consider $\mathbf{s} = (s_1, \dots, s_K)$ with $s_i = \hat{s}_i$ if $i \in S$ and $s_i = 0$ otherwise. Then by definition of $\hat{\mathbf{s}}$, also $\mathbf{a}_j^{\mathbf{s}}$ for $j \in J$ are mutually distinct.

We next note that for every $p \in \mathbb{N}$, $\mathbf{a}_j^{p\mathbf{s}} = (\mathbf{a}_j^{\mathbf{s}})^p$ and if we define $\mathbf{A}$ to be the $|J| \times |J|$-matrix such that $A_{jj'} := \mathbf{a}_j^{j'\mathbf{s}}$ then this will be an invertible matrix (Vandermonde). We use this invertibility to show that $n_{j_0} = m_{j_0}$.

Let $\mathbf{n}_J := (n_j \mid j \in J)$ and $\mathbf{m}_J = (m_j \mid j \in J)$. If we inspect the $j'$th entry of $\mathbf{n}_J \cdot \mathbf{A}$, then this is equal to

$$\sum_{j \in J} n_j \mathbf{a}_j^{j'\mathbf{s}} = \sum_{j \in [e]} n_j \mathbf{a}_j^{j'\mathbf{s}} - \sum_{\substack{j \in [e] \\ S \not\subseteq \mathsf{supp}(\mathbf{a}_j)}} n_j \mathbf{a}_j^{j'\mathbf{s}} - \sum_{\substack{j \in [e] \\ S \subset \mathsf{supp}(\mathbf{a}_j)}} n_j \mathbf{a}_j^{j'\mathbf{s}}.$$

We want to reduce the above expression to

$$\sum_{j\in J} n_j \mathbf{a}_j^{j'\mathbf{s}} = \sum_{j\in[e]} n_j \mathbf{a}_j^{j'\mathbf{s}} - \sum_{\substack{j\in[e]\\ S\subset \mathsf{supp}(\mathbf{a}_j)}} n_j \mathbf{a}_j^{j'\mathbf{s}}.$$

To see that this holds, we verify that when $S \not\subseteq \mathsf{supp}(\mathbf{a}_j)$ then $\mathbf{a}_j^{j'\mathbf{s}} = 0$. Indeed, take an $\ell \in S$ such that $\ell \notin \mathsf{supp}(\mathbf{a}_j)$. Then, $\mathbf{a}_j^{j'\mathbf{s}}$ contains the factor $a_{\ell j}^{j's_\ell} = 0^{s_\ell}$ with $s_\ell = \hat{s}_\ell \neq 0$. Hence, $\mathbf{a}_j^{j'\mathbf{s}} = 0$.

Now, by the maximality of $S$, for all $j$ with $S \subset \mathsf{supp}(\mathbf{a}_j)$ we have $n_j = m_j$ and thus

$$\sum_{\substack{j\in[e]\\ S\subset \mathsf{supp}(\mathbf{a}_j)}} n_j \mathbf{a}_j^{j'\mathbf{s}} = \sum_{\substack{j\in[e]\\ S\subset \mathsf{supp}(\mathbf{a}_j)}} m_j \mathbf{a}_j^{j'\mathbf{s}}.$$

Since $\sum_{j\in[e]} n_j \mathbf{a}_j^{j'\mathbf{s}} = \sum_{j\in[e]} m_j \mathbf{a}_j^{j'\mathbf{s}}$, we thus also have that

$$\sum_{j\in J} n_j \mathbf{a}_j^{j'\mathbf{s}} = \sum_{j\in J} m_j \mathbf{a}_j^{j'\mathbf{s}}.$$

Since this holds for all $j' \in J$, we have $\mathbf{n}_J \cdot \mathbf{A} = \mathbf{m}_J \cdot \mathbf{A}$ and by the invertibility of $\mathbf{A}$, $\mathbf{n}_J = \mathbf{m}_J$. In particular, since $j_0 \in J$, $n_{j_0} = m_{j_0}$ contradicting our assumption.

As a consequence, $n_j = m_j$ for all $j \in [e]$ and thus we have our desired bijection.

It remains to verify that we only need a finite number of $\mathcal{F}$-pattern trees to conclude that $n_j = m_j$ for all $j \in [e]$. In fact, the above proof indicates that we just need to check test for each root label $a$, we need to check identities for the finite number of pattern trees used to define the matrix $\mathbf{A}$.

**Proof of equivalence 2** This equivalence is shown just like proof of Theorem 4.4. in Grohe [2020a] with the additional techniques from Lemma 4.2 in Grohe [2020b].

$\boxed{\Longrightarrow}$ We first show that $G \equiv_{\mathcal{F}\text{-WL}}^{(d)} H$ implies $\mathsf{hom}(T, G) = \mathsf{hom}(T, H)$ for unrooted $\mathcal{F}$-pattern trees $T$ of depth at most $d$.

Assume that $V_X \cap V_Y = \emptyset$ for $X, Y \in \{G, H\}$. For $x \in V_X$ and $y \in V_Y$, define $x \sim_d y$ if and only if $\mathsf{hom}(T^r, X^x) = \mathsf{hom}(T^r, Y^y)$ for all $\mathcal{F}$-pattern trees $T^r$ of depth at most $d$. Let $R_1, \ldots, R_e$ be the $\sim_d$-equivalence classes and for each $j \in [e]$, let $p_j := |R_j \cap V_G|$ and $q_j := |R_j \cap V_H|$. Suppose that $G \equiv_{\mathcal{F}\text{-WL}}^{(d)} H$. This implies that $p_j = q_j$ for every $j \in [e]$.

Let $T$ be an unrooted $\mathcal{F}$-pattern tree of depth at most $d$, let $r$ be any vertex on the backbone of $T$, and let $T^r$ be the rooted $\mathcal{F}$-pattern tree obtained from $T$ by declaring $r$ as its root. By definition, for $X \in \{G, H\}$, any $x \in V_X \cap R_j$, $\mathsf{hom}(T^r, X^x)$ are all the same number, only dependent on $j \in [e]$. Hence,

$$\mathsf{hom}(T, G) = \sum_{v\in V(G)} \mathsf{hom}(T^r, G^v) = \sum_{j\in[e]} p_j \mathsf{hom}(T^r, G^{v_j})$$

$$= \sum_{j\in[e]} q_j \mathsf{hom}(T^r, H^{w_j}) = \sum_{w\in V(H)} \mathsf{hom}(T^r, H^w) = \mathsf{hom}(T, H),$$

where $v_j$ and $w_j$ are arbitrary vertices in $R_j \cap V_G$ and $R_j \cap V_H$, respectively, and where we used that $\mathsf{hom}(T^r, G^{v_j}) = \mathsf{hom}(T^r, H^{w_j})$ and $p_j = q_j$. Since this holds for any unrooted $\mathcal{F}$-pattern tree $T$ of depth at most $d$, we have show the desired implication.

$\boxed{\Longleftarrow}$ We next check the other direction. That is, we assume that $\mathsf{hom}(T, G) = \mathsf{hom}(T, H)$ holds for any unrooted $\mathcal{F}$-pattern tree $T$ of depth at most $d$ and verify that $G \equiv_{\mathcal{F}\text{-WL}}^{(d)} H$.

For $x \sim_d y$ to hold for $x \in V_X, y \in V_Y$ and $X, Y \in \{G, H\}$, we earlier showed that this corresponds to checking whether $\mathsf{hom}(T_i^{r_i}, X^x) = \mathsf{hom}(T_i^{r_i}, Y^y)$ for a *finite* number $K$ rooted $\mathcal{F}$-pattern trees $T_i^{r_i}$. By definition of the $R_j$'s, $a_{ij} := \mathsf{hom}(T_i^{r_i}, X^x)$ for $x \in R_j$ is well-defined (independent of the choice of $X \in \{G, H\}$ $x \in V_X$). For the rooted $T_i^{r_i}$'s we denote by $T_i$ its unrooted version. Similarly as before,

$$\mathsf{hom}(T_i, G) = \sum_{j\in[e]} a_{ij} p_j = \sum_{j\in[e]} a_{ij} q_j = \mathsf{hom}(T_i, H).$$

We next show that $p_j = q_j$ for $j \in [e]$. In fact, this is shown in precisely the same way as in our previous characterisation and based on Lemma 4.2 in Grohe [2020b]. That is, we again consider trees obtained by joining copies of the $T_i$'s, to obtain, for compatible $\mathbf{s} \in \mathbb{N}^K$,

$$\sum_{j \in [e]} a_{ij}^{s_i} p_j = \sum_{j \in [e]} a_{ij}^{s_i} q_j.$$

It now suffices to repeat the same argument as before (details omitted). □

# B  Proofs of Section 4

## B.1  Additional details of standard concepts

**Core and treewidth.**  A graph $G$ is a *core* if all homomorphisms from $G$ to itself are injective. The *treewidth* of a graph $G = (V, E, \chi)$ is a measure of how much $G$ resembles a tree. This is defined in terms of the *tree decompositions* of $G$, which are pairs $(T, \lambda)$, for a tree $T = (V_T, E_T)$ and $\lambda$ a mapping that associates each vertex $t$ of $V_T$ with a set $\lambda(t) \subseteq V$, satisfying the following:

- The union of $\lambda(t)$, for $t \in V_T$, is equal to $V$;
- The set $\{t \in V_T \mid v \in \lambda(t)\}$ is connected, for all $v \in V$; and
- For each $\{u, v\} \in E$ there is $t \in V_T$ with $\{u, v\} \in \lambda(t)$.

The *width* of $(T, \lambda)$ is $\min_{t \in T}(|\lambda(t)|) - 1$. The treewidth of $G$ is the minimum width of its tree decompositions. For instance, trees have treewidth one, cycles have clique two, and the $k$-clique $K_k$ has treewidth $k - 1$ (for $k > 1$).

If $P^r$ is a pattern, then its treewidth is defined as the treewidth of the graph $P$. Similarly, $P^r$ is a core if $P$ is.

**$k$-WL.**  A *partial isomorphism* from a graph $G$ to a graph $H$ is a set $\pi \subseteq V_G \times V_H$ such that all $(v, w), (v', w') \in \pi$ satisfy the equivalences $v = v' \Leftrightarrow w = w'$, $\{v, v'\} \in E_G \Leftrightarrow \{w, w'\} \in E_H$, $\chi_G(v) = \chi_H(w)$ and $\chi_G(v') = \chi_H(w')$. We may view $\pi$ as a bijective mapping from a subset $X \subseteq V_G$ to a subset of $Y \subseteq V_H$ that is an isomorphism from the induced subgraph $G[X]$ to the induced subgraph $H[Y]$. The *isomorphism type* $\mathsf{isotp}(G, \bar{v})$ of a $k$-tuple $\bar{v} = (v_1, \ldots, v_k)$ is a label in some alphabet $\Sigma$ such that $\mathsf{isotp}(G, \bar{v}) = \mathsf{isotp}(H, \bar{w})$ if and only if $\pi = \{(v_1, w_1), \ldots, (v_k, w_k)\}$ is a partial isomorphism from $G$ to $H$.

Let $k \geq 1$ and $G = (V, E, \chi)$. The $k$-dimensional Weisfeiler-Leman algorithm ($k$-WL) computes a sequence of labellings $\chi_{k,G}^{(d)}$ from $V^k \to \Sigma$. We denote by $\chi_{k,G,\bar{v}}^{(d)}$ the label assigned to the $k$-tuple $\bar{v} \in V^k$ in round $d$. The initial labelling $\chi_{k,G}^{(0)}$ assigns to each $k$-tuple $\bar{v}$ is isomorphism type $\mathsf{isotp}(G, \bar{v})$. Then, for round $d$,

$$\chi_{k,G,\bar{v}}^{(d)} := \big( \chi_{k,G,\bar{v}}^{(d-1)}, M_{\bar{v}}^{(d-1)} \big),$$

where $M_{\bar{v}}^{(d-1)}$ is the multiset

$$\Big\{\!\!\Big\{ \big( \mathsf{isotp}(v_1, \ldots, v_k, w), \chi_{k,G,(v_1,\ldots,v_{k-1},w)}^{(d-1)}, \chi_{k,G,(v_1,\ldots,v_{k-2},w,v_k)}^{(d-1)}, \ldots, \chi_{k,G,(w,v_2,\ldots,v_k)}^{(d-1)} \big) \Big| w \in V \Big\}\!\!\Big\}.$$

As observed in Dell et al. [2018], if $k \geq 2$ holds, then we can omit the entry $\mathsf{isotp}(v_1, \ldots, v_k, w)$ from the tuples in $M_{\bar{v}}$, because all the information it contains is also contained in the entries $\chi_{k,G,\ldots}^{(d-1)}$ of these tuples. Also, WL = 1-WL in the sense that $\chi_{G,v}^{(d)} = \chi_{G,v'}^{(d)}$ if and only if $\chi_{1,G,v}^{(d)} = \chi_{1,G,v'}^{(d)}$ for all $v, v' \in V$. The $k$-WL algorithm is run until the labelings stabilises, i.e., if for all $\bar{v}, \bar{w} \in V^k$, $\chi_{k,G,\bar{v}}^{(d)} = \chi_{k,G,\bar{w}}^{(d)}$ if and only if $\chi_{k,G,\bar{v}}^{(d+1)} = \chi_{k,G,\bar{w}}^{(d+1)}$. We say that $k$-WL *distinguishes two graphs $G$ and $H$* if the multisets of labels for all $k$-tuples of vertices in $G$ and $H$, respectively, coincides. Similar notions as are place for distinguishing $k$-tuples, and for distinguishing graphs (or vertices) based on labels computed by a given number of rounds.

We remark that $k$-WL algorithm given here is sometimes referred to as the "folklore" version of the $k$-dimensional Weisfeiler-Leman algorithm. It is known that indistinguishability of graphs by $k$-WL is equivalent to indistinguishability by sentences in the the $k + 1$-variable fragment of first order logic with counting Cai et al. [1992], and to $\mathsf{hom}(P, G) = \mathsf{hom}(P, H)$ for every graph of treewidth $k$ Dvorak [2010], Dell et al. [2018].

## B.2  Proof of Proposition 3

We show that for each finite set $\mathcal{F}$ of patterns, the expressive power of $\mathcal{F}$-WL is bounded by $k$-WL, where $k$ is the largest treewidth of a pattern in $\mathcal{F}$.

We first recall the following characterisation of $k$-WL-equivalence Dvorak [2010], Dell et al. [2018]. For any two graphs $G$ and $H$,

$$G \equiv_{k\text{-WL}} H \iff \mathsf{hom}(P, G) = \mathsf{hom}(P, H)$$

for every graph $P$ of treewidth at most $k$. On the other hand, we know from Theorem 1 that

$$G \equiv_{\mathcal{F}\text{-WL}} H \iff \mathsf{hom}(T, G) = \mathsf{hom}(T, H)$$

for every $\mathcal{F}$-pattern tree $T$. Hence, we may conclude that

$$G \equiv_{k\text{-WL}} H \implies G \equiv_{\mathcal{F}\text{-WL}} H$$

if we can show that any $\mathcal{F}$-pattern tree has treewidth at most $k$.

Suppose that $k$ is the maximal treewidth of a pattern in $\mathcal{F}$. To conclude the proof, we verify that the treewidth of any $\mathcal{F}$-pattern tree is bounded by $k$.

**Lemma 1.** *If $k$ is the maximal treewidth of a pattern in $\mathcal{F}$, then the treewidth of any $\mathcal{F}$-pattern tree $T$ is bounded by $k$.*

*Proof.* The proof is by induction on the number of patterns joined at any leaf of $T$. Clearly, if no patterns are joined, then $T$ is simply a tree and its treewidth is 1. Otherwise, consider a $\mathcal{F}$-pattern tree $T = (V, E, \chi)$ whose treewidth is at most $k$, and a pattern $P^r$ of treewidth $k$ that is to be joined at vertex $t$ of $T$. By the induction hypothesis, there is a decomposition $(H, \lambda)$ for $T$ witnessing its bounded treewidth, that is,

1. The union of all $\lambda(h)$, for $h \in V_H$, is equal to $V$;
2. The set $\{h \in V_H \mid t \in \lambda(h)\}$ is connected, for all $t \in V$;
3. For each $\{u, v\} \in E$ there is $h \in V_H$ with $\{u, v\} \in \lambda(h)$; and
4. The size of each set $\lambda(h)$ is at most $k + 1$.

Likewise, by assumption, for pattern $P^r$ we have such a tree decomposition, say $(H^P, \lambda^P)$.

Now consider any vertex $h$ of the decomposition of $T$ such that $\lambda(h)$ contains vertex $t$ in $T$ to which $P^r$ is to be joined at its root. We can create a joint tree decomposition for the join of $P^r$ and $T$ (at node $t$) by merging $H$ and $H^P$ with an edge from vertex $h$ in $H$ to the root of $H^P$ (recall $H^P$ is a tree by definition). It is readily verified that this decomposition maintains all necessary properties. Indeed, condition 1 is clearly satisfied since $\lambda$ and $\lambda^p$ combined cover all vertices of the join of $T$ with $P^r$. Furthermore, since the only node shared by $T$ and $P^r$ is the join node, and we merge $H$ and $H^P$ by putting an edge from node $h$ in $H$ to the root of $H^P$, connectivity of is guaranteed and condition 2 is satisfied. Moreover, since the operation of joining $T$ and $P^r$ does not create any extra edges, condition 2 is immediately verified, and so is 3, because we do not create any new vertices, neither in $H$ nor in $H^P$, and we already know that $\lambda$ and $\lambda^P$ are bounded by $k + 1$. $\qquad\square$

## B.3  Proof of Theorem 3

We show that if $\mathcal{F}$ contains a pattern $P^r$ which is a core and has treewidth $k$, then $\mathcal{F}$-WL is not bounded by $(k-1)$-WL. In other words, we construct two graphs $G$ and $H$ that can be distinguished by $\mathcal{F}$-WL but not by $(k-1)$-WL. It suffices to find such graphs that can be distinguished by $\{P^r\}$-WL but not by $(k-1)$-WL. The proof relies on the characterisation of $(k-1)$-WL indistinguishability in terms of the $k$-variable fragment $\mathsf{C}^k$ of first logic with counting and of $k$-pebble bijective games in particular Cai et al. [1992], Hella [1996]. More precisely, $G \equiv_{(k-1)\text{-WL}} H$ if and only if no sentence in $\mathsf{C}^k$ can distinguish $G$ from $H$. In other words, for any sentence $\varphi$ in $\mathsf{C}^k$, $G \models \varphi$ if and only if $H \models \varphi$. We denote indistinguishability by $\mathsf{C}^k$ by $G \equiv_{\mathsf{C}^k} H$. We heavily rely on the constructions used in Atserias et al. [2007] and Bova and Chen [2019]. In fact, we show that the graphs $G$ and $H$ constructed in those works, suffice for our purpose, by extending their strategy for the $k$-pebble game to $k$-pebble bijective games.

**Construction of the graphs $G$ and $H$.** Let $P^r$ be a pattern in $\mathcal{F}$ which is a core and has treewidth $k$. For a vertex $v \in V_P$, we gather all its edges in $E_v := \{\{v, v'\} \mid \{v, v'\} \in E_P\}$. Let $v_1$ be one of the vertices in $V_P$.

For $G$, as vertex set $V_G$ we take vertices of the form $(v, f)$ with $v \in V_P$ and $f : E_v \to \{0, 1\}$. We require that

$$\sum_{e \in E_{v_1}} f(e) \mod 2 = 1 \text{ and } \sum_{e \in E_v} f(e) \mod 2 = 0 \text{ for } v \neq v_1, v \in V_P.$$

For $H$, as vertex set $V_H$ we take vertices of the form $(v, f)$ with $v \in V_P$ and $f : E_v \to \{0, 1\}$. We require that $\sum_{e \in E_v} f(e) \mod 2 = 0$, for all $v \in V_P$. We observe that $G$ and $H$ have the same number of vertices.

The edge sets $E_G$ and $E_H$ of $G$ and $H$, respectively, are defined as follows: $(v, f)$ and $(v', f')$ are adjacent if and only if $v \neq v'$ and furthermore,

$$f(\{v, v'\}) = f'(\{v, v'\}).$$

It is known that $\mathsf{hom}(P, G) = 0$ (here it is used that $P$ is a core), $\mathsf{hom}(P, H) \neq 0$ and $G$ and $H$ are indistinguishably by means of sentences in the $k$-variable fragment $\mathsf{FO}^k$ of first order logic Atserias et al. [2007], Bova and Chen [2019]. To show our theorem, we thus need to verify that $G \equiv_{\mathsf{C}^k} H$ as well. Indeed, for if this holds, then $G \equiv_{(k-1)\text{-WL}} H$ yet $G \not\equiv^{(0)}_{\{P\}\text{-WL}} H$. Indeed, Theorem 1 implies that for $G \not\equiv^{(0)}_{\{P\}\text{-WL}} H$ to hold, $\mathsf{hom}(P, G) = \mathsf{hom}(P, H)$, which we know not to be true. Hence, $G \not\equiv_{\{P\}\text{-WL}} H$, as desired.

**Showing $\mathsf{C}^k$-indistinguishability of $G$ and $H$.** We next show that the graphs $G$ and $H$ are indistinguishable by sentences in $\mathsf{C}^k$. This will be shown by verifying that the Duplicator has a winning strategy for the $k$-pebble bijective game on $G$ and $H$ [Hella, 1996].

The $k$-pebble bijective game. We recall that the $k$-pebble bijective game is played between two players, the Spoiler and the Duplicator, each placing at most $k$ pebbles on the vertices of $G$ and $H$, respectively. The game is played in a number of rounds. The pebbles placed after round $r$ are typically represented by a partial function $p^{(r)} : \{1, \ldots, k\} \to V_G \times V_H$. When $p^{(r)}(i)$ is defined, say, $p^{(r)}(i) = (v, w)$, this means that the Spoiler places the $i$th pebble on vertex $v$ and the Duplicator places the $i$th pebble on $w$. Initially, no pebbles are placed on $G$ and $H$ and hence $p^{(0)}$ is undefined everywhere.

Then in round $r > 0$, the game proceeds as follows:

1. The Spoiler selects a pebble $i$ in $[k]$. All other already placed pebbles are kept on the same vertices. We define $p^{(r)}(j) = p^{(r-1)}(j)$ for all $j \in [k]$, $j \neq i$.
2. The Duplicator responds by choosing a bijection $h : V_G \to V_H$. This bijection should be *consistent* with the pebbles in the restriction of $p^{(r-1)}$ to $[k] \setminus \{i\}$. That is, for every $j \in [k]$, $j \neq i$, if $p^{(r-1)}(j) = (v, w)$ then $w = h(v)$.
3. Next, the Spoiler selects an element $v \in V_G$.
4. The Duplicator defines $p^{(r)}(i) = (v, h(v))$. Hence, after this round, the $i$th pebble is placed on $v$ by the Spoiler and on $h(v)$ by the Duplicator.

Let $\mathsf{dom}(p^{(r)})$ be the elements in $[k]$ for which $p^{(r)}$ is defined. For $i \in \mathsf{dom}(p^{(r)})$ denote by $(v_i, w_i) \in V_G \times V_H$ the pair of vertices on which the $i$th pebble is placed. The Duplicator *wins* round $r$ if the mapping $v_i \mapsto w_i$ is partial isomorphism between $G$ and $H$. More precisely, it should hold that for all edges $\{v_i, v_j\} \in E_G$ if and only if $(w_i, w_j) \in E_H$. In this case, the game continues to the next round. Infinite games are won by the Duplicator. A winning strategy consists of defining a bijection in step 2 in each round, allowing the game to continue, irregardless of which vertex $v$ the Spoiler places a pebble in Step 3.

Winning strategy. We will now provide a winning strategy for the $k$-bijective game on our constructed graphs $G$ and $H$. We recall that $V_G$ and $V_H$ have the same number of vertices, so a bijection between $V_G$ and $V_H$ exists. We show how the Duplicator can select a "good" bijection in Step 2 of the game, by induction on the number of rounds.

To state our induction hypothesis, we first recall some notions and properties from Atserias et al. [2007] and Bova and Chen [2019].

Let $W$ be a walk in $P$ and let $e$ be an edge in $E_P$. Then, $\mathsf{occ}_W(e)$ denotes the number of occurrences of the edge $e$ in the walk. More precisely, if $W = (a_1, \ldots, a_\ell)$ is a walk in $P$ of length $\ell$, then

$$\mathsf{occ}_W(e) := |\{i \in [\ell - 1] \mid e = \{a_i, a_{i+1}\}\}|.$$

Furthermore, for a subset $S \subseteq V_P$, we define

$$\mathsf{avoid}(S) := \bigcup_{\{M \in \mathcal{M}, M \cap S = \emptyset\}} M,$$

where $\mathcal{M}$ is an arbitrary bramble of $P$ of order $> k$. A bramble $\mathcal{M}$ is a set of *connected* subsets of $V_P$ such that for any two elements $M_1$ and $M_2$ in $\mathcal{M}$, either $M_1 \cap M_2 \neq \emptyset$, or there exists a vertex $a \in M_1$ and $b \in M_2$ such that $\{a, b\} \in E_P$. The order of a bramble is the minimum size of a hitting set for $\mathcal{M}$. It is known that $P$ has treewidth $\geq k$ if and only if it has a bramble of order $> k$. In what follows, we let $\mathcal{M}$ be any such bramble.

**Lemma 2** (Lemma 14 in Bova and Chen [2019]). *For any $1 \leq \ell \leq k$, let $(a_1, f_1), \ldots, (a_\ell, f_\ell)$ be vertices in $V_G$. Let $W$ be a walk in $P$ from $v_1$ to $\mathsf{avoid}(\{a_1, \ldots, a_\ell\})$. For all $i \in [\ell]$, let $f_i' : E_{a_i} \to \{0, 1\}$ be defined by*

$$f_i'(e) = f_i(e) + \mathsf{occ}_W(e) \mod 2$$

*for all $e \in E_{a_i}$. Then, the mapping $(a_i, f_i) \mapsto (a_i, f_i')$, for all $i \in [\ell]$, is a partial isomorphism from $G$ to $H$.* □

We use this lemma to show that the bijection (to be defined shortly) selected by the Duplicator induces a partial isomorphism between $G$ and $H$ on the pebbled vertices.

We can now state our induction hypothesis: In each round $r$, there exists a bijection $h : V_G \to V_H$ which is

(a) consistent with the pebbles in the restriction of $p^{(r-1)}$ to $[k] \setminus \{i\}$ (Recall, Pebble $i$ is selected by the Spoiler in Step 1.)

(b) If $p^{(r)}(j) = (a_j, f_j, h(a_j, f_j))$ for $j \in \mathsf{dom}(p^{(r)})$, then there exists a walk $W^{(r)}$ in $P$, from $v$ to $\mathsf{avoid}(\{a_j \mid j \in \mathsf{dom}(p^{(r)})\})$, such that

$$h(a_j, f_j) = (a_j, f_j'),$$

where $f_j'(e) = f_j(e) + \mathsf{occ}_{W^{(r)}}(e) \mod 2$ for every $e \in E_{a_j}$. In other words, on the vertices in $V_G$ pebbled by $p^{(r)}$, the bijection $h$ is, by the previous Lemma, a partial isomorphism from $G$ to $H$.

If this holds, then the strategy for the Duplicator is selecting that bijection $h$ in each round.

Verification of the induction hypothesis. We assume that the special vertex $v_1$ in $P$ has at least two neighbours. Such a vertex exists since otherwise $P$ consists of a single edge while we assume $P$ to be of treewidth at least two.

*Base case.* For the base case ($r = 0$) we define two walks: $W_1 = v_1, v_2$ and $W_2 = v_1, t$ with $v_2 \neq t$ and $v_2, t$ are neighbours of $v_1$. We define $h(a_i, f) = (a_i, f')$ with $f'(e) = f(e) + \mathsf{occ}_{W_1}(e) \mod 2$ if $a_i \neq t$, and $h(t, f) = (t, f')$ with $f'(e) = f(e) + \mathsf{occ}_{W_2}(e) \mod 2$.

The mapping $h$ is a bijection from $V_G$ to $V_H$. We note that it suffices to show that $h$ is injective since $V_G$ and $V_H$ contain the same number of vertices. Since $h(a_i, f_i) \neq h(a_j, f_j)$ whenever $a_i \neq a_j$, we can focus on comparing $h(a_i, f)$ and $h(a_i, g)$ with $f \neq g$. This implies that $f(e) \neq g(e)$ for at least one edge $e \in N_{a_i}$. Clearly, this implies that $f'(e) = f(e) + \mathsf{occ}_W(e) \mod 2 \neq g'(e) = g(e) + \mathsf{occ}_W(e) \mod 2$. In fact this, holds for any walk $W$ and thus in particular for $W_1$ and $W_2$. We further observe that $h$ is consistent simply because no pebbles have been placed yet. For the same reason we can take the walk $W^{(0)}$ to be either $W_1$ or $W_2$.

*Inductive case.* Assume that the induction hypothesis holds for round $r$ and consider round $r + 1$. Let $p^{(r)} = (a_j, f_j, a_j, f_j')$ for $j \in \mathsf{dom}(p^{(r)})$. By induction, there exists a bijection $h' : V_G \to V_H$ such

that $h(a_j, f_j) = (a_j, f_j')$ and furthermore, $f_j'(e) = f_j(e) + \mathsf{occ}_{W^{(r)}}(e) \mod 2$ for every $e \in N_{a_j}$, for some walk $W^{(r)}$ from $v_1$ to $t \in \mathsf{avoid}(\{a_j \mid j \in \mathsf{dom}(p^{(r)})\})$.

Assume that the Spoiler selects $i \in [k]$ in Step 1 in round $r + 1$. We define the Duplicator's bijection $h : V_G \to V_H$ for round $r + 1$, as follows. Recall that $t \in V_P$ is the vertex in which the walk $W^{(r)}$ ends.

- For all $(a, f) \in V_G$ such that $a \neq t$, we define $h(a, f) = (a, f')$ where for each $e \in E_a$:

$$f'(e) = f(e) + \mathsf{occ}_{W^{(r)}}(e) \mod 2.$$

- For all $(t, f) \in V_G$, we will extend $W^{(r)}$ with a walk $W'$ so that it ends in a vertex $t'$ different from $t$. Suppose that $M \in \mathcal{M}$ such that $t \in M$. We want to find an $M' \in \mathcal{M}$ such that $M' \cap (\{a_j \mid j \in \mathsf{dom}(p^{(r)}), j \neq i\} \cup \{t\}) = \emptyset$. We can then take $t'$ to be a vertex in $M'$ and since $M$ and $M'$ are both connected, and either have a vertex in common or an edge between them, we can let $W'$ be a walk from $t$ to $t'$ entirely in $M$ and $M'$. Now, such an $M'$ exists since otherwise $\{a_j \mid j \in \mathsf{dom}(p^{(r)}), j \neq i\} \cup \{t\}$ would be a hitting set for $\mathcal{M}$ of size at most $k$. We know, however, that any hitting set $\mathcal{M}$ must be of size $k + 1$ because of the treewidth $k$ assumption for $P$. We now define the bijection as $h(t, f) = (t, f')$ where for each $e \in E_t$:

$$f'(e) = f(e) + \mathsf{occ}_{W^{(r)}, W'}(e) \mod 2.$$

This concludes the definition of $h : V_G \to V_H$. We need to verify a couple of things: (i) $h$ is bijection; (ii) $h$ is consistent with all pebbles in $p^{(r)}$ except for the "unpebbled" one $p^{(r)}(i)$; and (iii) it induces a partial isomorphism on pebbled vertices.

(i) $h$ is a bijection. Since $V_G$ and $V_H$ are of the same size, it suffices to show that $h$ is an injection. Clearly, $h(a_1, f_1) \neq h(a_2, f_2)$ whenever $a_1 \neq a_2$. We can thus focus on $h(a, f_1)$ and $h(a, f_2)$ with $f_1 \neq f_2$. Then, $f_1$ and $f_2$ differ in at least one edge $e \in E_a$ and for this edge:

$$f_1'(e) = f_1(e) + \mathsf{occ}_W(e) \mod 2 \neq f_2(e) + \mathsf{occ}_W(e) \mod 2 = f_2'(e).$$

for any walk $W$. In particular, this holds for both walks used in the definition of $h$: $W^{(r)}$, used when $a \neq t$, and $W^{(r)}, W'$ used when $a = t$. Hence, $h$ is indeed a bijection.

(ii) $h$ is consistent. For each $j \in \mathsf{dom}(p^{(r+1)})$ with $j \neq i$, let $p^{(r+1)} = (a_j, f_j, a_j, f_j')$. Now, by induction, $W^{(r)}$ ended in a vertex $t$ distinct from any of these $a_j$'s and thus none of these $a_j$'s are equal to $t$. This implies that $h(a_j, f_j) = (a_j, f_j'')$ with $f_j''(e) = f_j(e) + \mathsf{occ}_{W^{(r)}}(e)$ mod 2. But this is precisely how $p^{(r)}$ placed its pebbles, by induction. Hence, $f_j''(e) = f_j'(e)$ and thus $h$ is consistent.

(iii) $p^{(r+1)}$ induces a partial isomorphism. After the Spoiler picked an element $(a_i, f_i) \in V_G$, we now know that $p^{(r+1)}(j) = (a_j, f_j, h(a_j, f_j))$ for all $j \in \mathsf{dom}(p^{(r+1)})$. We recall that $h$ is defined in two possible ways, using two distinct walks: $W^{(r)}$, for vertices in $V_G$ not involving $t$, or, otherwise using the walk $W^{(r)}, W'$, for vertices in $V_G$ involving $t$.

Hence, when all $a_j$'s for $p^{(r+1)}$ are distinct from $t$, then $h(a_j, f_j) = (a_j, f_j')$ with $f_j'(e) = f_j(e) + \mathsf{occ}_{W^{(r)}}(e) \mod 2$ and we can simply take the new walk $W^{(r+1)}$ to be $W^{(r)}$. Then, Lemma 2 implies that the mapping $(a_j, f_j) \to h(a_j, f_j)$, for $j \in \mathsf{dom}(p^{(r+1)})$ is a partial isomorphism from $G$ to $H$, as desired.

Otherwise, we know that $a_j \neq t$ for $j \neq i$ but $a_i = t$. That is, the Spoiler places the $i$th pebble on a vertex of the form $(t, f)$ in $V_G$. We now have that $h$ is defined in two ways for the pebbled elements using the two distinct walks. We next show that $W^{(r)}, W'$ can be used for both types of pebbled elements in $p^{(r+1)}$, those of the form $(a_j, f)$ with $a_j \neq t$ and $(t, f)$. For the last type this is obvious since we defined $h(t, f)$ in terms of $W^{(r)}, W'$. For the former type, we note that $a_j \notin M$ and $a_j \notin M'$ for $j \neq i$. If we take an edge $e \in N_{a_j}$, then $\mathsf{occ}_{W^r, W'}(e) = \mathsf{occ}_{W^{(r)}}(e)$ because $W'$ lies entirely in $M$ and $M'$. As a consequence, for $(a_j, f_j)$ with $j \neq i$, for all $e \in N_j$:

$$f_j'(e) = f_j(e) + \mathsf{occ}_{W^{(r)}}(e) \mod 2$$

$$= f_j(e) + \mathsf{occ}_{W^{(r)}, W'}(e) \mod 2.$$

Then, Lemma 2 implies that the mapping $(a_j, f_j) \to h(a_j, f_j)$, for $j \in \mathsf{dom}(p^{(r+1)})$ is a partial isomorphism from $G$ to $H$, because we can use the same walk $W^{(r), W'}$ for all pebbled vertices.

## B.4  Proof of Proposition 4

We show that no finite set $\mathcal{F}$ of patterns suffices for $\mathcal{F}$-WL to be equivalent to $k$-WL, for $k > 1$, in terms of expressive power. The proof is by contradiction. That is, suppose that there exists a set $\mathcal{F}$ such that $G \equiv_{\mathcal{F}\text{-WL}} H \Leftrightarrow G \equiv_{k\text{-WL}} H$ for any two graphs $G$ and $H$. In particular, $G \equiv_{\mathcal{F}\text{-WL}} H \Rightarrow G \equiv_{k\text{-WL}} H$ and thus also $G \equiv_{\mathcal{F}\text{-WL}} H \Rightarrow G \equiv_{2\text{-WL}} H$, since the 2-WL-test is upper bounded by any $k$-WL-test for $k > 2$. We argue that no finite set $\mathcal{F}$ exists satisfying $G \equiv_{\mathcal{F}\text{-WL}} H \Rightarrow G \equiv_{2\text{-WL}} H$.

Let $m$ denote the maximum number of vertices of any pattern in $\mathcal{F}$.[5] Furthermore, consider graphs $G$ and $H$, where $G$ is the disjoint union of $m + 2$ copies of the cycle $C_{m+1}$, and $H$ is the union of $m + 1$ copies of the cycle $C_{m+2}$. Note that $G$ and $H$ have the same number of vertices.

We observe that any homomorphism from a pattern $P^r$ in $\mathcal{F}$ to $G^v$ or $H^w$, for vertices $v \in V_G$ and $w \in V_H$, maps $P^r$ to either a copy of $C_{m+1}$ (for $G$) or a copy of $C_{m+2}$ (for $H$). Furthermore, any such homomorphism maps $P^r$ in a subgraph of $C_{m+1}$ or $C_{m+2}$, consisting of at most $m$ vertices. There is, however, a unique (up to isomorphism) subgraph of $m$ vertices in $C_{m+1}$ and $C_{m+2}$. Indeed, such subgraphs will be a path of length $m$. This implies that $\mathsf{hom}(P^r, G^v) = \mathsf{hom}(P^r, H^w)$ for any $v \in V_G$ and $w \in V_H$. Since the argument holds for any pattern $P^r$ in $\mathcal{F}$, all vertices in $G$ and $H$ will have the same homomorphism count for patterns in $\mathcal{F}$. Furthermore, since both $G$ and $H$ are regular graphs (each vertex has degree two), this implies that $\mathcal{F}$-WL cannot distinguish between $G$ and $H$. This is formalised in the following lemma. We recall that a $t$-regular graph is a graph in which every vertex has degree $t$.

**Lemma 3.** *For any set $\mathcal{F}$ of patterns and any two $t$-regular (unlabelled) graphs $G$ and $H$ such that $\mathsf{hom}(P^r, X^x) = \mathsf{hom}(P^r, Y^y)$ for $P^r \in \mathcal{F}$, $X, Y \in \{G, H\}$, $x \in V_X$ and $y \in V_Y$ holds, $G \equiv_{\mathcal{F}\text{-WL}} H$.*

*Proof.* The lemma is readily verified by induction on the number $d$ of rounds of $\mathcal{F}$-WL. We show a stronger result in that $\chi^{(d)}_{\mathcal{F}, X, x} = \chi^{(d)}_{\mathcal{F}, Y, y}$ for any $d$, $X, Y \in \{G, H\}$, $x \in V_X$ and $y \in V_Y$, from which $G \equiv_{\mathcal{F}\text{-WL}} H$ follows. By our Theorem 1, it suffices to show that $\mathsf{hom}(T^r, X^x) = \mathsf{hom}(T^r, Y^y)$ for $\mathcal{F}$-pattern trees of depth at most $d$. Let $\mathcal{F} = \{P_1^r, \ldots, P_\ell^r\}$. For the base case, let $T^r$ be a join pattern $\mathcal{F}^{\mathbf{s}}$ for some $\mathbf{s} = (s_1, \ldots, s_\ell) \in \mathbb{N}^\ell$. Then,

$$\mathsf{hom}(T^r, X^x) = \prod_{i=1}^{\ell} (\mathsf{hom}(P_i^r, X^x))^{s_i} = \prod_{i=1}^{\ell} (\mathsf{hom}(P_i^r, Y^y))^{s_i} = \mathsf{hom}(T^r, Y^y),$$

since $\mathsf{hom}(P_i^r, X^x) = \mathsf{hom}(P_i^r, Y^y)$ for any $P_i^r \in \mathcal{F}$. Then, for the inductive case, assume that $\mathsf{hom}(S^r, X^x) = \mathsf{hom}(S^r, Y^y)$ for any $\mathcal{F}$-pattern tree $S^r$ of depth at most $d - 1$, $X, Y \in \{G, H\}$, $x \in V_X$ and $y \in V_Y$, and consider an $\mathcal{F}$-pattern $T^r$ of depth $d$. Let $S_1^{c_1}, \ldots, S_p^{c_p}$ be the $\mathcal{F}$-pattern trees of depth at most $d - 1$ rooted at the children $c_1, \ldots, c_p$ of $r$ in the backbone of $T^r$. As before, let $\mathcal{F}^{\mathbf{s}}$ the pattern joined at $r$ in $T^r$. Then,

$$\mathsf{hom}(T^r, X^x) = \mathsf{hom}(\mathcal{F}^{\mathbf{s}}, X^x) \prod_{i=1}^{p} \sum_{x' \in N_X(x)} \mathsf{hom}(S_i^{c_i}, X^{x'}) = \mathsf{hom}(\mathcal{F}^{\mathbf{s}}, X^x) \prod_{i=1}^{p} t \cdot \mathsf{hom}(S_i^{c_i}, X^{\tilde{x}})$$

$$= \mathsf{hom}(\mathcal{F}^{\mathbf{s}}, Y^y) \prod_{i=1}^{p} t \cdot \mathsf{hom}(S_i^{c_i}, Y^{\tilde{y}}) = \mathsf{hom}(\mathcal{F}^{\mathbf{s}}, Y^y) \prod_{i=1}^{p} \sum_{y' \in N_Y(y)} \mathsf{hom}(S_i^{c_i}, Y^{y'})$$

$$= \mathsf{hom}(T^r, Y^y),$$

---

[5]Strictly speaking, we can use the diameter of any pattern in $\mathcal{F}$ instead, but it is easier to convey the proof simply by taking number of vertices.

where we used that $N_X(x)$ and $N_Y(y)$ both consists of $t$ vertices (regularity), by the induction hypothesis all vertex have the same homomorphism counts for $\mathcal{F}$-patterns trees of depth at most $d-1$, and where $\tilde{x}$ and $\tilde{y}$ are taken to be arbitrary vertices in $N_X(x)$ and $N_Y(y)$, respectively. $\quad\square$

Hence, since $G$ and $H$ are 2-regular and satisfy the conditions of the lemma, we may indeed infer that $G \equiv_{\mathcal{F}\text{-WL}} H$. We note, however, that $G \not\equiv_{2\text{-WL}} H$. Indeed, from Dvorak [2010] and Dell et al. [2018] we know that $G \equiv_{2\text{-WL}} H$ implies that $\mathsf{hom}(P, G) = \mathsf{hom}(P, H)$ for any graph $P$ of treewidth at most two. In particular, $G \equiv_{2\text{-WL}} H$ implies that $\mathsf{hom}(C_\ell, G) = \mathsf{hom}(C_\ell, H)$ for all cycles $C_\ell$. We now conclude by observing that $\mathsf{hom}(C_{m+1}, G) \neq \mathsf{hom}(C_{m+1}, H)$ by construction. Recall that $G$ consists of $m + 2$ disjoint copies of $C_{m+1}$ and H consists of $m + 1$ copies of $C_{m+2}$. Consider $\mathsf{hom}(C_{m+1}, G)$ and $\mathsf{hom}(C_{m+1}, H)$. We can write these as $(m+2)(m+1)\mathsf{hom}(C^r_{m+1}, C^v_{m+1})$ and $(m+1)(m+2)\mathsf{hom}(C^r_{m+1}, C^w_{m+2})$ for some fixed vertices $v$ and $w$ and where all cycles are now rooted. It now suffices to observe that all homomorphisms $h$ from $C^r_{m+1}$ to $C^w_{m+2}$ are such that the image $h(C^r_{m+1})$ contain $< m + 1$ vertices. And moreover, with any such $h$, we can associate a unique $h'$ from $C^r_{m+1}$ to $C^v_{m+1}$ (such that also $h'(C^r_{m+1})$ contains $< m + 1$ vertices). Finally, we note that there are two homomorphisms from $C^r_{m+1}$ to $C^v_{m+1}$ which are surjections and thus cover all $m + 1$ vertices in $C^v_{m+1}$. We may thus conclude that $\mathsf{hom}(C^r_{m+1}, C^w_{m+2}) < \mathsf{hom}(C^r_{m+1}, C^v_{m+1})$, from which we can infer that $\mathsf{hom}(C_{m+1}, G) \neq \mathsf{hom}(C_{m+1}, H)$, as desired. We have thus found two graphs with cannot be distinguished by $\mathcal{F}$-WL, but that can be distinguished by 2-WL, contradicting our assumption that $G \equiv_{\mathcal{F}\text{-WL}} H \Rightarrow G \equiv_{2\text{-WL}} H$.

## C   Proofs of Section 5

### C.1   Proof of Proposition 5

We show that for any $k > 3$, $\{C^r_3, \ldots, C^r_k\}$-WL is more expressive than $\{C^r_3, \ldots, C^r_{k-1}\}$-WL. More precisely, we construct two graphs $G$ and $H$ such that $G$ and $H$ cannot be distinguished by $\{C^r_3, \ldots, C^r_{k-1}\}$-WL, but they can be distinguished by $\{C^r_3, \ldots, C^r_k\}$-WL.

The proof is analogous to the proof of Proposition 4. Indeed, it suffices to let $G$ consist of $k$ disjoint copies of $C_{k+1}$ and $H$ to consist of $k + 1$ disjoint copies of $C_k$. Then, as observed in the proof of Proposition 4, $G$ and $H$ will be indistinguishable by $\{C^r_3, \ldots, C^r_{k-1}\}$-WL simply because each pattern has at most $k - 1$ vertices. Yet, by construction, $\mathsf{hom}(C_k, G) \neq \mathsf{hom}(C_k, H)$ and thus $G$ and $H$ are distinguishable (already by the initial labelling) by $\{C^r_3, \ldots, C^r_k\}$-WL.

### C.2   Proof of Proposition 6

Let $P^r = P^r_1 \star P^r_2$ be a pattern that is the join of two smaller patterns. We show that for any any set $\mathcal{F}$ of patterns, we have that $\mathcal{F} \cup \{P^r\}$-WL is upper bounded by $\mathcal{F} \cup \{P^r_1, P^r_2\}$-WL. That is, for every two graphs $G$ and $H$, $G \equiv_{\mathcal{F} \cup \{P^r_1, P^r_2\}\text{-WL}} H$ implies $G \equiv_{\mathcal{F} \cup \{P^r\}\text{-WL}} H$. By definition, $G \equiv_{\mathcal{F} \cup \{P^r_1, P^r_2\}\text{-WL}} H$ is equivalent to $\{\!\{\chi^{(d)}_{\mathcal{F} \cup \{P^r_1, P^r_2\}, G, v} \mid v \in V_G\}\!\} = \{\!\{\chi^{(d)}_{\mathcal{F} \cup \{P^r_1, P^r_2\}, H, w} \mid w \in V_H\}\!\}$. In other words, with every $v \in V_G$ we can associate a unique $w \in V_H$ such that $\chi^{(d)}_{\mathcal{F} \cup \{P^r_1, P^r_2\}, G, v} = \chi^{(d)}_{\mathcal{F} \cup \{P^r_1, P^r_2\}, H, w}$. We show, by induction on $d$, that this implies that $\chi^{(d)}_{\mathcal{F} \cup \{P^r\}, G, v} = \chi^{(d)}_{\mathcal{F} \cup \{P^r\}, H, w}$. This suffices to conclude that $\{\!\{\chi^{(d)}_{\mathcal{F} \cup \{P^r\}, G, v} \mid v \in V_G\}\!\} = \{\!\{\chi^{(d)}_{\mathcal{F} \cup \{P^r\}, H, w} \mid w \in V_H\}\!\}$ and thus $G \equiv_{\mathcal{F} \cup \{P^t\}\text{-WL}} H$.

Base case. We show that $\{\!\{\chi^{(d)}_{\mathcal{F} \cup \{P^r_1, P^r_2\}, G, v} \mid v \in V_G\}\!\} = \{\!\{\chi^{(d)}_{\mathcal{F} \cup \{P^r_1, P^r_2\}, H, w} \mid w \in V_H\}\!\}$ implies that with every $v \in V_G$ we can associate a unique $w \in V_H$ satisfying $\chi^{(0)}_{\mathcal{F} \cup \{P^r\}, G, v} = \chi^{(0)}_{\mathcal{F} \cup \{P^r\}, H, w}$. Indeed, as already observed, $\{\!\{\chi^{(d)}_{\mathcal{F} \cup \{P^r_1, P^r_2\}, G, v} \mid v \in V_G\}\!\} = \{\!\{\chi^{(d)}_{\mathcal{F} \cup \{P^r_1, P^r_2\}, H, w} \mid w \in V_H\}\!\}$ implies that with every $v \in V_G$ we can associate a unique $w \in V_H$ such that $\chi^{(d)}_{\mathcal{F} \cup \{P^r_1, P^r_2\}, G, v} = \chi^{(d)}_{\mathcal{F} \cup \{P^r_1, P^r_2\}, H, w}$. This in turn implies that $\chi^{(0)}_{\mathcal{F} \cup \{P^r_1, P^r_2\}, G, v} = \chi^{(0)}_{\mathcal{F} \cup \{P^r_1, P^r_2\}, H, w}$, which implies that $\mathsf{hom}(P^r_1, G^v) = \mathsf{hom}(P^r_1, H^w)$ and $\mathsf{hom}(P^r_2, G^v) = \mathsf{hom}(P^r_2, H^w)$ and $\mathsf{hom}(Q^r, G^v) = \mathsf{hom}(Q^r, H^w)$ for every $Q^r \in \mathcal{F}$. As a consequence, from properties of the graph join operators, since $P^r = P^r_1 \star P^r_2$:

$$\mathsf{hom}(P^r, G^v) = \mathsf{hom}(P^r_1, G^v) \cdot \mathsf{hom}(P^r_2, G^v) = \mathsf{hom}(P^r_1, H^w) \cdot \mathsf{hom}(P^r_2, H^w) = \mathsf{hom}(P^r, H^w),$$

and thus also $\chi^{(0)}_{\mathcal{F}\cup\{P^r\},G,v} = \chi^{(0)}_{\mathcal{F}\cup\{P^r\},H,w}$.

Inductive case. We assume that $\{\!\{\chi^{(d)}_{\mathcal{F}\cup\{P_1^r,P_2^r\},G,v} \mid v \in V_G\}\!\} = \{\!\{\chi^{(d)}_{\mathcal{F}\cup\{P_1^r,P_2^r\},H,w} \mid w \in V_H\}\!\}$ implies $\chi^{(e)}_{\mathcal{F}\cup\{P^r\},G,v} = \chi^{(e)}_{\mathcal{F}\cup\{P^r\},H,w}$, and want to show that it also implies $\chi^{(e+1)}_{\mathcal{F}\cup\{P^r\},G,v} = \chi^{(e+1)}_{\mathcal{F}\cup\{P^r\},H,w}$. We again use the fact that we can associate with every $v \in V_G$ a unique vertex $w \in V_H$ such that $\chi^{(d)}_{\mathcal{F}\cup\{P_1^r,P_2^r\},G,v} = \chi^{(d)}_{\mathcal{F}\cup\{P_1^r,P_2^r\},H,w}$. In particular, this implies that $\chi^{(e)}_{\mathcal{F}\cup\{P_1^r,P_2^r\},G,v} = \chi^{(e)}_{\mathcal{F}\cup\{P_1^r,P_2^r\},H,w}$ and $\chi^{(e+1)}_{\mathcal{F}\cup\{P_1^r,P_2^r\},G,v} = \chi^{(e+1)}_{\mathcal{F}\cup\{P_1^r,P_2^r\},H,w}$. From the definition of the -WL-test, it must also be the case that the multisets $\{\chi^{(e)}_{\mathcal{F}\cup\{P_1^r,P_2^r\},G,v'} \mid v' \in N_G(v)\}$ and $\{\chi^{(e)}_{\mathcal{F}\cup\{P_1^r,P_2^r\},H,w'} \mid v' \in N_H(w)\}$ must be equal as well, i.e., we can find a one-to-one corresponence between neighbors of $v$ in $G$ and neighbors of $w$ in $H$ that have the same label. From the induction hypothesis we then have that $\chi^{(e)}_{\mathcal{F}\cup\{P^r\},G,v} = \chi^{(e)}_{\mathcal{F}\cup\{P^r\},H,w}$ and also that the multisets $\{\chi^{(e)}_{\mathcal{F}\cup\{P^r\},G,v'} \mid v' \in N_G(v)\}$ and $\{\chi^{(e)}_{\mathcal{F}\cup\{P^r\},H,w'} \mid v' \in N_H(w)\}$ are equal, which implies, by the definition of the WL-test, that $\chi^{(e+1)}_{\mathcal{F}\cup\{P^r\},G,v} = \chi^{(e+1)}_{\mathcal{F}\cup\{P^r\},H,w}$, as was to be shown.

### C.3   Proof of Theorem 4

We show that $\mathcal{F} \cup \{Q^r\}$-WL, where $Q^r$ is pattern whose core has treewidth $k$, is more expressive than $\mathcal{F}$-WL if every pattern $P^r \in \mathcal{F}$ satisfies one of the following conditions: (i) $P^r$ has treewidth $< k$; or (ii) $P^r$ does not map homomorphically to $Q^r$.

Let $c(Q)^r$ to denote the (rooted) core of $Q$, in which the root of $c(Q)^r$ is any vertex which is the image of the root of $Q^r$ in a homomorphism from $Q^r$ to $c(Q)^r$. By assumption, $c(Q)^r$ has treewidth $k$.

Clearly, $\mathcal{F}$-WL is upper bounded by $\mathcal{F} \cup \{Q^r\}$-WL. Thus, all we need for the proof is to find two graphs that are indistinguishable by $\mathcal{F}$-WL but are in fact distinguished by $\mathcal{F} \cup \{Q^r\}$-WL.

Those two graphs are, in fact, the graphs $G$ and $H$ constructed for $c(Q)^r$ (of treewidth $k$) in the proof of Theorem 3. From that proof, we know that:

(a) $\mathsf{hom}(c(Q), G) = 0$ and $\mathsf{hom}(c(Q), H) \neq 0$; and
(b) $G \equiv_{\mathsf{C}^k} H$.

We note that (a) immediately implies that $G$ and $H$ can be distinguished by $\mathcal{F} \cup \{Q^r\}$-WL. In fact, they are distinguished in already by the initial labelling in round 0. We next show that $G$ and $H$ are indistinguishable by $\mathcal{F}$-WL.

Let us first present a small structural result that helps us deal with patterns in $\mathcal{F}$ satisfying the second condition of the Theorem.

**Lemma 4.** *If a rooted pattern $P^r$ does not map homomorphically to $Q^r$, then* $\mathsf{hom}(P, G) = \mathsf{hom}(P, H) = 0$

*Proof.* We use the following property of graphs $G$ and $H$, which can be directly observed from their construction (and was already noted in Atserias et al. [2007] and Bova and Chen [2019]). Define $G^r$ and $H^r$ by setting as their root any vertex $(a_r, f)$, for $a_r$ the root of $c(Q)^r$. Then there is a homomorphism from $G^r$ to $c(Q)^r$, and there is a homomorphism from $H^r$ to $c(Q)^r$.

Now, any homomorphism $h$ from $P^r$ to $G$ can be extended to a homomorphism from $P^r$ to $Q^r$: we compose $h$ with the homomorphism mentioned above from $G$ to $c(Q)^r$, which by definition again maps homomorphically to $Q^r$. Since by definition we have that $P^r$ does not map to $Q^r$, $h$ cannot exist. The proof for $H$ is analogous. □

Now, let $\mathcal{F}'$ be the set of patterns obtained by removing from $\mathcal{F}$ all patterns which do not map homomorphically to $Q^r$. By Lemma 4, we have that $G$ and $H$ are distinguished by the $\mathcal{F}$-WL-test if and only if they are distinguished by $\mathcal{F}'$-WL.

But all patterns in $\mathcal{F}'$ must have treewidth less than $k$, and by (b) $G$ and $H$ are indistinguishable by $k$-WL. Proposition 3 then implies that $G$ and $H$ are indistinguishable by $\mathcal{F}$-WL, as desired.

# D   Connections to existing formalisms

We here provide more details of how $\mathcal{F}$-MPNNs connect to MPNNs from the literature which also augment the initial labelling.

**Vertex degrees.**   We first consider so-called *degree-aware* MPNNs Geerts et al. [2021] in which the message functions of the MPNNs may depend on the vertex degrees. The Graph Convolution Networks (GCNs) Kipf and Welling [2017] are an example of such MPNNs. Degree-aware MPNNs are known to be equivalent, in terms of expressive power, to standard MPNNs in which the initial labelling is extended with vertex degrees Geerts et al. [2021]. Translated to our setting, we can simply let $\mathcal{F} = \{\begin{smallmatrix}\bullet\\\circ\end{smallmatrix}\}$ since $\mathsf{hom}(\begin{smallmatrix}\bullet\\\circ\end{smallmatrix}, G^v)$ is equal to the degree of vertex $v$ in $G$. When considering graphs without an initial vertex labelling (or a uniform labelling which assigns every vertex the same label), our characterisation (Theorem 1) implies $G \equiv^{(d)}_{\begin{smallmatrix}\bullet\\\circ\end{smallmatrix}\text{-WL}} H$ if and only if $\mathsf{hom}(T, G) = \mathsf{hom}(T, H)$ for every $\{\begin{smallmatrix}\bullet\\\circ\end{smallmatrix}\}$-pattern tree of depth at most $d$. This in turn is equivalent to $\mathsf{hom}(T, G) = \mathsf{hom}(T, H)$ for every (standard) tree of depth at most $d + 1$. Indeed, $\{\begin{smallmatrix}\bullet\\\circ\end{smallmatrix}\}$-pattern trees of depth at most $d$ are simply trees of depth $d + 1$. Combining this with the characterisation of WL by Dvorak [2010] and Dell et al. [2018], we thus have for unlabelled graphs that $G \equiv^{(d)}_{\begin{smallmatrix}\bullet\\\circ\end{smallmatrix}\text{-WL}} H$ if and only if $G \equiv^{(d+1)}_{\text{WL}} H$.

So, by considering $\mathcal{F} = \{\begin{smallmatrix}\bullet\\\circ\end{smallmatrix}\}$-MPNNs one gains one round of computation compared to considering standard MPNNs. To lift this to labeled graphs, instead of $\mathcal{F} = \{\begin{smallmatrix}\bullet\\\circ\end{smallmatrix}\}$ one has to include labeled versions of the single edge pattern, in order to count the number of neighbours of a specific label for each vertex. This is done, e.g., by Ishiguro et al. [2020], who use the WL labelling obtained after the first round to augment the initial vertex labelling. This corresponds indeed by adding $\mathsf{hom}(T^r, G^v)$ as feature for every labeled tree of depth one. This results in that $G \equiv^{(d)}_{\begin{smallmatrix}\bullet\\\circ\end{smallmatrix}\text{-WL}} H$ if and only if $G \equiv^{(d+1)}_{\text{WL}} H$ for labelled graphs.

**Walk counts.**   The *Graph Feature Networks* by Chen et al. [2019a] can be regarded as a generalisation of the previous approach. Instead of simply adding vertex degrees, the number of walks of certain lengths emanating from vertices are added. Translated to our setting, this corresponds to considering $\{L_2, L_3, \ldots, L_\ell\}$-MPNNs, where $L_\ell$ denotes a rooted path of length $\ell$. For unlabelled graphs, our characterisation (Theorem 1) implies that $G \equiv^{(d)}_{L_1, \ldots, L_\ell\text{-WL}} H$ is upper bounded by $G \equiv^{(d+\ell)}_{\text{WL}} H$, simply because every $\{L_2, L_3, \ldots, L_\ell\}$-pattern tree of depth $d$ is a standard tree of depth at most $d + \ell$.

**Cycles.**   Li et al. [2019] extend MPNNs by varying the notion of neighbourhood over which is aggregated. One particular instance corresponds to an aggregation of features, weighted by the number of cycles of a certain length in each vertex (see discussion at the end of Section 4 in Li et al. [2019]). Translated to our setting, this corresponds to considering $\{C_\ell\}$-MPNNs where $C_\ell$ denotes the cycle of length $\ell$. As mentioned in the main body of the paper, these extend MPNNs and result in architectures bounded by 2-WL (Proposition 5). This is in line with Theorem 3 from Li et al. [2019] stating that their framework strictly extends MPNNs and thus 1-WL.

**Isomorphism counts.**   Another, albeit similar, approach to add structural information to the initial labelling is taken in the paper *Graph Substructure Networks* by Bouritsas et al. [2020]. The idea there is to extend the initial features with information about how often a vertex $v$ appears in a subgraph of $G$ which is isomorphic to $P$More precisely, Bouritsas et al. [2020] consider a connected unlabelled graph $P$ as pattern and partition its vertex set $V_P$ orbit-wise. That is, $V_P = \biguplus_{i=1}^{o_P} V_P^i$ where $o_P$ denotes the number of orbits of $P$. Here, $v, v' \in V_P^i$ whenever there is an automorphism $h$ in $\mathsf{Aut}(P)$ mapping $v$ to $v'$. Next, they consider all distinct subgraphs $G_1, \ldots, G_k$ in $G$ which are isomorphic to $P$, denoted by $P \cong G_j$ for $j \in [k]$. We write $P \cong_f G_j$ when $P \cong G_j$ using a specific isomorphism $f$. Then for each orbit partition $i \in [o_P]$ and vertex $v \in V$, they define:

$$\mathsf{iso}(P, G, v, i) = |\{G_j \cong P \mid v \in V_{G_j}, \text{and there exists an } f \text{ s.t. } G_j \cong_f P \text{ and } f(v) \in V_P^i, j \in [k]\}|.$$

That is, the number of subgraphs $G_j$ in $G$ that can be isomorphically mapped to $P$ are counted, provided that this can be done by an isomorphism which maps vertex $v$ in $G_j$ (and thus $G$) to one of the vertices in the $i$th orbit partition $V_P^i$ of the pattern. A similar notion is proposed for edges, which we will not consider here. Similar to our extended features, the initial features of each vertex $v$ is then augmented with $\big(\mathsf{iso}(P, G^v, i) \mid P \in \mathcal{F}, i \in [o_P]\big)$ for some set $\mathcal{F}$ of patterns. Standard MPNNs are executed on these augmented initial features. We refer to Bouritsas et al. [2020] for more details.

We can view the above approach as an instance of our framework. Indeed, given a pattern $P$ in $\mathcal{F}$, for each orbit partition, we replace $P$ by a different rooted version $P^{r_i}$, where $r_i$ is a vertex in $V_P^i$. Which vertex in the orbit under consideration is selected as root is not important (because they are equivalent by definition of orbit). We then see that the standard notion of subgraph isomorphism counting directly translates to the quantity used in Bouritsas et al. [2020]:

$$\mathsf{sub}(P^{r_i}, G^v) := \text{number of subgraphs in } G \text{ containing } v, \text{ isomorphic to } P^{r_i} = \mathsf{iso}(P, G, v, i).$$

It thus remains to express $\mathsf{sub}(P^{r_i}, G^v)$ in terms of homomorphism counts. This, however, follows from Curticapean et al. [2017] in which it is shown that $\mathsf{iso}(P^{r_i}, G^v)$ can be computed by a linear combination of $\mathsf{hom}(Q^{r_i}, G^v)$ where $Q^{r_i}$ ranges over all graphs on which $P^{r_i}$ can be mapped by means of a surjective homomorphism. For a given $P^{r_i}$, the finite set of such patterns is called the *spasm* of $P^{r_i}$ in Curticapean et al. [2017] and can be easily computed.

Proposition 2 now readily follows. Indeed, consider a $\mathcal{P}$-GSN and replace each $P \in \mathcal{P}$ by its rooted versions $P^{r_i}$, for $i \in [o_P]$. Let $\mathcal{P}^+$ be the resulting set of (rooted) patterns. Then, the results by Curticapean et al. [2017] imply that $\mathsf{sub}(P^{r_i}, G^v)$ can be computed in terms of $\mathsf{hom}(Q^r, G^v)$, where $Q^r$ is a pattern in the spasm $\mathsf{s}(\mathcal{P}^+)$ of $\mathcal{P}^+$. As a consequence, the expressive power of $\mathcal{P}$-GSNs is bounded by the power of $\mathsf{s}(\mathcal{P}^+)$-MPNNs (and thus also by the power of $\mathsf{s}(\mathcal{P}^+)$-WL).

Conversely, given an $\mathcal{F}$-MPNN one can, again using results by Curticapean et al. [2017], define a set $\mathcal{F}^\star$ of patterns, such that the subgraph isomorphism counts of patterns in $\mathcal{F}^\star$ can be used to compute the homomorphism counts of patterns in $\mathcal{F}$. Here, the set $\mathcal{F}^\star$ consists of the extensions of patterns in $\mathcal{F}$. An extension of a graph is a supergraph over the same set of vertices. As a consequence, $\mathcal{F}$-MPNNs are upper bounded by $\mathcal{F}^\star$-GSNs. This is all in agreement with Curticapean et al. [2017] in which it is shown that homomorphism counts, subgraph isomorphism counts and other notions of pattern counts are all interchangeable. Nevertheless, by using homomorphism counts one can gracefully extend known results about WL and MPNNs, as we have shown in the paper, and add little overhead.

We conclude this section be a sketch of the proof of Theorem 2. We already observed that given a $\mathcal{P}$-GSN we can view it as an MPNN using rooted graph patterns $\mathcal{P}^+$. The difference now lies in that subgraph isomorphism counts rather than homomorphism counts are used. If we inspect, however, the proof of Theorem 1, then one sees that the two most important properties of homomorphism counts used are: (a) $\mathsf{hom}(P^r \star Q^r, G^v) = \mathsf{hom}(P^r, G^v) \cdot \mathsf{hom}(Q^r, G^v)$; and (b) if we consider a $\mathcal{P}^+$-tree $T^r$ of depth $d$, then $\mathsf{hom}(T^r, G^v)$ decomposes in an expression using homomorphism counts of $\mathcal{P}^+$-trees of depth $d-1$. It is important to observe that these properties do not necessarily hold when replacing homomorphism counts by subgraph isomorphism counts, however. Nevertheless, they do hold for the revised notion of homomorphism defined in the main paper. Indeed, when considering $\mathsf{shom}(T^r, G^v)$ we treat the backbone tree of $T^r$ differently from the patterns joined at its vertices. More precisely, we only count those mappings $h : V_T \to V_G$ such that $h$ is a homomorphism on the backbone of $T^r$, whilst for the joined patterns at the backbone's vertices, we require local isomorphisms. Moreover, we require $h(P^r, G^u)$ to be isomorphic to $P^r$ for every copy of $P^r$ joined with a vertex $t$ in the backbone of $T^r$. These conditions imply that the function $\mathsf{shom}$ satisfies properties (a) and (b) used in the proof of Theorem 1 and replacing $\mathsf{hom}$ by $\mathsf{shom}$ in the proof suffices to show Theorem 2.

# E    Additional experimental information

## E.1    Experimental setup

One of the crucial questions when studying the effect of adding structural information to the initial vertex labels is whether these additional labels enhance the performance of graph neural networks. In order to reduce the effect of specific implementation details of GNNs and choice of hyper-parameters,

we start from the GNN implementations and choices made in the benchmark by Dwivedi et al. [2020][6]. and only change the initial vertex labels, while leaving the GNNs themselves unchanged. This ensures that we only measure the effect of augmenting initial features with homomorphism counts. We will use the GNNs from the benchmark, without extended features, as our baselines. For the same reasons, we use datasets proposed in the benchmark for their ability to statistically separate the performance of GNNs. All other parameters are taken as in Dwivedi et al. [2020] and we refer to that paper for more details.

**Selected GNNs**    Dwivedi et al. [2020] divide the benchmarked GNNs into two classes: the MPNNs and the "theoretically designed" WL-GNNs. The first class is found to perform stronger and train faster. Hence, we chose to include the five following MPNN models from the benchmark:

- Graph Attention Network (GAT) as described in Velickovic et al. [2018]
- Graph Convolutional Network (GCN) as described in Kipf and Welling [2017]
- GraphSage as described in Hamilton et al. [2017]
- Mixed Model Convolutional Networks (MoNet) as described in Monti et al. [2017]
- GatedGCN as described in Bresson and Laurent [2017].

For GatedGCN we used the version in which positional encoding Belkin and Niyogi [2003] is added to the vertex features, as it is empirically shown to be the strongest performing version of this model by for the selected datasets Dwivedi et al. [2020]. We denote this version by $GatedGCN_{E,PE}$, referring to the presence of edge features and this positional encoding. Details, background and a mathematical formalization of the message passing layers of these models can be found in the supplementary material of Dwivedi et al. [2020].

As explained in the experimental section of the main paper, we enhance the vertex features with the log-normalized counts of the chosen patterns in every vertex of every graph of every dataset. The first layers of some models of Dwivedi et al. [2020] are adapted to take in this variation in input size. All other layers where left identical to their original implementation as provided by Dwivedi et al. [2020].

**Hardware, compute and resources**    All models for ZINC, PATTERN and COLLAB were trained on a GeForce GTX 1080 337 Ti GPU, for CLUSTER a Tesla V100-SXM3-32GB GPU was used. Tables 6, 9, 12 and 15 report the training times for all combination of models and additional feature set. A rough estimate of the $CO_2$ emissions based on the total computing times of reported experiments (2 074 hours GeForce GTX 1080, 372 hours Tesla V100-SXM3-32GB), the computing times of not-included experiments (1 037 hours GeForce GTX 1080, 181 hours Tesla V100-SXM3-32GB), the GPU types (GeForce GTX 1080, Tesla V100-SXM3-32GB) and the geographical location (undisclosed to preserve anonymity) of our cluster results in a carbon emission of 135 kg $CO_2$ equivalent. This estimation was conducted using the MachineLearning Impact calculator presented in Lacoste et al. [2019].

## E.2    Graph learning tasks

We here report the full results of our experimental evaluation for graph regression (Section E.2.1), link prediction (Section E.2.2) and vertex classification (Section E.2.3) as considered in Dwivedi et al. [2020]. More precisely, a full listing of the patterns and combinations used and the obtained results for the test sets can be found in Tables 4, 7, 10 and 13. Average training time (in hours) and the number of epochs are reported in Tables 6, 9, 12 and 15. Finally, the total number of model parameters are reported in Tables 5, 8, 11 and 14. All averages and standard deviations are over 4 runs with different random seeds. The main take-aways from these results are included in the main paper.

### E.2.1    Graph regression with the ZINC dataset

Just as in Dwivedi et al. [2020] we use a subset (12K) of ZINC molecular graphs (250K) dataset Irwin et al. [2012] to regress a molecular property known as the constrained solubility. For each molecular graph, the vertex features are the types of heavy atoms and the edge features are the types

---

[6]The original implementations can be found on https://github.com/graphdeeplearning/benchmarking-gnns

of bonds between them. The following are taken from Dwivedi et al. [2020]:

**Splitting**. ZINC has $10\,000$ train, $1\,000$ validation and $1\,000$ test graphs.

**Training**.[7] For the learning rate strategy, an initial learning rate is set to $5 \times 10^{-5}$ , the reduce factor is 0.5, and the stopping learning rate is $1 \times 10^{-6}$, the patience value is 25 and the maximal training time is set to 12 hours.

**Performance Measure** The performance measure is the mean absolute error (MAE) between the predicted and the ground truth constrained solubility for each molecular graph.

**Number of layers** Most experiments are performed with 16 MPNN layers, following the best performers in the benchmark. We also report results using 4 MPNN layers.

**Hidden feature size** The hidden feature sizes are (for GAT, GCN, GraphSage, MoNet and GatedGCN respectively) 144, 145, 108, 90 and 70.

Table 4: Full results of the mean absolute error (predicted constrained solubility vs. the ground truth) for selected cycle combinations and GNNs on the ZINC data set. The top part of the table refers to experiments using 16 layers, the bottom part to experiments using 4 layers. In addition, in the last two rows of each part of the table we compare between homomorphism counts (hom) and subgraph isomorphism counts (iso).

| Pattern set $\mathcal{F}$ | GAT | GCN | GraphSage | MoNet | GatedGCN$_{E,PE}$ |
|---|---|---|---|---|---|
| None | 0,47±0,02 | 0,35±0,01 | 0,25±0,01 | 0,44±0,01 | 0,34±0,05 |
| $\{C_3\}$ | 0,45±0,01 | 0,36±0,01 | 0,25±0,00 | 0,44±0,00 | 0,30±0,01 |
| $\{C_4\}$ | 0,34±0,02 | 0,29±0,02 | 0,26±0,01 | 0,30±0,01 | 0,27±0,06 |
| $\{C_5\}$ | 0,44±0,02 | 0,34±0,02 | 0,23±0,01 | 0,42±0,01 | 0,27±0,03 |
| $\{C_6\}$ | 0,31±0,00 | 0,27±0,02 | 0,25±0,01 | 0,30±0,01 | 0,26±0,09 |
| $\{C_3,C_4\}$ | 0,33±0,01 | 0,27±0,01 | 0,24±0,02 | 0,32±0,01 | 0,23±0,03 |
| $\{C_5,C_6\}$ | 0,28±0,01 | 0,26±0,01 | 0,23±0,01 | 0,28±0,01 | 0,20±0,03 |
| $\{C_4,C_5,C_6\}$ | 0,24±0,00 | 0,21±0,00 | 0,20±0,00 | 0,25±0,01 | 0,16±0,02 |
| $\{C_3,C_4,C_5,C_6\}$ | 0,23±0,00 | 0,21±0,00 | 0,20±0,01 | 0,26±0,02 | 0,18±0,02 |
| $\{C_3,\ldots,C_{10}\}$ (hom) | **0,22±0,01** | **0,20±0,00** | 0,19±0,00 | **0,2376±0,01** | **0,1352±0,01** |
| $\{C_3,\ldots,C_{10}\}$ (iso) | 0,24±0,01 | 0,22±0,01 | **0,16±0,01** | 0,2408±0,01 | 0,1357 ± 0,01 |

| | | | | | |
|---|---|---|---|---|---|
| None | 0,48±0,01 | 0,46±0,00 | 0,36±0,01 | 0,45±0,00 | 0,26±0,03 |
| $\{C_3,\ldots,C_{10}\}$ (hom) | **0,20±0,02** | 0,25±0,02 | 0,19±0,01 | **0,17±0,01** | 0,13±0,01 |
| $\{C_3,\ldots,C_{10}\}$ (iso) | 0,21±0,00 | **0,23±0,03** | **0,16±0,01** | 0,19±0,01 | **0,11±0,01** |

### E.2.2 Link Prediction with the Collab dataset

Another set used in Dwivedi et al. [2020] is COLLAB, a link prediction dataset proposed by the Open Graph Benchmark (OGB) Hu et al. [2020] corresponding to a collaboration network between approximately 235K scientists, indexed by Microsoft Academic Graph. Vertices represent scientists and edges denote collaborations between them. For vertex features, OGB provides 128-dimensional vectors, obtained by averaging the word embeddings of a scientist's papers. The year and number of co-authored papers in a given year are concatenated to form edge features. The graph can also be viewed as a dynamic multi-graph, since two vertices may have multiple temporal edges between if they collaborate over multiple years. The following are taken from Dwivedi et al. [2020]:

**Splitting.** We use the real-life training, validation and test edge splits provided by OGB. Specifically, they use collaborations until 2017 as training edges, those in 2018 as validation edges, and those in 2019 as test edges.

**Training.** All GNNs use the same learning rate strategy: an initial learning rate is set to $1 \times 10^{-3}$ , the reduce factor is 0.5, the patience value is 10, and the stopping learning rate is $1 \times 10^{-5}$ .

**Performance Measure.** We use the evaluator provided by OGB Hu et al. [2020], which aims to measure a model's ability to predict future collaboration relationships given past collaborations. Specifically, they rank each true collaboration among a set of $100\,000$ randomly-sampled negative collaborations, and count the ratio of positive edges that are ranked at $K$-place or above (Hits@K). The value $K = 50$ as this gives the best value for statistically separating the performance of GNNs.

---

[7]Here and in the next tasks we are using the parameters used in the code accompanying Dwivedi et al. [2020]. In the paper, slightly different parameters are used.

Table 5: Total model parameters for selected cycle combinations and GNNs on the ZINC data set. The top part of the table refers to experiments using 16 layers, the bottom part to experiments using 4 layers. In addition, in the last two rows of each part of the table we compare between homomorphism counts (hom) and subgraph isomorphism counts (iso).

| Pattern set $\mathcal{F}$ | GAT | GCN | GraphSage | MoNet | GatedGCN$_{E,PE}$ |
|---|---|---|---|---|---|
| None | 358 273 | 360 742 | 388 963 | 401 148 | 408 135 |
| $\{C_3\}$ | 358 417 | 360 887 | 389 071 | 401 238 | 408 205 |
| $\{C_4\}$ | 358 417 | 360 887 | 389 071 | 401 238 | 408 205 |
| $\{C_5\}$ | 358 417 | 360 887 | 389 071 | 401 238 | 408 205 |
| $\{C_6\}$ | 358 417 | 360 887 | 389 071 | 401 238 | 408 205 |
| $\{C_3, C_4\}$ | 358 561 | 361 032 | 389 179 | 401 328 | 408 275 |
| $\{C_5, C_6\}$ | 358 561 | 361 032 | 389 179 | 401 328 | 408 275 |
| $\{C_4, C_5, C_6\}$ | 358 705 | 361 177 | 389 287 | 401 418 | 408 345 |
| $\{C_3, C_4, C_5, C_6\}$ | 358 849 | 361 322 | 389 395 | 401 508 | 408 415 |
| $\{C_3, \ldots, C_{10}\}$ (hom) | 359 425 | 361 902 | 389 827 | 401 868 | 408 695 |
| $\{C_3, \ldots, C_{10}\}$ (iso) | 359 425 | 361 902 | 389 827 | 401 868 | 408 695 |
| None | 102529 | 103222 | 105139 | 106092 | 106575 |
| $\{C_3, \ldots, C_{10}\}$ (hom) | 103681 | 104382 | 106003 | 106812 | 107135 |
| $\{C_3, \ldots, C_{10}\}$ (iso) | 103681 | 104382 | 106003 | 106812 | 107135 |

Table 6: Average training time in hours and number of epochs for selected cycle combinations and GNNs on the ZINC data set. The top part of the table refers to experiments using 16 layers, the bottom part to experiments using 4 layers. In addition, in the last two rows of each part of the table we compare between homomorphism counts (hom) and subgraph isomorphism counts (iso).

| Model: | GAT | | GCN | | GraphSage | | MoNet | | GatedGCN$_{E,PE}$ | |
|---|---|---|---|---|---|---|---|---|---|---|
| Pattern set $\mathcal{F}$ | Time | Epochs | Time | Epochs | Time | Epochs | Time | Epochs | Time | Epochs |
| None | 2,40 | 377 | 10,99 | 463 | 2,46 | 420 | 1,53 | 345 | 12,08 | 136 |
| $\{C_3\}$ | 2,88 | 444 | 12,03 | 363 | 2,03 | 500 | 0,91 | 298 | 12,07 | 148 |
| $\{C_4\}$ | 2,30 | 351 | 11,36 | 324 | 2,31 | 396 | 1,70 | 382 | 12,06 | 139 |
| $\{C_5\}$ | 2,42 | 375 | 12,03 | 333 | 1,70 | 444 | 1,06 | 370 | 12,06 | 202 |
| $\{C_6\}$ | 2,40 | 369 | 9,98 | 421 | 2,58 | 446 | 1,25 | 288 | 12,08 | 136 |
| $\{C_3, C_4\}$ | 2,98 | 461 | 12,03 | 332 | 2,56 | 458 | 1,41 | 321 | 12,09 | 132 |
| $\{C_5, C_6\}$ | 2,76 | 422 | 12,04 | 319 | 2,67 | 464 | 1,53 | 356 | 12,06 | 137 |
| $\{C_4, C_5, C_6\}$ | 2,45 | 381 | 10,13 | 419 | 1,67 | 463 | 1,04 | 382 | 12,04 | 229 |
| $\{C_3, C_4, C_5, C_6\}$ | 2,65 | 408 | 10,38 | 420 | 2,09 | 503 | 1,26 | 364 | 12,08 | 135 |
| $\{C_3, \ldots, C_{10}\}$ (hom) | 2,65 | 428 | 12,03 | 350 | 2,76 | 478 | 1,48 | 363 | 12,06 | 175 |
| $\{C_3, \ldots, C_{10}\}$ (iso) | 2,78 | 497 | 11,72 | 419 | 2,63 | 547 | 1,58 | 440 | 11,62 | 148 |
| None | 0,32 | 158 | 2,14 | 201 | 0,26 | 171 | 0,28 | 184 | 5,47 | 223 |
| $\{C_3, \ldots, C_{10}\}$ (hom) | 0,33 | 166 | 2,35 | 190 | 0,16 | 164 | 0,36 | 207 | 5,80 | 210 |
| $\{C_3, \ldots, C_{10}\}$ (iso) | 0,26 | 176 | 1,66 | 182 | 0,20 | 186 | 0,31 | 222 | 5,06 | 272 |

**Number of layers** 3 MPNN layers are used for every model.
**Hidden feature size** The hidden feature sizes are (for GAT, GCN, GraphSage, MoNet and GatedGCN respectively) 57, 74, 38, 53 and 35.

### E.2.3 Vertex classification with PATTERN and CLUSTER

Finally, also used in Dwivedi et al. [2020] are the PATTERN and CLUSTER graph data sets, generated with the Stochastic Block Model (SBM) Abbe [2018], which is widely used to model communities in social networks by modulating the intra- and extra-communities connections, thereby controlling the difficulty of the task. A SBM is a random graph which assigns communities to each vertex as follows: any two vertices are connected with probability $p$ if they belong to the same community, or they are connected with probability $q$ if they belong to different communities (the value of $q$ acts as the noise level).

Table 7: Full Results (Hits @50) for all selected pattern combinations and GNNs on the COLLAB data set.

| Pattern set $\mathcal{F}$ | GAT | GCN | GraphSage | MoNet | GatedGCN$_{E,PE}$ |
|---|---|---|---|---|---|
| None | 50,32±0,55 | 51,36±1,30 | 49,81±1,56 | 50,33±0,68 | 51,00±2,54 |
| $\{K_3\}$ | **52,87±0,87** | 53,57±0,89 | 50,18±1,38 | 51,10±0,38 | **51,57±0,68** |
| $\{K_4\}$ | 51,33±1,42 | 52,84±1,32 | **51,76±1,38** | 51,13±1,60 | 49,43±1,85 |
| $\{K_5\}$ | 52,41±0,89 | **54,60±1,01** | 50,94±1,30 | **51,39±1,23** | 50,31±1,59 |
| $\{K_3, K_4\}$ | 52,68±1,82 | 53,49±1,35 | 50,88±1,73 | 50,97±0,68 | 51,36±0,92 |
| $\{K_3, K_4, K_5\}$ | 51,81±1,17 | 54,32±1,02 | 49,94±0,23 | 51,01±1,00 | 51,11±1,06 |

Table 8: Total number of model parameters for all selected pattern combinations and GNNs on the COLLAB data set.

| Pattern set $\mathcal{F}$ | GAT | GCN | GraphSage | MoNet | GatedGCN$_{E,PE}$ |
|---|---|---|---|---|---|
| None | 25 992 | 40 479 | 39 751 | 26 487 | 27 440 |
| $\{K_3\}$ | 26 049 | 40 553 | 39 804 | 26 525 | 27 475 |
| $\{K_4\}$ | 26 049 | 40 553 | 39 804 | 26 525 | 27 475 |
| $\{K_5\}$ | 26 049 | 40 553 | 39 804 | 26 525 | 27 475 |
| $\{K_3, K_4\}$ | 26 106 | 40 627 | 39 857 | 26 563 | 27 510 |
| $\{K_3, K_4, K_5\}$ | 26 163 | 40 701 | 39 910 | 26 601 | 27 545 |

Table 9: Average training times and number of epochs for all selected pattern combinations and GNNs on the COLLAB data set.

| Model: | GAT | | GCN | | MoNet | | GraphSage | | GatedGCN$_{E,PE}$ | |
|---|---|---|---|---|---|---|---|---|---|---|
| Pattern set $\mathcal{F}$ | Time | #Epochs | Time | #Epochs | Time | #Epochs | Time | #Epochs | Time | #Epochs |
| None | 0,81 | 167 | 0,85 | 141 | 1,62 | 190 | 12,05 | 115,67 | 2,22 | 167 |
| $\{K_3\}$ | 0,67 | 165 | 0,90 | 153 | 1,70 | 184 | 12,10 | 67,00 | 2,48 | 186 |
| $\{K_4\}$ | 1,06 | 188 | 0,95 | 160 | 2,16 | 188 | 12,04 | 113,50 | 1,26 | 188 |
| $\{K_5\}$ | 0,50 | 167 | 1,13 | 165 | 1,04 | 193 | 12,05 | 124,00 | 1,82 | 174 |
| $\{K_3, K_4\}$ | 1,20 | 189 | 0,86 | 128 | 2,15 | 189 | 12,05 | 113,25 | 1,51 | 183 |
| $\{K_3, K_4, K_5\}$ | 0,44 | 149 | 0,90 | 134 | 0,98 | 186 | 12,05 | 124,00 | 1,84 | 177 |

For the PATTERN dataset, the goal of the vertex classification problem is the detection of a certain pattern $P$ embedded in a larger graph $G$. The graphs in $G$ consist of 5 communities with sizes randomly selected between $[5, 35]$. The parameters of the SBM for each community is $p = 0.5$, $q = 0.35$, and the vertex features in $G$ are generated using a uniform random distribution with a vocabulary of size 3, i.e., $\{0, 1, 2\}$. Randomly, 100 patterns $P$ composed of 20 vertices with intra-probability $p_P = 0.5$ and extra-probability $q_P = 0.5$ are generated (i.e., 50% of vertices in $P$ are connected to $G$). The vertex features for $P$ are also generated randomly using values in $\{0, 1, 2\}$. The graphs consist of 44-188 vertices. The output vertex labels have value 1 if the vertex belongs to $P$ and value 0 belongs to $G$.

For the CLUSTER dataset, the goal of the vertex classification is the detection of which cluster a vertex belongs. Here, six SBM clusters are generated with sizes randomly selected between $[5, 35]$ and probabilities $p = 0.55$ and $q = 0.25$. The graphs consist of 40-190 vertices. Each vertex can take an initial feature value in range $\{0, 1, 2, \ldots, 6\}$. If the value is $i$ then the vertex belongs to class $i - 1$. If the value is 0, then the class of the vertex is unknown and need to be inferred. There is only one labelled vertex that is randomly assigned to each community and most vertex features are set to 0. The output vertex labels are defined as the community/cluster class labels.

The following are taken from Dwivedi et al. [2020]:

**Splitting** The PATTERN dataset has 10 000 train, 2 000 validation and 2 000 test graphs. The CLUSTER dataset has 10 000 train, 1 000 validation and 1 000 test graphs. We save the generated splits and use the same sets in all models for fair comparison.

**Training** For all GNNs, an initial learning rate is set to $1 \times 10^{-3}$, the reduce factor is 0.5, the patience value is 10, and the stopping learning rate is $1 \times 10^{-5}$ .

**Performance measure** The performance measure is the average vertex-level accuracy weighted with respect to the class sizes.

**Number of layers** 16 MPNN layers are used for every model, following the best performers in the benchmark. For the PATTERN dataset, we also report results using 4 MPNN layers.

**Hidden feature size** The hidden feature sizes are (for GAT, GCN, GraphSage, MoNet and GatedGCN respectively) 136, 146, 108, 90 and 70.

Table 10: Full results of the weighted accuracy for selected pattern combinations and GNNs on the CLUSTER data set.

| Pattern set $\mathcal{F}$ | GAT | GCN | MoNet | GraphSage | GatedGCN$_{E,PE}$ |
|---|---|---|---|---|---|
| None | 70,86±0,06 | **70,64±0,39** | 71,15±0,33 | 72,25±0,52 | **74,28±0,15** |
| $\{K_3\}$ | 71,60±0,15 | 64,88±4,16 | 72,21±0,19 | 72,97±0,23 | 74,14±0,12 |
| $\{K_4\}$ | 71,40±0,24 | 60,64±2,93 | 72,14±0,19 | 72,57±0,19 | 74,16±0,24 |
| $\{K_5\}$ | 71,26±0,39 | 66,60±1,47 | **72,34±0,09** | 72,60±0,24 | 74,23±0,07 |
| $\{K_3, K_4\}$ | **71,80±0,28** | 50,94±22,98 | 72,32±0,27 | **73,03±0,25** | 74,17±0,13 |
| $\{K_3, K_4, K_5\}$ | 71,63±0,26 | 63,03±3,72 | 72,32±0,36 | 72,65±0,13 | 74,03±0,19 |

Table 11: Total number of model parameters for all selected pattern combinations and GNNs on the CLUSTER data set.

| Pattern set $\mathcal{F}$ | GAT | GCN | MoNet | GraphSage | GatedGCN$_{E,PE}$ |
|---|---|---|---|---|---|
| None | 395 396 | 362 849 | 399 373 | 386 835 | 406 755 |
| $\{K_3\}$ | 395 396 | 362 849 | 399 373 | 386 835 | 406 755 |
| $\{K_4\}$ | 395 548 | 362 995 | 399 463 | 386 943 | 406 825 |
| $\{K_5\}$ | 395 700 | 363 141 | 399 553 | 387 051 | 406 895 |
| $\{K_3, K_4\}$ | 395 700 | 363 141 | 399 553 | 387 051 | 406 895 |
| $\{K_3, K_4, K_5\}$ | 396 004 | 363 433 | 399 733 | 387 267 | 407 035 |

Table 12: Training times (in hours) and number of epochs for all selected pattern combinations and GNNs on the CLUSTER data set.

| Model: | GAT | | GCN | | MoNet | | GraphSage | | GatedGCN$_{E,PE}$ | |
|---|---|---|---|---|---|---|---|---|---|---|
| Pattern set $\mathcal{F}$ | Time | #Epochs | Time | #Epochs | Time | #Epochs | Time | #Epochs | Time | #Epochs |
| None | 1,62 | 109 | 2,83 | 117 | 1,54 | 125 | 0,95 | 101 | 10,40 | 92 |
| $\{K_3\}$ | 1,52 | 107 | 2,67 | 85 | 1,72 | 145 | 1,08 | 102 | 11,01 | 89 |
| $\{K_4\}$ | 1,18 | 107 | 1,94 | 80 | 1,62 | 149 | 0,90 | 102 | 10,23 | 90 |
| $\{K_5\}$ | 1,23 | 106 | 2,30 | 84 | 1,68 | 143 | 0,92 | 99 | 10,68 | 91 |
| $\{K_3, K_4\}$ | 1,53 | 102 | 1,97 | 82 | 1,89 | 153 | 0,94 | 99 | 10,80 | 90 |
| $\{K_3, K_4, K_5\}$ | 1,62 | 105 | 1,96 | 82 | 1,95 | 157 | 0,97 | 100 | 10,25 | 91 |

Table 13: Full results of the weighted accuracy for selected pattern combinations and GNNs on the PATTERN data set. The top of the table refers to experiments using 16 layers, the bottom part to experiments using 4 layers.

| Pattern set $\mathcal{F}$ | GAT | GCN | MoNet | GraphSage | GatedGCN$_{E,PE}$ |
|---|---|---|---|---|---|
| None | 78,83±0,60 | 71,42±1,38 | 85,90±0,03 | 70,78±0,19 | **86,15±0,08** |
| $\{K_3\}$ | 84,34±0,09 | 61,54±2,20 | 86,59±0,02 | 84,75±0,11 | 85,02±0,20 |
| $\{K_4\}$ | 84,43±0,40 | 63,40±1,55 | 86,60±0,02 | 84,51±0,06 | 85,40±0,28 |
| $\{K_5\}$ | 83,47±0,11 | 64,18±3,88 | 86,57±0,02 | 83,73±0,10 | 85,63±0,22 |
| $\{K_3, K_4\}$ | 85,44±0,24 | 81,29±2,82 | 86,58±0,02 | 85,85±0,13 | 85,80±0,20 |
| $\{K_3, K_4, K_5\}$ | **85,50±0,23** | **82,49±0,48** | **86,63±0,03** | **85,88±0,15** | 85,56±0,33 |
|  |  |  |  |  |  |
| None | 77,64±1,66 | 61,23±0,57 | 85,82±0,05 | 70,85±1,25 | 85,94±0,08 |
| $\{K_3, K_4, K_5\}$ | **86,54±0,07** | **83,89±0,81** | **86,63±0,03** | **86,50±0,05** | **85,98±0,11** |

Table 14: Total number of model parameters for selected pattern combinations and GNNs on the PATTERN data set. The top part of the table refers to experiments using 16 layers, the bottom part to experiments using 4 layers.

| Pattern set $\mathcal{F}$ | GAT | GCN | MoNet | GraphSage | GatedGCN$_{E,PE}$ |
|---|---|---|---|---|---|
| None | 394 632 | 362 117 | 398 921 | 386 291 | 406 403 |
| $\{K_3\}$ | 394 784 | 362 263 | 399 011 | 386 399 | 406 473 |
| $\{K_4\}$ | 394 784 | 362 263 | 399 011 | 386 399 | 406 473 |
| $\{K_5\}$ | 394 784 | 362 263 | 399 011 | 386 399 | 406 473 |
| $\{K_3, K_4\}$ | 394 936 | 362 409 | 399 101 | 386 507 | 406 543 |
| $\{K_3, K_4, K_5\}$ | 395 088 | 362 555 | 399 191 | 386 615 | 406 613 |
|  |  |  |  |  |  |
| None | 110088 | 101069 | 103865 | 102467 | 104843 |
| $\{K_3, K_4, K_5\}$ | 110544 | 101507 | 104135 | 102791 | 105053 |

Table 15: Training times (in hours) and number of epochs for selected pattern combinations and GNNs on the PATTERN data set. The top part of the table refers to experiments using 16 layers, the bottom part to experiments using 4 layers.

| Model: | GAT | | GCN | | MoNet | | GraphSage | | GatedGCN$_{E,PE}$ | |
|---|---|---|---|---|---|---|---|---|---|---|
| Pattern set $\mathcal{F}$ | Time | Epochs | Time | Epochs | Time | Epochs | Time | Epochs | Time | Epochs |
| None | 1,96 | 87 | 3,41 | 102 | 1,68 | 116 | 0,77 | 103 | 10,32 | 101 |
| $\{K_3\}$ | 0,97 | 97 | 2,58 | 80 | 1,42 | 107 | 0,69 | 105 | 9,12 | 95 |
| $\{K_4\}$ | 0,90 | 90 | 2,68 | 80 | 1,46 | 106 | 0,67 | 95 | 9,47 | 94 |
| $\{K_5\}$ | 0,89 | 95 | 2,36 | 80 | 1,26 | 100 | 0,58 | 98 | 9,14 | 99 |
| $\{K_3, K_4\}$ | 2,11 | 91 | 3,62 | 98 | 1,68 | 108 | 0,86 | 97 | 9,50 | 87 |
| $\{K_3, K_4, K_5\}$ | 1,02 | 91 | 3,26 | 94 | 1,48 | 109 | 0,76 | 102 | 8,84 | 88 |
|  |  |  |  |  |  |  |  |  |  |  |
| None | 0,86 | 156 | 1,95 | 98 | 0,82 | 169 | 0,61 | 122 | 3,40 | 89 |
| $\{K_3, K_4, K_5\}$ | 0,57 | 102 | 1,78 | 97 | 0,44 | 92 | 0,66 | 105 | 2,80 | 89 |