# OpenReview forum: "Graph Neural Networks with Local Graph  Parameters"
_NeurIPS.cc/2021/Conference — NeurIPS 2021 Poster_

### Official Review · Reviewer_t8ms · 2021-07-14

**Rating:** 7
**Confidence:** 2

**Summary:**

This paper proposes local graph parameter enabled GNNs as a framework for studying the local higher-order graph structural information. Local graph parameters can be added to any GNN architecture, and are cheap to compute. In terms of expressive power, their proposal lies in the middle of GNNs and their higher-order counterparts. Further, they propose several techniques to aide in choosing the right local graph parameters. The experimental evaluation shows that adding local graph parameters often has a positive effect for a variety of GNNs, datasets and grpah learning tasks.


**Limitations And Societal Impact:**

Yes, this work mentions the limitations in the main paragraph

**Main Review:**

For the main part, as a resubmission from ICML, this work fixed the issues as they mentioned in the last submission. The overall quality of the current version is solid and it can deliver  the main idea of F-MPNN method. And this work can inherit the merits from arbitrary GNNs which makes the value of this method more general

For the minor part, this work still has typos and need further proof-read. For example, P2 line 51.

**Time Spent Reviewing:**

4

---

> ### Author Response · Authors · 2021-08-10
> **Answer to Reviewer #4**
>
> We thank the reviewer for the positive review and appreciate that the reviewer sees merits in the broad applicability, as a plug-in for arbitrary GNNs, of our proposal. We will address the reviewer’s minor comment on typo’s in the final paper.

---

### Official Review · Reviewer_wpY3 · 2021-07-16

**Rating:** 7
**Confidence:** 4

**Summary:**

This paper provides a theoretical framework attempting to unify a variety of recent methods in the Graph Neural Network literature, which all follow a similar conceptual approach: they introduce extra structural features in the input of the Neural Network that cannot be computed by traditional message passing. The main formalism underlying the framework is that of homomorphism, which is an adjacency-preserving mapping between two graphs. Differently from subgraph isomorphism, homomorphisms are not necessarily bijective (think of walks vs paths).  The authors define $\mathcal{F}$-MPNNs as a message passing framework where node features are enhanced with the homomorphism counts of every pattern in a collection $\mathcal{F}$. The authors derive a deep characterization of the expressive power of $\mathcal{F}$-MPNNs, both in comparison to the k-WL hierarchy and between different instantiations thereof,  show how it can incorporate other approaches, and answer numerous research questions that have been left open by the most related paper, that of Bouritsas et al., arxiv’20. The method is also validated experimentally, by enhancing GNNs with homomorphism counts of cycles and cliques, on node, edge and graph level tasks, showing consistent improvement against various conventional MPNN counterparts.

**Limitations And Societal Impact:**

*Limitations*: The limitations are clearly stated in section 1, by mainly referring to the fact that the patterns need to be selected by hand. I would also add a discussion on the computational complexity of homomorphism counting.

*Negative societal impact*: A satisfactory discussion is included in the end of the experimental section.

**Main Review:**

Although injecting structural features in the input of GNNs is common in the GNN community, usually, the theoretical justifications are limited in showing improvements over the 1-WL, while sometimes the structural features are chosen in an ad-hoc way. However, this paper goes much deeper in the theoretical justifications showing that homomorphisms allow a more accurate characterisation of expressivity, while most importantly, using the observation that different types of subgraph counting (motifs, graphlets, homomorphisms…) can be derived from one another via linear combinations (e.g., see  Borgs et al., “Counting Graph Homomorphisms” Topics in Discrete Mathematics, 2006), they motivate why homomorphisms can be an elegant and unifying framework.

I must admit that it is a bit hard to follow for non-experts due to the utilisation of many specialised notions in graph theory, so there is room for improvement in that respect. Specific comments follow:

**Strengths**:

- I found this paper quite useful to the community, bringing important ideas from graph theory to graph neural networks, and shedding light on many open questions in the expressive power of this class of GNNs. I also appreciate the fact that the authors are honest w.r.t. the main limitation of the work, i.e. the pattern selection.
- The paper addresses a number of open research questions described in GSNs (although not directly applicable to GSNs, the corresponding results on $\mathcal{F}$-MPNNs provide important insights to the reader):
     - How do substructure-enabled MPNNs relate to k-WL (Proposition 2 and 3)?
     - What type of patterns grant F-MPNNs the ability to distinguish graphs that k-WL cannot (cores of treewidth k – Theorem 2; this is in agreement with the 4-clique counterexample used in the GSN paper)?
     - In which cases can we be sure that expressivity increases when adding patterns to the collection (Propositions 4,5 and Theorem 3)? Moreover, Proposition 5 implies a compositionality property of certain patterns, i.e., when two patterns $P_1$, $P_2$ are added to the collection, expressivity increases more than by simply adding their composition $P_{1,2}$. I am still not sure though if this implies that a $\\{P_1, P_2\\}$-MPNN can count $P_{1,2}$ homomorphisms.
-	The experimental results are diverse, spanning all types of graph-related problems, showing consistent improvement against the conventional MPNN baselines.


**Weaknesses**:
-	In my opinion, the paper is a bit hard to follow. Although this is expected when discussing more involved concepts, I think it would be beneficial for the exposition of the manuscript and in order to reach a larger audience, to try to make it more didactic. Some suggestions:
    -	A visualization showing a counting of homomorphisms vs subgraph isomorphism counting.
    - It might be a good idea to include a formal or intuitive definition of the treewidth since it is central to all the proofs in the paper.
    - The authors define rooted patterns (in a similar way to the orbit counting in GSN), but do not elaborate on why it is important for the patterns to be rooted, neither how they choose the roots. A brief discussion is expected, or if non-rooted patterns are sufficient, it might be better for the sake of exposition to discuss this case only in the supplementary material.
-	The authors do not adequately discuss the computational complexity of counting homomorphisms. They make brief statements (e.g., L 145 “Better still, homomorphism counts of small graph patterns can be efficiently computed even on large datasets”), but I think it will be beneficial for the paper to explicitly add the upper bounds of counting and potentially elaborate on empirical runtimes.
-	Comparison with GSN: The authors mention in section 2 that F-MPNNs are a unifying framework that includes GSNs. In my perspective, given that GSN is a quite similar framework to this work, this is an important claim that should be more formally stated. In particular, as shown by Curticapean et al., 2017, in order to obtain isomorphism counts of a pattern P, one needs not only to compute P-homomorphisms, but also those of the graphs that arise when doing “non-edge contractions” (the spasm of P). Hence a spasm(P)-MPNN would require one extra layer to simulate a P-GSN. I think formally stating this will give the interested reader intuition on the expressive power of GSNs, albeit not an exact characterisation (we can only say that P-GSN is at most as powerful as a spasm(P)-MPNN but we cannot exactly characterise it; is that correct?)
-	Also, since the concept of homomorphisms is not entirely new in graph ML, a more elaborate comparison with the paper by NT and Maehara, “Graph Homomorphism Convolution”, ICML’20 would be beneficial. This paper can be perceived as the kernel analogue to F-MPNNs. Moreover, in this paper, a universality result is provided, which might turn out to be beneficial for the authors as well.



**Additional comments**:
- I think that something is missing from Proposition 3. In particular, if I understood correctly the proof is based on the fact that we can always construct a counterexample such that F-MPNNs will not be equally strong to 2-WL (which by the way is a stronger claim). However, if the graphs are of bounded size, a counterexample is not guaranteed to exist (this would imply that the reconstruction conjecture is false). Maybe it would help to mention in Proposition 3 that graphs are of unbounded size?
-	Moreover, there is a detail in the proof of Proposition 3 that I am not sure that it’s that obvious. I understand why the subgraph counts of $C_{m+1}$ are unequal between the two compared graphs, but I am not sure why this is also true for homomorphism counts.
-	Theorem 3: The definition of the core of a graph is unclear to me (e.g., what if P contains cliques of multiple sizes?)
-	In the appendix, the authors mention they used 16 layers for their dataset. That is an unusually large number of layers for GNNs. Could the authors comment on this choice?
-	In the same context as above, the experiments on the ZINC benchmark are usually performed with either ~100K or 500K parameters. Although I doubt that changing the number of parameters will lead to a dramatic change in performance, I suggest that the authors repeat their experiments, simply for consistency with the baselines.
-	The method of Bouritsas et al., arxiv’20 is called “Graph Substructure Networks” (instead of “Structure”). I encourage the authors to correct this.

-------
-------
### After rebuttal

The authors have adequately addressed all my concerns. Enhancing MPNNs with structural features is a family of well-performing techniques that have recently gained traction. This paper introduces a unifying framework, in the context of which many open theoretical questions can be answered, hence significantly improving our understanding. Therefore, I will keep my initial recommendation and vote for acceptance. Please see my comment below for my final suggestions which, along with some improvements on the presentation, I hope will increase the impact of the paper.


**Time Spent Reviewing:**

12

---

> ### Author Response · Authors · 2021-08-10
> **Answer to Reviewer #3**
>
> We would like to thank the reviewer for his detailed and positive feedback.
>
> **Presentation.**
> The reviewer makes a couple of suggestions to explain certain complex concepts in a more didactic way. We thank the reviewer for these suggestions, which we will take on board when preparing the final version. The reviewer also would like to see a brief discussion related to the use of  rooted patterns. The root in the pattern tells how we want to embed the pattern locally around vertices in the data graph. We note that for symmetric patterns such as cliques or cycles, which we use in the experiments, the choice of root vertex is irrelevant. For asymmetric patterns, or unrooted patterns, we recommend, similarly as done in GSNs, to use copies of the pattern with different roots (one copy for each different vertex orbit in the pattern). Alternatively, one could use an unrooted pattern but then only count those homomorphisms whose image contains the vertex in the data graph. We do not know for certain how this model compares to the rooted one, but we speculate it is less expressive, as we are always aggregating together all rooted versions of the patterns. A discussion related to rooted versus unrooted patterns will be included in the final paper. Finally, we fix "Graph Structure Networks" to "Graph Substructure Networks", thanks for pointing this out.
>
> **On the comparison with GSNs.**
> The reviewer correctly observes that $\mathcal P$-GSNs are bounded by $\mathsf{spasm}(\mathcal P)$-MPNNs, as we also explain in the supplementary material. We will put this claim as a formal statement in the main paper, as suggested by the reviewer. As a side note, we can obtain a precise characterization of GSNs in our framework if we specialize the notion of homomorphisms/isomorphism of $\mathcal P$-trees so that they embed the backbone of a $\mathcal P$-tree homomorphically, but the $\mathcal P$-patterns in the leaves of the tree to isomorphic subgraphs. For clarity of exposition, we decided to stick to the standard notion of homomorphisms, at the expense of moving from $\mathcal P$ to $\mathsf{spasm}(\mathcal P)$ in the comparison between GSNs and $\mathcal F$-MPNNs
>
> **On the use of homomorphisms.**
> The reviewer would like to see a theoretical comparison between the use of homomorphism counts in $\mathcal F$-MPNNs and and in the Graph Homomorphism Convolutions (GHCs) by NT and Maehara, cited in the paper. There are indeed interesting connections: Whereas the GHCs use only homomorphism counts of patterns in $\mathcal F$ (followed by the use of an SVM), $\mathcal F$-MPNNs expand the set $\mathcal F$ of patterns to $\mathcal F$-pattern trees (Theorem 1 in our paper). We provide an example in the supplementary material of two graphs, sharing the same homomorphism counts for a set $\mathcal F$ of patterns, but having different homomorphism counts for $\mathcal F$-pattern trees of depth $>1$. In combination with the techniques by NT and Maehara, one could therefore possibly learn functions beyond those that are $\mathcal F$-invariant. We will elaborate on this interesting connection in the final paper.
>
> **On the complexity of counting homomorphisms.**
> The reviewer suggest to include theoretical upper bound for counting homomorphisms, as suggestion also made by reviewer #1. The worst case complexity of computing $\mathsf{hom}(P,G)$ is $\mathcal O(n^k)$ where $n$ is the size of $G$ and $k$ is the size of $P$. Better upper bounds can be obtained when a tree-decomposition of the pattern $P$ is provided. Despite the exponential dependency on $k$, the system that we use to compute our additional features  (DISC, Zhang et al. 2020) leverages state-of-the-art distributed join processing algorithms and in this way, makes counting homomorphisms feasible for small patterns and large scale datasets. And indeed, on our selected datasets, this cost was negligible compared to the cost incurred by learning the parameters. For Collab, a dataset consisting of one large graph of 236k nodes and 2358k edges, the counting of 3,4 and 5-cliques homomorphisms for all nodes took 13s , 8s and 34s respectively. Other datasets were large collections of small graphs and homomorphism counts can be obtained at the time of loading the graphs.
>
> **Clarifications related to the statement of Proposition 3.**
> The reviewer requires some additional explanation related to the statement of Proposition 3.
> To avoid any confusion, we will clarify that the proof of Proposition 3 provides counterexample graphs of sizes that are dependent on the sizes of patterns in $\mathcal F$. We note, however, that the statement of Proposition 3 is correct. It states that, for any $k>1$, there does not exist a finite set $\mathcal F$ of patterns such that $\mathcal F$-WL is equivalent in expressive power to $k$-WL. To show this, it suffices to exhibit, for any $k>1$ and finite set $\mathcal F$ of patterns, two graphs $G$ and $H$, such that $k$-WL can distinguish them, whereas $\mathcal F$-WL cannot.
> We remark that no restriction on the sizes of  $G$ and $H$ are implied (or required).
> Furthermore, we will clarify that it suffices to show the Proposition for $k=2$, as observed by the reviewer, since this is how Proposition 3 is proved in the supplementary material.
>
> The reviewer further asks for some clarifications on the proof of Proposition 3. We thank the reviewer for checking the proofs in the supplementary material. We will add the following clarification to the supplementary material.
> Recall from the proof that $G$ consists of $m+2$ disjoint copies of $C_{m+1}$ and $H$ consists of $m+1$ copies of $C_{m+2}$.
> Consider $\mathsf{hom}(C_{m+1},G)$ and $\mathsf{hom}(C_{m+1},H)$.
> We can write these as
> $(m+2)(m+1)\mathsf{hom}(C_{m+1}^r,C_{m+1}^v)$ and $(m+1)(m+2)\mathsf{hom}(C_{m+1}^r,C_{m+2}^w)$  for some fixed vertices $v$ and $w$ and where all cycles are now rooted.
> It now suffices to observe that all homomorphisms $h$ from $C_{m+1}^r$ to $C_{m+2}^w$ are such that the image $h(C_{m+1}^r)$ contain $< m+1$ vertices. And moreover, with any such $h$, we can associate a unique $h’$ from $C_{m+1}^r$ to $C_{m+1}^v$ (such that also $h’(C_{m+1}^r)$ contains $<m+1$ vertices). Finally, we note that there are two homomorphisms from $C_{m+1}^r$ to $C_{m+1}^v$ which are surjections and thus cover all $m+1$ vertices in $C_{m+1}^v$.  We may thus conclude that $\mathsf{hom}(C_{m+1}^r,C_{m+2}^w) < \mathsf{hom}(C_{m+1}^r,C_{m+1}^v)$, from which we can infer that $\mathsf{hom}(C_{m+1},G)\neq \mathsf{hom}(C_{m+1},H)$, as desired.
>
> **Definition of core of a graph.**
> The reviewer asks for a clarification of the definition of a core of a graph. Many thanks for pointing this out: Our current definition is wrong as we forgot to add an important condition, i.e., that the core of $P$ must be a homomorphic image of $P$ (in addition to simply being a core of itself). This is precisely what causes the reviewer's confusion regarding the case when $P$ is the union of two cliques, say $P = K_3 + K_4$. The core of $P$ is $K_4$, as there is a homomorphic image from $P$ into it and $K_4$ is itself a core. In turn, $K_3$ is not a core of $P$ as there is no homomorphism from $P$ into $K_3$. We will fix this of course.
>
> **Experiments**
> The reviewers asks for a comment on the use of 16 layers and parameter budgets used in some of the experiments. Our experiments are based on the benchmarking initiative by Dwivedi et al. For the ZINC and PATTERN datasets, the performance of GNNs with ~100k parameters and ~500k parameters is reported by Dwivedi et al, which correspond respectively to 4 and 16 message passing layers. As the latter showed a stronger performance in all cases we focused on augmenting these larger models with the additional features as it was our intention to compare against the strongest baseline for these benchmarks. To have a full comparison we will repeat our experiments by setting the number of message passing layers to 4 and corresponding ~100k parameter budget. A baseline with the same parameters and 4 layers will also be included in the final paper.
> We also would like to emphasize that we trained the baseline models ourselves instead of just reporting the numbers of Dwivedi et al. This was done to discard any possible minor differences in hyperparameter choices between our implementation and the implementations of Dwivedi (due to different versions, unmentioned details etc.) and in this way ensure a completely fair comparison.

---

> > ### Comment · Reviewer_wpY3 · 2021-08-26
> > **Final suggestions after reading the rebuttal**
> >
> > I am happy with the authors’ response since all my concerns were addressed. Homomorphism-enabled MPNNs provide a theoretically elegant framework to explain MPNNs enhanced with structural features. Although by examining the reported results, homomorphisms do not seem to have a clear practical advantage over subgraph isomorphisms in terms of performance, they are amenable to a deeper theoretical analysis and understanding and they seem faster to compute. As in GSNs, selecting the subgraph patterns is undeniably a limitation and this remains an open question for the research community.
> > Some final suggestions for the updated version:
> > - I am not sure I understand the procedure to obtain the precise characterisation of GSNs. However, since the authors seem to be confident about the proof technique, personally I would be interested to see that in an updated version of their manuscript, since this will provide an answer to an open research question and is thus of independent theoretical importance. Moreover, the analogy between rooted patterns in homomorphism counts and orbits in subgraph isomorphism counts should be highlighted.
> > - I advise the authors to include the discussion on the computational complexity in the final version since the topic is of practical importance. Please also provide a table/figure with more details for different graph distributions.
> > - A final suggestion regarding the experiments with 500K and 100K parameters on ZINC: Since ZINC is now a standard benchmark in the community, maybe the authors would like to include a detailed table with all the competitors in their final version. The obtained results are competitive with the state-of-the-art, hence this will increase the impact of their work. For example, see here https://paperswithcode.com/sota/graph-regression-on-zinc-100k and here https://paperswithcode.com/sota/graph-regression-on-zinc-500k and in the GSN paper for a more extensive list of results.

---

> > > ### Author Response · Authors · 2021-08-30
> > > **Response to final suggestions by Reviewer #3**
> > >
> > > We thank the reviewer for the final suggestions. Just to clarify a bit more on how a precise characterization of GSNs can be obtained:
> > > The idea is to define a special kind of homomorphism  $h:T^r\to G^v$ from (rooted) pattern trees $T^r$ to (rooted) graphs $G^v$ such that:
> > > * (i) $h$ is a standard homomorphism from the backbone of $T^r$ to $G^v$; and
> > > * (ii) for each single pattern $P^s$ in a leaf $s$ in $T^r$, $h(P^s)$ should be an isomorphic copy of $P^s$ attached to $h(s)$.
> > >
> > > Condition (ii) implies that when there are two patterns $P^s$ and $Q^s$ in a leaf, $h(P^s)$ and $h(Q^s)$ are isomorphic copies of $P^s$ and $Q^s$, independently. Similarly for when more leaf patterns are present. Then, if we denote by $\mathsf{shom}(T^r,G^v)$ the number of such special homomorphisms, then one can verify that this function inherits all properties from $\mathsf{hom}(T^r,G^v)$ on which we rely in the proof of Theorem 1. In particular, the $\mathsf{shom}$ of a product of patterns in a leaf becomes the product of $\mathsf{shom}$’s of the individual patterns by condition (ii). At the same time, we can still inductively compute $\mathsf{shom}$ by descending the pattern tree via its backbone because of assumption (i). Since, as the reviewer mentions, this may of independent theoretical interest, we will mention this in the paper and add details to the supplementary material.

---

### Official Review · Reviewer_pwMn · 2021-07-17

**Rating:** 6
**Confidence:** 3

**Summary:**

This paper proposes a novel GNN framework, F-MPNN, to efficiently increase the expressive power of MPNNs. The proposed framework uses local higher-order graph parameters which are the homomorphism counts of small patterns (graph sets). The adoption of different patterns will result in different MPNNs.

From a theoretical perspective, the expressive power of F-MPNN is characterized based on the WL algorithm, and is compared to higher-order MPNNs. Since the performance of F-MPNN depends on the selection of patterns, how different patterns affect the expressive power is also investigated.

The theoretical findings are further validated via extensive experiments with various GNNs, datasets, and graph-related tasks. The results demonstrate the promising improvements achieved by using the proposed efficient local graph parameters.

**Limitations And Societal Impact:**

The authors have addressed the limitations of their work in Introduction.

**Main Review:**

The proposed framework successfully unifies several existing extensions of MPNNs with solid theoretical analysis. The connections and comparisons between the proposed F-MPNNs and higher-order MPNNs characterize the expressive power of related models with more insights and details. Besides, the effects of different selection of patterns on the expressive power are also investigated, which helps to better understand the power of the proposed model.

The experiments are done with broad coverage of models, datasets, and tasks to evaluate the performance of the proposed method. The results are discussed in detail and the promising performance of the proposed methods is validated clearly.

The paper is overall well organized. However, the authors should double-check the grammar if accepted:

Line 14: aide --> aid

Line 17: effect for --> effect on

Line 161: the the --> the

Line 221: in that more --> in more

Line 222: be distinguish --> be distinguished

Line 230: less rounds --> fewer rounds

Line 368: have greatest impact for --> have the greatest impact on

Line 401: deserves --> deserve





**Time Spent Reviewing:**

3

---

> ### Author Response · Authors · 2021-08-10
> **Answer to Reviewer #2**
>
> We thank the reviewer for the positive review and appreciate that the reviewer finds our paper well-structured, likes the unified approach that F-MPNNs provide, and recognizes the value of our theoretical analysis and broad coverage of the experimental setting. We will address the reviewer’s only criticism related to grammar-checking in the final paper and thank the reviewer for the list of typo’s.

---

### Official Review · Reviewer_vuNp · 2021-07-21

**Rating:** 5
**Confidence:** 4

**Summary:**

- This paper proposes F-MPNN, which extends the expressive power of MPNN whilst preserving their O(n) cost in each iteration.
- This paper provides some theoretical guarantees for F-MPNNs.
-- It shows that F-MPNN can be at most as expressive as F-WL-test
-- It compares the expressive power between F-WL-test and k-WL-test.
- Certain improvements are shown in experiments.


**Limitations And Societal Impact:**

See above.

**Main Review:**

Pro:
- It is a smart idea to utilize the design of the F-WL-test to design the F-MPNN, so that this type of GNN can enjoy the theoretical guarantee of distinguishing power as what the F-WL-test has.
- This method is less expensive than some higher-order MPNN in terms of the complexity in each iteration.

Cons:
- The proofs heavily depend on existing works but are not suitably mentioned in the main text. I think it is totally fine to make a statement in the main text because a novel combination of existing proof techniques is also valuable.
- As already mentioned by the authors, in practice it is hard to pre-select a suitable set of patterns since they cannot be learned by gradients. The theoretical results in Sec 4 do provide certain information but they are not enough to serve as practical guidance since the search space is just way too large.
- Experiments can be conducted more rigorously (see below).

Detailed questions for experiments:
- By adding the patterns, the dimension of the augmented node features will become larger and the model will contain more parameters. When making the comparison, have all models used the same feature dimension and the same number of parameters? This should be controlled to exclude the effect of using a larger model. (e.g., the results in Table 1 (b))
- The key baseline in this paper is the ISO in Table 1 (a). Why results of ISO are missing in Table 2 and Table 3? If one also searches the best performing pattern for ISO using the same computing cost as the search for F-MPNN, what will be the best performing accuracy for ISO on PATTERN and COLLAB?
- Except for ISO, there are several other works that also augment the initial feature with precomputed node features. Some simple ones include random feature [1], port numbering / weak-coloring feature [2], local matrix feature [3]. I noticed that these three are also cited in related work. It will be better to see a comparison to these simple augmented approaches.
- Minor question: The hyperparameters of the GNNs are not described. What's the hidden size? Number of hops? Will different choices of these hyperparameters affect the improvement from augmenting the features?

Other question:
- Although it is a one-time preprocessing stage, how is the complexity?

Summary:
- This paper provides theoretical guarantees for the proposed F-MPNN in its distinguishing power, which is valuable. However, since the theory is insufficient to provide practical guidance for choosing a suitable set of F patterns, I think the empirical experiments become important to support the claims in this paper. Therefore, my main concern is that the experiments are not convincing enough (see details above).

[1] Sato,  R.,  Yamada,  M.,  and  Kashima,  H. Random features strengthen graph neural networks.arXiv preprintarXiv:2002.03155, 2020.

[2] Sato, R., Yamada, M., and Kashima, H. Approximation ratios of graph neural networks for combinatorial problems.In Advances in Neural Information Processing Systems, pp. 4081–4090, 2019.

[3] Vignac, Clément, Andreas Loukas, and Pascal Frossard. "Building powerful and equivariant graph neural networks with structural message-passing." NeurIPS. 2020.



**After reading the response**

Thank the authors for the detailed response! I appreciate the theoretical contribution of this paper, but I still have concerns about its practical usage, without the experimental comparison to other augmented GNN models. The authors claimed that this method can extend any existing GNNs, including those augmented GNNs. However, although the paper has theoretically shown the improvement in expressiveness compared to MPNN, the improvement to existing *augmented* MPNNs [1,2,3] are unclear, which will require additional experiments to verify. Moreover, these augmented GNNs [1,2,3] and some other alternatives that use distance-based features are less expensive (pre-processing) and do not require choosing the patterns (that means fewer tuning parameters). How useful is the proposed method compared to these **simpler** methods (which also have theoretical guarantees) is unclear. This is the major reason why I choose to retain my initial score. I personally believe this will be a very nice paper if additional experiments with convincing results are provided.


**Time Spent Reviewing:**

4

---

> ### Author Response · Authors · 2021-08-10
> **Answer to Reviewer #1**
>
> We thank the reviewer for the constructive feedback.
>
> **On the presentation of proof techniques.**
> We appreciate that the reviewer recognizes that our proof techniques considerably extend and combine existing techniques in a novel way. We do cite all the relevant papers, in which the techniques used can be found, alongside our results in the main paper, and hint how we extend and combine the techniques. We are brief, mainly due to space limitations, and pushed many details to the supplementary material. The reviewer would find it valuable to have more details in the main paper. We will provide more details in the final paper as these are indeed part of our main contributions.
>
> **On the selection of patterns.**
> The reviewer correctly points out that in practice a suitable set of patterns needs to be selected, either by experts based on domain knowledge, or by suitable feature engineering. However, we think our paper and in particular the results in Section 4 in the paper, do provide practical guidance: it is a good idea to start by selecting small patterns that we know (by Proposition 4 and Theorem 3) cannot be distinguished by standard MPNNs, such as triangles, 4-cliques or 4-cycles. This guidance comes from theory (Proposition 5 tells us that by adding such patterns, more complex graph structures obtained by combining the simple patterns are captured as well), and also by experiments: all datasets seem to gain some benefit when incorporating those simple patterns. Notice also that Proposition 5 also helps us reduce the search space by a great extent: For example, once we add triangles, we no longer need to search for any pattern that incorporates a triangle at the root.
> Finally, we also want to stress that our experiments provide preliminary evidence that we don’t really need to look for *the* most informative pattern in the dataset, just adding standard patterns already provides a considerable boost in the learning power.
> We agree, however, that questions related to pattern selection open up very interesting pathways for future work, as we mention also in our Conclusion section.
>
> **On the complexity of counting homomorphisms.**
> The reviewer asks for the complexity of counting homomorphisms, a request also made by reviewer #3. The worst case complexity of computing $\mathsf{hom}(P,G)$ is $\mathcal O(n^k)$ where $n$ is the size of $G$ and $k$ is the size of $P$. Better upper bounds can be obtained when a tree-decomposition of the pattern $P$ is provided. Despite the exponential dependency on $k$, the homomorphism counting problem is the most common task in databases, and in particular graph databases, and there is a broad range of literature, algorithms and systems capable of performing homomorphism counts in very reasonable times. Indeed, the system that we use to compute our additional features  (DISC, Zhang et al. 2020) leverages state-of-the-art distributed graph join processing algorithms and in this way, makes counting homomorphisms feasible for small patterns and large scale datasets. On our selected datasets, this cost was negligible compared to the cost incurred by learning the parameters, as mentioned in the paper. We emphasize that if one would use higher-order GNNs, which may be able to detect complex graph patterns, each of their layers requires $\mathcal O(n^k)$ complexity, because they operate on $k$-dimensional tensors. In our setting, we pay the $\mathcal O(n^k)$ price only once and can leverage efficient state-of-the-art implementations.
>
> **Experiments.**
>
> __*On the feature dimensions.*__  The reviewer asks for the differences in sizes of the models, when augmented with additional features. For a given model, *only* the  feature dimension (and consequently the number of parameters) in the first layer vary. This is needed in order to accommodate for the varying number of homomorphism count parameters (that is, our additional features). For all other layers, feature dimensions and the number of parameters in any message-passing and readout layers are identical for a given model, independent of whether or not the initial layer contains additional homomorphism count parameters.
>
> We emphasize, however, that the additional number of parameters in the first layer is only a very small fraction of the total number of parameters, as reported in Tables 5, 8, 11 and 14 in the supplementary material. For example, for Table 1(b), reporting the GAT architecture on the ZINC dataset, the baseline (no additional features) has a total of 358273 parameters, the model with all cycles up to length 10, has only 1152 extra parameters. The differences are smaller for the other architectures. For example, the GatedGCN model has 408135 parameters for the baseline and 408695 for the model with all cycle counts, a difference of 560 additional parameters.
>
> Furthermore, for MPNNs, the number of parameters in the first layer is directly related to the number of features used. So, say that the baseline contains $f$ features, and one of the other models uses $f+g$ features ($g$ being the number of homomorphism counts added). Then, to obtain the same number of parameters, we need to add $g$ features to the baseline as well. It is unclear to us what precisely these $g$ additional features should be.
> We also note that the additional homomorphism count features cannot be learned from the baseline, even if we augment these with arbitrarily many dummy (constant, copies of existing) features. This is due to the limited power of MPNNs.
>
> __*Missing ISO counts for PATTERN and COLLAB.*__  We thank the reviewer for this comment. The reason that subgraph isomorphisms counts (ISO) are not explicitly mentioned for PATTERN and COLLAB is because for these datasets the additional features are homomorphism counts (HOM) of small cliques. For cliques, HOM and ISO on any graph only differ by a linear factor (the number of automorphisms of the clique). This means that, after normalization, there is no difference in adding HOM or ISO counts of cliques as features. For this reason, ISO is excluded. We do mention this in Section 6 “We remark that for cliques, homomorphism counts coincide with subgraph isomorphism counts (up to a constant factor)”, but will emphasize this even more in the final paper.
>
> __*Parameters of GNNs.*__ The reviewer asks for details about the models used. We use the parameters as used in the publicly available benchmark implementation by Dwivedi et al. and therefore only state hyperparameters for every graph task in the experimental section of the supplementary material (Section E.2). For the number of layers, following the best performing benchmarks for every model we use 16 layers for ZINC and PATTERN and 3 layers for COLLAB. The size of the hidden features vary depending on the selected GNN model. This variation comes from the fact that the comparison of different GNNs in the benchmark is done with a similar parameter budget for all models. We will include the hidden feature sizes in section E.2. in the supplementary material, but provide the table for hidden feature dimensions below, for the reviewer's convenience.
> We remark that we did not perform any hyperparameter tuning, as our focus was to identify the effect of the additional features and select features from the set of small cycles or cliques rather than finding the optimal choice of hyperparameters for a given graph task.
>
> Hidden feature dimension table:
>
> |           | ZINC | COLLAB | PATTERN |
> |-----------|------|--------|---------|
> | GAT       | 144  |   57   | 136     |
> | GCN       |  145 | 74 | 146   |
> | GraphSage |    108 | 38 | 108  |
> | MoNet     |   90 | 53 | 90  |
> | GatedGCN  |    70 | 35| 70    |
>
> __*Additional comparisons.*__ The reviewer suggests including comparisons with three approaches that are cited in our paper. We would like to emphasise that our goal is not to compare directly to other approaches, but instead investigate how much existing methods can benefit from adding our extended features (homomorphism counts of patterns). Indeed, we believe that one of the important aspects of our proposal is that it provides a generic plug-in that can be used for any method, including the ones mentioned by the reviewer. This is why we chose to use the benchmark initiative by Dwivedi et al. in our experiments, as it gave us a common starting ground to report the improvement on several MPNN architectures. Regarding a more in-depth comparison with other approaches such as [1], [2] or [3]: we can obtain an approximate comparison by looking at other approaches that have used the same datasets in their experiments (such as [3], which tests on the ZINC dataset). In this approximate comparison, we also observe that the models in the benchmark become competitive to the state of the art when augmented with homomorphism counts. However, we prefer not to put this numbers in the paper because it takes us away from our original goal of investigating how much existing methods can benefit from adding our extended features.

---

### Author Response · Authors · 2021-08-10
**General response**

We thank the reviewers for their constructive feedback. All reviewers appreciate our augmentation technique which is applicable to any GNN model, find our theoretical analysis interesting and observe that our experiments cover a broad set of models. The reviewers make suggestions as to how the presentation of the technical results can be improved and made more didactic. We are happy to take these suggestions on board. Moreover, since the preprocessing needed to compute the additional features is crucial in our proposal, two reviewers ask for more details on the complexity of the preprocessing phase. This will be addressed in the paper. Similarly, clarifications related to our results, graph concepts and proofs will be added. An important aspect to our approach is that the choice of patterns defining our additional features is a difficult task. Several of our theoretical results point in the direction of guiding this task, and the reviewers do appreciate this guidance. A reviewer points out that these results alone may not suffice, and indeed, additional domain knowledge may be needed. We agree, but also point out that for many graph datasets, simple patterns such as cliques or cycles, suffice, and this is also verified in the experiments. We base our experiments on a recent benchmark initiative and moved details (hyperparameters, layers,...) to the supplementary material. We will move some of the details back to the main paper, as suggested by the reviewers. Our emphasis in the experiments lies on showing that the models in the benchmark can substantially benefit from our feature augmentation. In the process of doing so, we arrive at some MPNN models that are competitive with state of the art approaches; we chose not to report on this issue because i) it is not the goal of our experiments, and ii) the comparison we can make is only through numbers reported in other work, and this is not ideal. We do expect to see future work using our augmentation technique to produce better practical results, and we see this as one of the strengths of our approach.

We provide detailed responses to the reviewers’ comments below.

---

### Decision · Program_Chairs · 2021-09-27

**Decision:**

Accept (Poster)

**Comment:**

This paper unifies several classes of recently proposed graph neural network architectures via the notion of graph homomorphisms. The reviewers agree that presented theoretical framework is novel. Even more importantly, it can be successfully applied to propose new GNN architectures. The paper is well written and provides rigorous mathematical analysis of several GNN methods that were previously analyzed only empirically.